# OmniEarth-Bench: Towards Holistic Evaluation of Earth's Six Spheres and Cross-Spheres Interactions with Multimodal Observational Earth Data

Fengxiang Wang[1,2], Mingshuo Chen[3], Xuming He[4], Yi-Fan Zhang, Feng Liu[2], Zijie Guo[2], Zhenghao Hu[5], Jiong Wang[2] Jingyi Xu[2], Zhangrui Li[2], Fenghua Ling[2], Ben Fei[2], Weijia Li[5], Long Lan,[1] Wenjing Yang[1]*, Wenlong Zhang[2]*, Lei Bai[2]

[1] NUDT [2]Shanghai AI Lab [3] BUPT [4] ZJU [5] SYSU [6] FDU

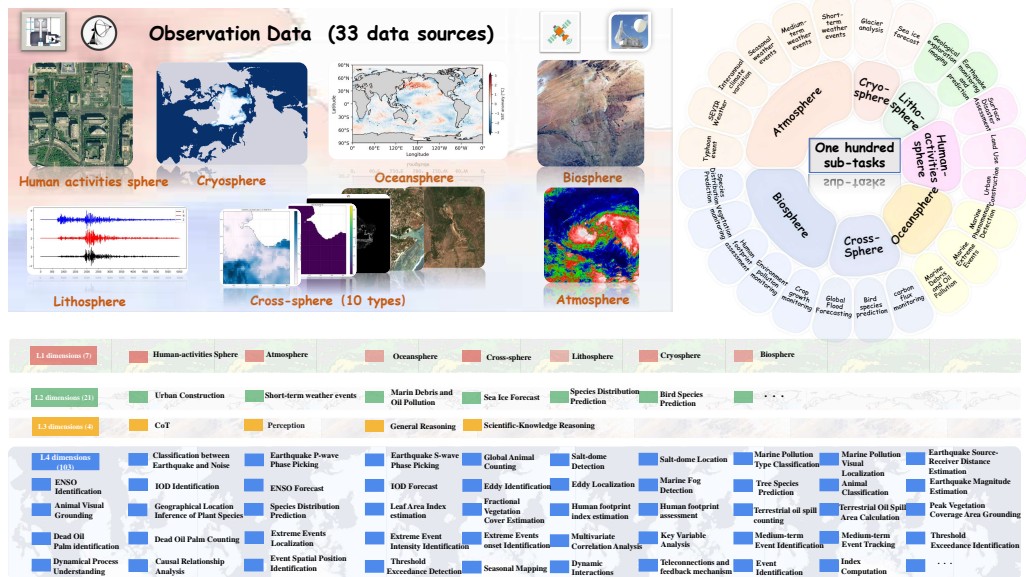

Figure 1: **Overview of OmniEarth-Bench.** Our benchmark spans six Earth science spheres and cross-sphere scenarios, encompassing 100 sub-tasks derived from 33 sensor types.

## Abstract

Existing benchmarks for Earth science multimodal learning exhibit critical limitations in systematic coverage of geosystem components and cross-sphere interactions, often constrained to isolated subsystems (only in Human-activities sphere or atmosphere) with limited evaluation dimensions ($\leq$ 16 tasks). To address these gaps, we introduce **OmniEarth-Bench**, the first comprehensive multimodal benchmark spanning all six Earth science spheres (atmosphere, lithosphere, Oceansphere, cryosphere, biosphere and Human-activities sphere) and cross-spheres with one hundred expert-curated evaluation dimensions. Leveraging observational data from satellite sensors and in-situ measurements, OmniEarth-Bench integrates 29,779 annotations across four tiers: perception, general reasoning, expert-knowledge deductive reasoning and chain-of-thought (CoT) reasoning. This involves the efforts of 2–5 experts per sphere to establish authoritative evaluation dimensions and curate relevant observational datasets, 40 crowd-sourcing annotators to assist experts for annotations, and finally, OmniEarth-Bench is validated via hybrid expert-crowd workflows to reduce label ambiguity. Experiments on 9 state-of-the-art MLLMs

---

*Corresponding authors

Submitted to 39th Conference on Neural Information Processing Systems (NeurIPS 2025). Do not distribute.

reveal that even the most advanced models struggle with our benchmarks, where none of them reach 35% accuracy. Especially, in some cross-spheres tasks, the performance of leading models like GPT-4o drops to 0.0%. OmniEarth-Bench sets a new standard for geosystem-aware AI, advancing both scientific discovery and practical applications in environmental monitoring and disaster prediction. The dataset, source code, and trained models were released at OmniEarth-Bench.

# 1 Introduction

Earth scientists address critical environmental and societal challenges through modeling Earth's interconnected systems [1]: the atmosphere, lithosphere, hydrosphere, cryosphere, biosphere, and human activities [2]. By analyzing cross-system interactions, researchers derive impactful findings such as flood prediction [3], a complex task requiring multi-domain expertise (e.g., atmospheric precipitation, biospheric soil moisture, and lithospheric runoff). These discoveries are systematically validated in high-impact journals including Nature and Science [4, 5, 6, 7, 8, 9].

Existing MLLMs (e.g., GPT-4o [10], Gemini [11] and Claude [12]) excel at considerable tasks and have motivated benchmarks that explicitly test core skills. These benchmarks span diverse evaluation dimensions and explicitly include: Visual understanding [13, 14], Vision–language alignment [15, 16], Long-context modeling [17, 18], Chain-of-Thought (CoT) reasoning [19, 20], Scientific knowledge reasoning [17, 21] and so on [22, 23, 24]. In Earth science, existing multimodal benchmarks often focus on visual question answering using remote sensing data, covering a variety of satellite observation modalities and resolutions [25, 26, 27]. However, these existing benchmarks mainly focus on the human-activities sphere, with few or no multimodal benchmarks for other spheres. Moreover, while the semantic information in the observation data of the human-activities sphere is well-defined (e.g., buildings, roads and ships), other Earth systems lack precise scientific information formulation. This presents a new challenge: **How to establish scientific information definitions across multi-sphere Earth observations for effectively evaluating multimodal models?**

To address this challenge, we introduce OmniEarth-Bench to evaluate the scientific information processing capabilities of multimodal models across six Earth science spheres and cross-sphere scenarios. Considering the professional expertise required for analyzing Earth observation data, we have established four tasks: basic perception tasks, general reasoning tasks, specialized scientific reasoning tasks, and specialized scientific CoT reasoning tasks. The basic perception tasks are designed to assess the model's ability to perceive and recognize fundamental features and patterns in the Earth observation data. The general reasoning tasks evaluate the ability to draw logical conclusions based on the perceived information. The specialized scientific reasoning tasks aim to assess the ability to interpret scientific knowledge related to observational data. The specialized scientific CoT reasoning tasks evaluate the ability to perform step-by-step analysis of the observation data and derive accurate conclusions based on scientific knowledge.

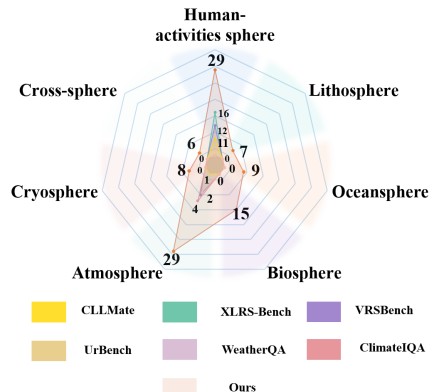

Figure 2: **Dimensions Categories of L4 dimensions.** Our benchmark spans 6 spheres and cross-sphere, across 100 typical subtasks (L4 dimensions).

Fig. 1 shows the typical examples across 6 spheres and cross-spheres. We engaged 2–5 experts (PhD holders or candidates) per sphere to identify representative real-world tasks, establish authoritative evaluation dimensions, and curate relevant observational datasets (either existing datasets or original data sourced from satellites like MODIS [28]). After defining these dimensions, we enlisted 40 crowd-sourcing annotators (undergraduate and master's students, 5–10 per sphere) to assist experts in annotation, followed by rigorous cross-validation to ensure quality. Ultimately, OmniEarth-Bench comprises 100 sub-dimensions (L-4 tasks) across seven categories (atmosphere, lithosphere, Oceansphere, cryosphere, biosphere, Human-activities sphere, and cross-sphere). As illustrated in Fig. 2, OmniEarth-Bench substantially surpasses existing benchmarks in comprehensiveness and coverage. Tab. 2 summarizes its quantitative and qualitative advantages. The key contributions are:

- **Comprehensive Evaluation Across All Six Spheres.** OmniEarth-Bench is the first benchmark to extensively cover all Earth science spheres, offering 58 practical and comprehensive evaluation dimensions that significantly surpass prior benchmarks.
- **Pioneering Cross-Sphere Evaluation Dimensions.** To address complex real-world scenarios, OmniEarth-Bench introduces cross-sphere evaluation capabilities for societally important tasks such as disaster prediction and ecological forecasting.
- **CoT-Based Reasoning Evaluations in Earth Science.** OmniEarth-Bench establishes, for the first time, CoT-based evaluations tailored for complex Earth science reasoning tasks, addressing scenarios where previous benchmarks showed near-zero accuracy, and explores how CoT strategies might enhance reasoning capabilities in the Earth domain.

## 2    Related Work

**Earth Multimodal Benchmark.** Recent advancements in large multimodal models (MLLMs) have accelerated progress in Earth sciences [29, 30], leading to the development of several evaluation benchmarks [25, 26, 31, 32]. Current benchmarks primarily target the Human-activities sphere and atmosphere. In the Human-activities sphere, remote sensing-based benchmarks include RSIEval [33], featuring 100 human-annotated captions and 936 VQA pairs; VRSBench [25], containing 29,614 images, 52,472 object references, and 123,221 QA pairs; and XLRS-Bench, which offers the largest dataset to date with an average resolution of 8500×8500. Atmospheric benchmarks include WeatherQA [31], designed specifically to evaluate severe weather predictions in two dimensions; ClimateIQA [32], built from climate reanalysis data for extreme weather event detection across four question types; and CLLMate [34], focused on weather and climate event forecasting using numerical meteorological data and textual event descriptions. However, these benchmarks exhibit notable limitations: 1) They typically address isolated spheres, neglecting cross-sphere interactions essential to real-world Earth science challenges. 2) They offer limited evaluation dimensions, with atmospheric benchmarks assessing fewer than four question types, and even the most extensive Human-activities sphere benchmark covering only 16 dimensions. Overall, comprehensive benchmarks addressing all six spheres and evaluating cross-sphere capabilities are still lacking in Earth sciences.

**General Multimodal Benchmark.** Large-scale vision-language models (VLMs) have shown great promise in multimodal tasks such as scene understanding and visual sentiment analysis, prompting the development of diverse benchmarks to quantitatively assess their capabilities. However, earlier benchmarks mostly targeted specific domains with limited evaluation tasks (*e.g.*, visual grounding [35, 36] or visual question answering (VQA) [37, 38, 39, 40, 41]). Recent efforts aim for more comprehensive assessments: MME [15] evaluates 14 perceptual and cognitive tasks; MMBench [13] offers over 3,000 questions spanning 20 skill dimensions like object localization and social reasoning; Seed-Bench [16] scales up further with 19,000 questions; MMT-Bench [24] integrates real-world scenarios like autonomous driving; and MME-Realworld [18] includes five real-world contexts with high-resolution imagery. Multimodal benchmarks focusing on scientific disciplines have also emerged. HLE [42] covers numerous academic disciplines with 2,500 questions; MMMU-Pro [43] evaluates multidisciplinary visual-textual integration skills at scale. Recently, multimodal chain-of-thought (CoT) benchmarks were developed: MME-CoT [19] includes 1,130 questions annotated with 3,865 reasoning steps; and ZeroBench [20] provides 100 handpicked questions and 334 simpler sub-questions. Despite these advancements, two critical limitations remain: 1) Earth sciences have been largely neglected, with only SuperGPQA featuring a minimal number (100) of geophysics-related textual questions, and multimodal CoT benchmarks lacking Earth science content entirely. 2) Existing benchmarks overlook the importance of observational data, a distinctive strength of Earth sciences (e.g., satellite imagery, climate data grids, seismic signals). In summary, current general-domain benchmarks fail to sufficiently evaluate multimodal models in Earth sciences, particularly concerning observational data and CoT reasoning scenarios.

## 3    OmniEarth-Bench

OmniEarth-Bench stands out from existing multimodal understanding benchmarks with three key features: i) It is the first benchmark based on Earth observational data to comprehensively cover all six Earth spheres, with evaluation dimensions grounded in real-world needs and rigorously validated by domain experts. ii) It firstly introduces the cross-sphere evaluation dimensions in geoscience, enabling

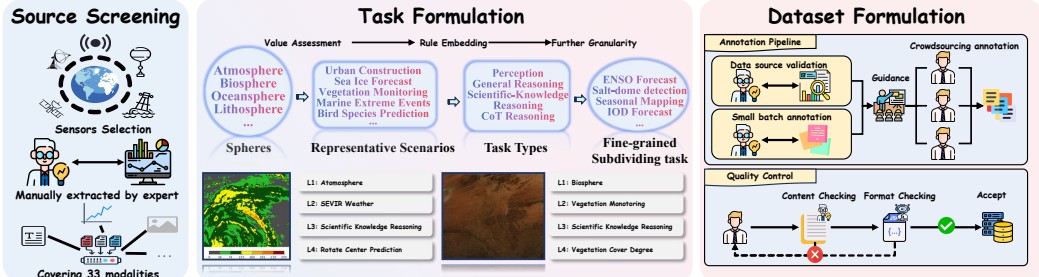

Figure 3: **Pipeline of OmniEarth-Bench.** Our pipeline comprises 4 stages—Source Screening, Task Formulation, Expert Annotation, and Quality Control—all led by experts. The first two stages are exclusively conducted by experts, while crowdsourcing annotators assist in the latter two stages.

MLLMs to be tested on realistic, interdisciplinary Earth science cross-sphere tasks. iii) It firstly establishes the Chain-of-Thought (CoT) reasoning benchmark for geoscience, using expert-reviewed human annotations and cross-validation to assess CoT effectiveness in complex scientific reasoning.

## 3.1 Pipeline of Benchmark

**Source Screening.** Our Benchmark comprises not only publicly available open-source datasets but also a significant portion of data manually extracted by experts from satellite imagery and raw observational sources. For example, Vegetation Monitoring uses satellite imagery from MODIS and expert-curated data from the Global Land Surface Satellite (GLASS), including Leaf Area Index, Fractional Vegetation Cover and Peak Vegetation Coverage Area. Moreover,

Table 1: **Data source of different spheres**, including open-source datasets, satellite websites and other observation data sources. We only exhibit the L1 and L2 dimensions.

| L1 dimensons | L2 deminesons | Data Source | Annotations Volume |
|---|---|---|---|
| Cross-sphere | Global Flood Forecasting | GFF [3] | 873 |
| | Bird Species Prediction | SatBird [44] | 2,253 |
| | Carbon Flux Monitoring | CarbonSense [45] | 330 |
| Human-activities sphere | Urban Construction | UBCv1 [46], BHdataset [47] | 3,161 |
| | Land Use | WHU-OHS [48] | 2,990 |
| | Surface Disaster Assessment | XView [49] | 3,851 |
| Biosphere | Species Distribution Prediction | TreeSatAI [50], Penguin [51] OAM-TCD [52], TaxaBench [53] | 2,819 |
| | Vegetation Monitoring | GLASS [54], MODIS [28] | 900 |
| | Environmental Pollution Monitoring | ROSID [55] | 246 |
| | Human Footprint Assessment | HFP [56], MODIS [28] | 600 |
| | Crop Growth Monitoring | MOPAD [57] | 1,656 |
| Atmosphere | SEVIR Weather | SEVIR [58] | 893 |
| | Typhoon Events | DigitalTyphoon [59] | 5,082 |
| | Short-term meteorological events | ERA5 [60] | 140 |
| | Medium-term meteorological events | ERA5 [60] | 160 |
| | Seasonal meteorological events | ERA5 [60] | 60 |
| | Interannual climate change | ERA5 [60] | 60 |
| Lithosphere | Earthquake monitoring and prediction | STRAD [61] | 1,500 |
| | Geological exploration imaging | TGS-Salt [62] | 631 |
| Oceansphere | Marine Debris and Oil Pollution | MADOS [63] | 221 |
| | Marine Extreme Events | ERASSTv5 [64] | 583 |
| | Marine Phenomenon Detection | COMS [65], M4Fog [66] | 570 |
| Cryosphere | Sea ice forecast | G02202 (SIC) [67], NSIDC-0079 [68] PIOMAS [69],GIOMAS [70] | 200 |
| | Glacier analysis | CryoSat-2 [71] IceBridge [72], ICESat-2 [73] | 30 |

for the Eddy data in oceansphere, the chlorophyll (CHL) data used in this study were obtained by applying the OCI empirical algorithm to Level-2 data acquired by the Geostationary Ocean Color Imager I (GOCI) aboard the Oceanography and Meteorology Satellite (COMS). After careful selection and integration, we compiled a comprehensive dataset covering 33 different data modalities across all Earth spheres. Tab.1 is a summary of the data sources used for each Earth sphere, with detailed data organization and construction procedures presented in the appendix.

**Task Formulation**. As shown in Fig.3, OmniEarth-Bench defines tasks across four hierarchical levels (L1–L4): L1 covers the seven domains based on established geophysical spheres: atmosphere, lithosphere, oceansphere, cryosphere, biosphere, Human-activities sphere and cross-sphere. L2 includes expert-approved, representative scenarios within each sphere, selected based on their scientific and practical value (e.g., earthquake prediction). Tab.1 illustrates representative scenarios covered by the L1 and L2 levels. Detailed descriptions of the L3 and L4 dimensions for each sphere are provided in the appendix. L3 comprises four core abilities: Perception, General Reasoning, Scientific-Knowledge Reasoning and CoT Reasoning. Perception and General Reasoning align with previous works such as MMBench [13] and XLRS-Bench [26], where Perception focuses on sensory inputs and Reasoning

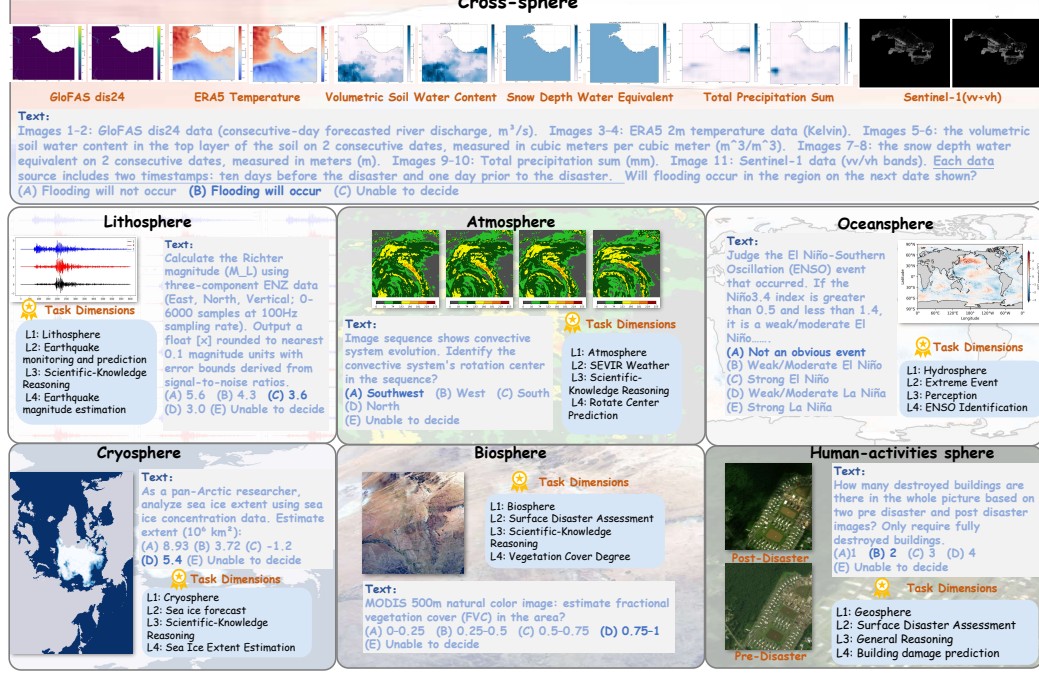

Figure 4: **Examples of OmniEarth-Bench.** OmniEarth-Bench comprises 100 unique L4 tasks, each with distinct questions, answers, and images. Spanning diverse data sources, timeframes, and natural variables, all tasks are jointly defined by domain experts across spheres.

on inference. Scientific-Knowledge Reasoning addresses complex reasoning tasks requiring deep domain expertise in Earth sciences. CoT Reasoning evaluates the effectiveness of chain-of-thought processes within Earth science scenarios. L4 provides further granularity by subdividing tasks based on the L1–L3 dimensions. Each L4 category is verified by domain experts to ensure practical relevance. Examples include fractional vegetation cover estimation in the biosphere and earthquake magnitude estimation in the lithosphere. Achieving robust general intelligence in Earth sciences requires MLLMs to perform effectively across all hierarchical levels. OmniEarth-Bench provides the first comprehensive framework designed for such an evaluation.

**Expert Annotations.** For each of the six Earth spheres, we enlisted 2–5 domain experts (Ph.D. holders or candidates) and 5–10 crowd-sourcing annotators (undergraduate and master's students). (1) For each sphere, evaluation dimensions were collaboratively defined by domain experts and MLLM specialists, ensuring high practical value and complexity. Cross-sphere tasks involved experts from multiple domains. This approach addresses the limitations observed when crowd-sourcing annotators proposed overly simplistic tasks—for example, "Estimated Maximum Precipitation Level" in atmosphere, which GPT-4o solved with 97.7% accuracy. Expert-led design ensures meaningful evaluation. (2) Experts were also responsible for defining data sources. Attempts to delegate this to annotators led to issues such as low sample difficulty and data scarcity. For complex tasks, annotators struggled with downloading and aligning data (e.g., MODIS and GLASS from NASA). Thus, experts curated and organized datasets, with annotators assisting.

**Quality Control.** To ensure data integrity and task relevance, the quality control process involved two main steps. Cross-Validation: Annotator outputs were systematically compared against expert-provided annotation examples. Any discrepancies were flagged and reviewed by domain experts to ensure annotation correctness, especially for complex tasks involving multi-source or challenging data. Final Quality Assessment: MLLM specialists conducted thorough reviews to confirm that annotations adhered to expert standards and maintained consistency across all tasks and Earth spheres. High-quality annotations were approved and incorporated into the dataset, while annotations that did not meet quality standards underwent iterative refinement through a feedback loop involving annotators and expert supervision. This cyclical process ensured continuous improvement and maintained the overall reliability of the dataset.

Table 2: **Comparison between existing vision-language benchmarks and our benchmark.** ✗ represents semi-automated, *i.e.*, machine generation followed by human verification.

| Dataset | Spheres | Cross-Sphere | Observation Data | Data Source Volume | VQA and Visual Grounding | | | | CoT | |
| --- | --- | --- | --- | --- | --- | --- | --- | --- | --- | --- |
| | | | | | Volume | Dimensions Volume | Expert Annotation | | Volume | Average key step annotation |
| ScienceQA [17] | ✗ | ✗ | ✗ | - | 21,000 | 127 | | ✗ | ✗ | ✗ |
| Seed-Bench [16] | ✗ | ✗ | ✗ | - | 19,242 | 12 | | ✗ | ✗ | ✗ |
| MME [15] | ✗ | ✗ | ✗ | - | 2,374 | 14 | | ✓ | ✗ | ✗ |
| MMBench [13] | ✗ | ✗ | ✗ | - | 3,217 | 20 | | ✓ | ✗ | ✗ |
| MME-Realworld [18] | ✗ | ✗ | ✗ | - | 29,429 | 43 | | ✓ | ✗ | ✗ |
| ZeroBench [20] | ✗ | ✗ | ✗ | - | ✗ | ✗ | | ✗ | 100 | ✗ |
| MME-CoT [19] | ✗ | ✗ | ✗ | - | ✗ | ✗ | | ✗ | 1,130 | 3.2 |
| VRSBench [25] | Human-activities sphere | ✗ | ✓ | 2 | 175,703 | 12 | | ✗ | ✗ | ✗ |
| XLRS-Bench [26] | Human-activities sphere | ✗ | ✓ | 6 | 45,008 | 16 | | ✓ | ✗ | ✗ |
| RSIEval [33] | Human-activities sphere | ✗ | ✓ | 1 | 933 | 1 | | ✗ | ✗ | ✗ |
| UrBench [27] | Human-activities sphere | ✗ | ✓ | 6 | 11,600 | 11 | | ✗ | ✗ | ✗ |
| WeatherQA [31] | Atmosphere | ✗ | ✓ | 1 | 8,000 | 2 | | ✗ | ✗ | ✗ |
| ClimateIQA [34] | Atmosphere | ✗ | ✓ | 2 | 254,040 | 4 | | ✗ | ✗ | ✗ |
| CLLMate [34] | Atmosphere | ✗ | ✓ | 2 | 7,747 | 1 | | ✗ | ✗ | ✗ |
| OmniEarth-Bench | 6 Spheres | ✓ | ✓ | 33 | 29,779 | 100 | | ✓ | 610 | 5.8 |

## 3.2 Task Dimensions

OmniEarth-Bench defines tasks across four hierarchical levels (L1–L4), comprising 7 L1 dimensions, 23 L2 dimensions, 4 L3 dimensions, and 103 expert-defined L4 subtasks with real-world applicability. One representative L4 subtask from each L1 sphere is illustrated in Fig 4. Detailed descriptions of the L3 and L4 dimensions are provided in the appendix.

**Cross-sphere.** Cross-sphere tasks in Earth science carry high practical and societal importance [4, 5, 6, 9]. To evaluate MLLMs, we select three representative L2 scenarios from socially impactful applications, including *Global Flood Forecasting (L2), Bird Species Prediction (L2) and Carbon Flux Monitoring (L2)*. Due to their reliance on expert knowledge and complex reasoning, all are categorized as Scientific-Knowledge Reasoning (L3). Their L4 dimensions are collaboratively defined by experts from the relevant spheres. Despite the complexity of cross-sphere scenarios, we successfully collaborated with domain experts to construct **6 high-value subtasks (L4 dimensions) .**

**Lithosphere.** We firstly construct an MLLM benchmark for the lithosphere based on observational data, comprising **7 practical subtasks (L4 dimensions) .** We define two representative L2 scenarios within the lithosphere: *Seismic Monitoring and Prediction (L2) and Geophysical Exploration (L2)*. Seismic monitoring and prediction [74], a critical domain in geosciences, aims to uncover Earth's internal dynamics and earthquake nucleation mechanisms, forming a theoretical basis for early warning and disaster mitigation. Geophysical exploration imaging [75], by analyzing subsurface responses to physical fields such as seismic waves, electromagnetic fields, and gravity/magnetic anomalies, enables high-resolution geological modeling essential for understanding subsurface structures, hydrocarbon exploration, and geological hazard assessment.

**Human-activities sphere.** The Human-activities sphere leverages remote sensing and mapping technologies across three key scenarios: *Urban Construction (L2), Land Use (L2), and Surface Disaster Assessment (L2)*. Urban construction supports planning and socio-economic analysis; land use classification underpins environmental monitoring and resource management; and disaster assessment enables rapid post-event response and risk mitigation. OmniEarth-Bench spans all four L3 capability dimensions in the Human-activities sphere—Perception, General Reasoning, Scientific-Knowledge Reasoning, and CoT Reasoning—with **29 subtasks (L4 dimensions)**, surpassing all existing benchmarks in this domain [25, 26].

**Atmosphere.** The atmosphere is a key domain in Earth sciences with high practical value and extensive research interest [76, 77]. While existing benchmarks target specific atmospheric sub-scenarios [31, 32, 34], they lack comprehensive domain-wide coverage. OmniEarth-Bench addresses this gap by defining evaluation dimensions across six representative scenarios using data from ERA5 [60], SEVIR [58], and Typhoon [59] datasets: *Short-term Weather Events (L2), Medium-term Weather Events (L2), Seasonal Weather Events (L2), Interannual Climate Variation (L2), Typhoon Event (L2), and SEVIR Weather (L2)*. For example, the *Typhoon Event* dimension serves as a flagship benchmark for atmospheric machine learning, supporting operational hazard forecasting and advancing research on tropical cyclone intensity and structure. These six scenarios (L2) span **30 expert-designed subtasks (L4 dimensions)** with strong real-world relevance, substantially surpassing existing atmospheric benchmarks. Full task details are provided in the appendix.

**Oceansphere.** We build a multi-layer MLLM benchmark for the oceansphere based primarily on satellite and analysis data products, featuring **9 practical L4 subtasks**. This domain includes three representative L2 scenarios: *Marine Oil Spills and Debris Monitoring (L2), Extreme Oceanic*

*Events Warning (L2), and Ocean Phenomena Detection (L2)*. The Marine Oil Spills and Debris Monitoring [78] scenario uses multi-source remote sensing and in situ water quality data to track the spatial distribution and temporal dynamics of oil contamination and floating debris, supporting environmental management and emergency response. The Extreme Oceanic Events Warning [79, 80] scenario targets the detection and prediction of major climate modes such as El Niño–Southern Oscillation (ENSO) and the Indian Ocean Dipole (IOD), aiming to mitigate their societal and economic impacts. The Ocean Phenomena Detection scenario [81, 82] involves identifying ocean features like eddies and marine fog, which are key for maritime safety and ecological studies.

**Cryosphere.** We conduct a MLLM benchmark for the cryosphere primarily based on sea ice reanalysis data, glacial imagery, and graphical plots, incorporating **8 practical L4 subtasks**. We identify two representative L2 scenarios within the cryosphere: *Sea Ice Forecasting (L2) and Glacier Analysis (L2)*. SSea ice forecasting focuses on predicting the dynamic changes of sea ice in polar regions. Arctic sea ice is crucial for understanding global climate change [83, 84]. Its continuous decline over the last few decades has made sea ice forecasting significant for navigating through the Arctic Ocean during melting seasons. Moreover, the loss of the Antarctic sea ice would greatly impact the global sea level. Glacier analysis [85, 86], aims to study the glacial movements and changes of glaciers over time.

**Biosphere.** We present a biosphere-focused MLLM benchmark built on observational data and retrieval products, featuring **16 practical L4 subtasks**. It includes four representative L2 scenarios: Vegetation Monitoring (L2), Human Footprint Assessment (L2), Environmental Pollution Monitoring (L2), Species Distribution Prediction (L2) and Crop Growth Monitoring (L2). Vegetation Monitoring [87] evaluates plant and ecosystem health to support function assessment, carbon accounting, and climate response. Human Footprint Assessment [88] quantifies human impact on nature, informing sustainability and biodiversity strategies. Environmental Pollution Monitoring [89] identifies pollution events and their extent, guiding environmental policy and mitigation. Species Distribution is a key concern in the biosphere, as it guides biodiversity conservation and supports modeling species range shifts under climate and land-use change. Crop Growth Monitoring [90] assesses crop health for precision agriculture and sustainable farming.

## 3.3 Statistics and Analyses

**Overview Statistics.** OmniEarth-Bench includes 100 expert-defined, high-value evaluation dimensions and 29,779 samples annotated by both experts and crowdsourced contributors. As shown in Tab. 2, it offers clear advantages over existing benchmarks. Uniquely built on observational Earth science data—rather than exam-style datasets—OmniEarth-Bench spans all six spheres and cross-sphere scenarios. Consistency metrics are reported in Tab. 3, with additional details and dimension-specific indicators provided in the appendix.

**Observational Data vs. Exam-questions Data.** Unlike subject-based benchmarks like ScienceQA [17], which rely on exam questions or online learning problems followed by manual filtering, our approach takes a fundamentally different path. While such methods could theoretically span all six Earth spheres, they face two key limitations: (1) Benchmarks like ScienceQA focus on scientific inquiry rather than practical geoscience applications, limiting their real-world relevance. (2) Their evaluation dimensions are constrained by a bottom-up design—questions are derived from existing image-text pairs in question banks or papers, then filtered and categorized. In contrast, OmniEarth-Bench follows a

Table 3: **Main statistics in OmniEarth-Bench**

| Statistic | Number |
|---|---|
| Total questions | 29,779 |
| - Cross-sphere | 3,456 |
| - Human-activities sphere | 9,362 |
| - Biosphere | 6,221 |
| - Atmosphere | 6,395 |
| - Lithosphere | 2,131 |
| - Oceansphere | 1,374 |
| - Cryosphere | 230 |
| Multiple-choice questions | 27,082 |
| Visual grounding questions | 2,697 |
| Single-image questions | 24,108 |
| Multi-image questions | 5,671 |
| Maximum question length | 213 |
| Average question length | 48.2 |
| CoT | |
| - Total key step annotation | 3,473 |
| - Average key step annotation | 5.8 |
| - Average key step length | 14.8 |
| - Maximum question length | 101 |
| - Average question length | 50.2 |

top-down strategy: domain experts first define evaluation dimensions based on real-world geoscience challenges and data availability, then curate corresponding data. This ensures each task is both meaningful and grounded in practical utility.

Table 4: **Experimental results on each sphere of VQA tasks, with models ranked by average performance.** '*Avg*' represents the average accuracy across sub-tasks. Proprietary models are highlighted in gray. '*Experts*' means evaluation results of 100 examples in each sphere by experts. We mark the highest score of each metric in red , and second highest underlined.

| Method | Speres (L1 dimensions) | | | | | | | Avg. |
|---|---|---|---|---|---|---|---|---|
| | Cross-sphere | Atmosphere | Lithosphere | Oceansphere | Cryosphere | Biosphere | Human-activities sphere | |
| Experts | 90 | 96 | 91 | 95 | 93 | 97 | 95 | 93.4 |
| *Closed-source MLLMs* | | | | | | | | |
| Claude-3.7-Sonnet [12] | 30.68 | 24.72 | 28.15 | 23.12 | 54.46 | 31.21 | 11.18 | 29.07 |
| Gemini-2.0 [11] | 16.93 | 20.83 | 38.94 | 16.94 | 58.52 | 20.83 | 23.74 | 28.1 |
| GPT-4o [10] | 0.04 | 9.64 | 12.8 | 13.35 | 37.48 | 1.97 | 2.76 | 11.15 |
| *Open-source MLLMs* | | | | | | | | |
| InternVL3-72b [97] | 19.19 | 33.98 | 23.39 | 20.22 | 74.56 | 31.99 | 29.46 | 33.26 |
| InternVL3-7b [97] | 42.85 | 30.1 | 37.47 | 20.28 | 49.27 | 28.74 | 23.18 | 33.13 |
| LLaVA-Onevision-7b [92] | 19.26 | 33.69 | 28.72 | 24.54 | 46.4 | 37.31 | 30.62 | 31.51 |
| Internlm-Xcomposer2.5-7b [98] | 19.78 | 17.45 | 28.88 | 21.06 | 40.04 | 30.67 | 24.76 | 26.09 |
| Qwen2.5-VL-7B [99] | 9.85 | 9.25 | 18.65 | 13.95 | 17.85 | 10.94 | 6.23 | 12.39 |
| Qwen2.5-VL-72B [99] | 3.92 | 4.82 | 22.43 | 16.27 | 5.88 | 14.91 | 8.63 | 10.98 |

**Human Annotations vs. GPT Annotations.** All annotations are finished by experts and crowd-sourcing annotators. Unlike MMBench [13], we did not use tools like GPT-4o [10]. It was driven by two key reasons: (1) GPT-4o cannot generate VQA data requiring deep domain expertise. Tasks under the Scientific-Knowledge Reasoning (L3) demand substantial background knowledge and must be constructed collaboratively by experts. (2) Although GPT-4o can generate samples for general perception or simple reasoning tasks, expert evaluation found the data to be low quality and insufficiently challenging. For example, in visual grounding task, GPT-4o only detects highly salient structures, failing to support our goal of testing MLLMs on locating diverse buildings across complex scenes. As a result, all OmniEarth-Bench data was exclusively created by experts and annotators.

# 4 Experiment

**Experimental Setup.** The MLLMs evaluated on OmniEarth-Bench are grouped into two categories: (a) open-source VLMs, including Qwen2.5-VL [91], LLava-Onevision [92], InterVL3 [93] and InternLM-XComposer-2.5 [94]; (b) closed-source VLMs, such as GPT-4o [10], Gemini-2.0 [11] and Claude 3.7 Sonnet [12] All models were evaluated using LMMs-Eval [95, 96]. Following MMBench [13] and MME-Realworld [18] methods, in the VQA task, we manually created 5 options for each question: one correct answer, three distractors and one special answer (unable to answer). We evaluated the accuracy and reported of L-1 dimension for the VQA task, with L-3 and L-4 results available in the appendix. All scores in Tables 4 are reported as percentages (%). For the Grounding task, we used precision, assessing accuracy based on the intersection between predicted and ground truth bounding boxes, with predictions deemed correct if IoU exceeds a threshold (0.5 and 0.7).

**All MLLMs exhibit suboptimal performance across all 7 domains.** As illustrated in Tab. 4 and Tab. 5, nearly all MLLMs achieve accuracy rates below 40%, significantly underperforming relative to their success on traditional perception or reasoning benchmarks [13, 100, 41]. Several factors likely contribute to this challenge. First, current multimodal large models are typically trained without domain-specific Earth science data, which impedes their ability to comprehend related queries. Second, many Earth science problems are inherently complex, particularly cross-domain prediction tasks that demand in-depth,

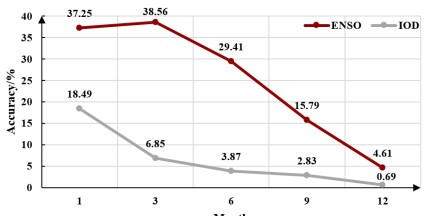

Figure 5: **GPT-4o performance on ENSO and IOD prediction with different lead months (previous).**

specialized knowledge, which existing LLMs or MLLMs may not possess. Finally, OmniEarth-Bench provides high-resolution, intricate imagery, and the task of interpreting such complex visuals presents unique obstacles for MLLMs. This underscores the pressing need for specialized models or advanced post-training techniques to effectively address these challenges.

**Time-sensitive task.** The Earth's seven spheres encompass numerous temporally correlated tasks. ENSO, a key climate mode influencing global weather extremes via teleconnections [101], has seen improved forecasts through domain-specific AI models [79, 102]. As shown in Fig. 5, prediction accuracy declines with longer lead times, echoing the limitations of specialized models. However, performance of MLLMs still lags behind tailored models. Performance drops further for Indian Ocean Dipole (IOD) predictions, aligning with challenges faced by existing methods [103, 80].

Table 5: **Visual grounding performance on OmniEarth-Bench.** 3 spheres have the groudning task.

| Benchmark | Method | GPT-4o | Gemini-2.0 | Claude-3.7-Sonnet | Qwen2.5-VL | LLaVA-OneVision | InternVL3 8B | InternVL3 78B |
|---|---|---|---|---|---|---|---|---|
| Human-activities sphere | Acc@0.5 | 0.02 | 0.03 | 0 | 0.59 | 0.2 | 0 | 2.36 |
| | Acc@0.7 | 0 | 0 | 0 | 0 | 0 | 0 | 0.2 |
| Lithosphere | Acc@0.5 | 0.08 | 0.13 | 0.02 | 5.3 | 0 | 8.94 | 4.3 |
| | Acc@0.7 | 0 | 0.04 | 0 | 0.33 | 0 | 1.66 | 0.33 |
| Oceansphere | Acc@0.5 | 0.12 | 0.34 | 0.2 | 1.81 | 1.51 | 6.63 | 13.86 |
| | Acc@0.7 | 0.01 | 0.06 | 0.07 | 0 | 0 | 0.6 | 3.61 |

**Limited Gains from Scaling Model Size on Earth-Science Tasks.** We evaluate two sizes of the InterVL3 model and find that the 72B InterVL3 does not provide a significant advantage over the 7B model in our benchmarks, with performance even declining in some evaluation metrics. This contrasts with the substantial improvements observed in general-domain tasks. This performance bottleneck likely stems from the lack of Earth-science-specific knowledge, rather than a limitation in model capacity. Even large MLLMs struggle to reason about unfamiliar scientific concepts without targeted training on domain-specific data. These findings highlight the importance of prioritizing the integration of domain-specific knowledge in future Earth-science MLLM development, rather than merely increasing model size.

**Impact of Model Safety on Results.** In Tab. 4, we observe that Qwen2.5-VL and GPT-4o perform very poorly, even falling below the level of random guessing. However, this does not mean that these two models have the worst perceptual and science-related abilities. We observe that these models tend to refuse to answer when they are uncertain, whereas InternVL3 and LLaVA-Onevision randomly guess an answer. This safety mechanism in the models leads to the poor performance of Qwen2.5-VL and GPT-4o. Detailed refusal rate statistics for each model are provided in the appendix—for instance, QwenVL2.5-VL-72B refused to answer 18,258 questions.

**CoT performance.** Following the MME-CoT [19], building upon our annotated key steps in Earth Observation data, we leverage two interpretable metrics to evaluate the CoT correctness: recall and precision. The two metrics respec-

Table 6: **CoT performance of OmniEarth-Bench**

| Models | LLaVA-OneVision-7B | Qwen2.5-VL-7B | InternVL3-8B | InternVL3-78B |
|---|---|---|---|---|
| Precision | 89.83 | 92.72 | 94.02 | 94.74 |
| Recall | 23.41 | 29.12 | 34.47 | 35.5 |
| F1 | 37.14 | 44.32 | 50.45 | 51.65 |

tively attend to the two aspects of the CoT correctness: informativeness and accuracy. Finally, we calculate the F1 score as the metric of CoT quality. As shown in Tab.6, InternVL3 outperformed Qwen2.5-VL and LaVA-OneVision with the highest F1 score. Larger open-source variants showed superior performance, underscoring the scalability of model size.

**Expert Validation.** Although OmniEarth-Bench's evaluation dimensions and data sources were curated by experts, we further validated its quality through expert upper-bound assessments. We randomly sampled 100 questions from each sphere and invited independent experts—unaffiliated with the annotation team—to answer them. As shown in Tab. 4, expert accuracy consistently exceeded 90%, confirming the benchmark's reliability. Occasional errors arose mainly in tasks requiring precise calculations or counting.

## 5 Conclusion

We have introduced OmniEarth-Bench, a comprehensive multimodal Earth science benchmark that encompasses all six spheres of the Earth system (atmosphere, lithosphere, Oceansphere, cryosphere, biosphere, and human-activities) along with their cross-sphere interactions. This benchmark introduces 100 expert-curated evaluation dimensions and four hierarchical levels of reasoning (perception, general reasoning, expert-knowledge reasoning, and chain-of-thought reasoning), representing a novel and rigorous evaluation design for geoscientific MLLMs. Our results show that even state-of-the-art MLLMs (e.g., Claude) struggle with OmniEarth-Bench; none of the tested models surpassed 35% accuracy. This stark performance gap underscores the benchmark's difficulty and exposes fundamental limitations in current models' geoscientific understanding. The significance of OmniEarth-Bench lies in its breadth and depth, providing a unified challenge that pushes beyond existing capabilities and highlighting the need for MLLMs that integrate visual perception with expert domain knowledge and advanced reasoning. We anticipate that OmniEarth-Bench will serve as a catalyst for future research in geoscientific AI, guiding the development of models capable of expert-level analysis across Earth's spheres and enabling advanced applications in environmental monitoring, climate science, and Earth system management. **Limitations.** Due to the high cost and difficulty of data acquisition, some spheres currently include only eight evaluation dimensions. We plan to expand these by partnering with relevant institutions and companies to obtain more data.

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
