# OmniEarth-Bench: Towards Holistic Evaluation of Earth's Six Spheres and Cross-Spheres Interactions with Multimodal Observational Earth Data

**Fengxiang Wang[1,2], Mingshuo Chen[3], Xuming He[4], Yi-Fan Zhang, Feng Liu[2], Zijie Guo[2],**
**Zhenghao Hu[5], Jiong Wang[2] Jingyi Xu[2], Zhangrui Li[2], Fenghua Ling[2], Ben Fei[2], Weijia Li[5],**
**Long Lan,[1] Wenjing Yang[1]\*, Wenlong Zhang[2]\*, Lei Bai[2]**
[1] NUDT [2]Shanghai AI Lab [3] BUPT [4] ZJU [5] SYSU [6] FDU

## 1 Appendix

### 1.1 Overview of the Appendix

This appendix supplements the proposed **OmniEarth-Bench** with details excluded from the main paper due to space constraints.

The appendix is organized as follows:

- Sec. 1.2: More details of OmniEarth-Bench.
- Sec. 1.3: Detailed results of specific sub-tasks (L-4 dimension).
- Sec. 1.4: Detailed results of specific sub-tasks (L-3 dimension).
- Sec. 1.5: Visualizations of samples and challenging cases.
- Sec. 1.6: Datasheets for the OmniEarth-Bench dataset.
- Sec. 1.7: Discussion on limitations and societal impact.

### 1.2 More Details of OmniEarth-Bench

We provide additional details about the dataset, with Table 1 and Table 2 presenting statistics for each L4 dimension, along with their relationships to the L3 and L2 dimension. This clarifies the dataset's structure and composition.

#### 1.2.1 Cross-sphere

- L2-Global Flood Forecasting:
    - **Flood Detecting**: Predicts whether a flood event will occur in the near future based on ground and atmospheric variables, including river discharge, 2-meter air temperature, top-layer volumetric soil water content, snow depth water equivalent, total precipitation, along with Sentinel VV / VH data from the preceding two days.
    - **Flood Predicting**: Predicts whether a flood event will occur in the near future based on the same variables used in Flood Detecting, along with Sentinel VV / VH data from the preceding two days.
- L2-Carbon flux monitoring:
    - **Carbon flux estimation:**Refers to the process of quantifying the rate and direction of carbon exchange (e.g. carbon dioxide) between the biosphere (vegetation and

---

*Corresponding authors

microorganisms, etc.) and the atmosphere, which tests the LLM's ability to interpret biogeochemical cycles, integrate multi-dimensional environmental data (e.g., satellite, sensor networks), and apply physics-based or statistical models for climate change analysis.

- L2-Bird species prediction:
  - **Most likely species to occur:** Predicting the species with the highest likelihood of presence in a specific habitat based on environmental variables (e.g., climate, habitat type), testing the LLM's ability to analyze spatial-environmental correlations and prioritize species under data-driven constraints.
  - **Species occurrence probability estimation:**Quantifying the probability of a species being present in a given geographic area, evaluating the LLM's grasp of probabilistic reasoning and ecological variable weighting.
  - **Species richness estimation:**Calculating the total number of distinct species within a defined ecosystem or region, testing the LLM's capacity to integrate multi-modal data to predict biodiversity.

### 1.2.2 Human-activities sphere

- L2-Surface Disaster Assessment:
  - **Change detection counting of post-disaster completely destroyed building:** Compares pre- and post-disaster images to count fully destroyed buildings, evaluating temporal change detection.
  - **Counting of post-disaster partially damaged building:** Detects and counts lightly or moderately damaged structures in post-disaster imagery.
  - **Building damage prediction:** Estimates potential damage severity from pre-disaster views, testing risk assessment without ground truth.
  - **Disaster prediction:** Predicts future disaster types using current imagery, evaluating temporal modeling capabilities.
  - **Disaster type classification:** Identifies disaster types (e.g., flood, earthquake) from satellite images, testing visual pattern recognition.
  - **Geolocation estimation of disaster-affected regions from imagery:** Predicts the geographic location of affected areas based on visual cues, assessing spatial referencing.
  - **Individual building damage assessment:** Compares pre- and post-event imagery to evaluate building-level structural changes.
  - **Multi-image individual visual localization task:** Uses multi-temporal or multi-view images to locate specific buildings, assessing multi-view reasoning.
  - **Spatial relationships under complex conditions:** Infers spatial relations (e.g., relative position, containment) between objects in imagery, testing 3D reasoning.
  - **Visual grounding of damaged individual buildings:** Locates damaged structures in post-disaster imagery, evaluating anomaly detection and spatial precision.
- L2-Urban Development:
  - **Fine-grained object type recognition:** Classifies specified buildings in high-resolution imagery, testing the model's ability to distinguish visually similar structures.
  - **Overall counting:** Counts all buildings or urban facilities in an image, evaluating object detection and counting under complex conditions.
  - **Counting under complex conditions:** Counts objects that meet given conditions (e.g., attributes or constraints), testing constrained multimodal reasoning.
  - **Overall building height estimation:**Estimates structural vertical dimensions using multi-source geospatial inputs, assessing 3D reconstruction accuracy and cross-sensor measurement consistency.
  - **Individual building height estimation:** Estimates the height of an individual building from satellite views, testing 2D-to-3D inference.
- L2-Land Use:
  - **Overall land type classification:** Identifies macro land cover types (e.g., urban, farmland, water), evaluating scene-level understanding.

- **Fine-grained land type classification:** Classifies specific land use (e.g., crop types) at finer scales, testing detailed semantic discrimination.
  - **Visual localization of land use types:** Locates specific land types within an image, evaluating spatial perception.
  - **Counting of land types under complex conditions:** Counts land use regions meeting complex conditions, assessing constrained visual reasoning.
  - **Visual groudning of land types:** Locates specific land types to evaluate the model's visual localization capability and land type classification ability.

### 1.2.3 Biosphere

- L2-Crop growth monitoring:

  - **Dead oil palm identification:** Identifies dead trees in unmanned aerial vehicle (UAV) imagery, testing the model's domain knowledge in crop growth.
  - **Dead oil palm counting:** Counting the number of dead trees in an image, testing the model's object counting capability.

- L2-Environmental pollution monitoring:

  - **Terrestrial oil spill counting:** Counting oil spill points in satellite imagery, testing the model's ability in environmental pollution recognition and object counting.
  - **Terrestrial oil spill area calculation:** Calculating the total area of oil spills in the image, evaluating the model's applicability in pollution events.

- L2-Human footprint assessment:

  - **Human footprint assessment:** Assessing the impact of human activities in the region based on imagery, testing the model's ability to recognize and reason about human activity features
  - **Human footprint index estimation:** Calculating the human footprint index of a region, testing the model's understanding of human activity patterns.

- L2-Species Distribution Prediction:

  - **Tree species prediction:** Identifying the type of tree that occupies the largest proportion, testing the model's ability to recognize features of different tree species.
  - **Tree species proportion prediction:** Identifying the proportion of specific tree species, testing the model's ability in species recognition and statistical reasoning.
  - **Animal classification:** Identifying animal species within the bounding box, testing the model's ability to extract local information and distinguish between different animal species.
  - **Geographical location inference of plant species:** Inferring the geographic coordinates from the image and the given tree species, testing the model's domain knowledge of tree species distribution.
  - **Global animal counting:** Counting the number of animals in the image, testing the model's ability in animal instance extraction and counting.
  - **Species distribution prediction:** Predicting the likely animal species in a region based on the image and geographic coordinates, testing the model's ability to extract ecological features and its knowledge of species distribution.

- L2-Vegetation monitoring:

  - **Fractional vegetation cover estimation:** Calculating the fractional vegetation cover in the image, testing the model's ability to recognize vegetation features.
  - **Leaf area index estimation:** Calculating the leaf area index from multi-band imagery, testing the model's ability to comprehensively understand and utilize multi-source information.
  - **Peak vegetation coverage area grounding:** Locating peak vegetation coverage areas in the image using multi-band imagery, testing the model's ability to localize vegetation features.

### 1.2.4 Atmosphere

- L2-Short-term weather events:

  - **Event intensity identification:**Determine extreme intensity or variable value at given position or region.
  - **Event localization:**Localize event center or moving direction of event.
  - **Event trend analysis:**Determine varying trend or speed of variable.
  - **Event type identification:**Determine type of current event.
  - **Dynamic feature identification:**Determine dynamic structure via multi-variable analysis.
  - **Event evolution analysis:**Determine stage of event via multi-variable analysis.
  - **Thermodynamic feature identification:**Determine thermodynamic features or structure via multi-variable analysis.

- L2-Medium-term weather events:

  - **Cyclone movement identification:** Determine moving direction of cyclone.
  - **Cyclone phase identification:** Determine the different phase of cyclone
  - **Event intensity identification:** Determine extreme intensity or variable value at given position or region.
  - **Event localization:** Localize event center or moving direction of event.
  - **Event trend analysis:**Determine varying speed or trend of current event.
  - **Geopotential pattern identification:** Determine pattern / structure of given geopotential.
  - **Moisture flux analysis:** Determine intensity of moisture flux transformation.
  - **System identification:** Determine dynamic structure via multi-variable analysis.
  - **System evolution trend analysis:** Determine evolution stage of current system via multi-variable analysis.

- L2-Typhoon:

  - **Pressure estimation:**Using the same image stacks, the task outputs the minimum sea-level pressure (hPa) at the cyclone eye; this complements wind speed and enables pressure–wind relationship validation.
  - **Radius of major gale axis estimation:**Using scatterometer-derived peak-gust layers, the model regresses the semi-major radius (km) of 50-kt gusts, characterising the reach of damaging winds for early warning.
  - **Radius of major storm axis estimation:**From the segmented wind-field map, the model estimates the semi-major radius (km) of 34-kt gale-force winds, quantifying the storm's main spatial extent and directly supporting surge-risk assessment.
  - **Radius of minor gale axis estimation:**Outputs the corresponding semi-minor radius, enabling a complete 2-D description of the gust envelope.
  - **Radius of minor storm axis estimation:**Analogous to the above, but for the semi-minor radius, capturing asymmetric size features critical to track-shift sensitivity analysis.
  - **Wind estimation:**Given time-synchronised multispectral satellite images, models must regress the storm-centre 1-min sustained surface wind speed in kt, providing a physics-consistent proxy for Saffir–Simpson intensity classification.

- L2-Seasonal weather events:

  - **Precipitation anomaly identification:**Determine precipitation anomaly value at given timestamp or region.
  - **Seasonal comparison:**Analysis of temperature/precipitation anomaly within a year.
  - **Temperature anomaly identification:**Determine temperature anomaly value at given timestamp or region.

- L2-Interannual climate variation:

  - **ENSO feature analysis:**Determine status or features of ENSO.

- **Long-term Precipitation trend analysis:** Determine trend of precipitation anomaly among years.
  - **Long-term Temperature trend identification:** Determine trend of temperature anomaly among years.
- L2-SEVIR Weather:
  - **Event type prediction:** Identifies storm event types based on visible and infrared channels from satellite, along with Vertical Integrated Liquid (VIL) data from wether radar.
  - **Miss alarm estimation:** Estimates the miss rate by comparing forecasted outputs against SEVIR ground truth.
  - **Movement prediction:** Given a sequence of VIL data, MLLMs are required to identify the move direction of convective system.
  - **Rotate center prediction**

### 1.2.5 Lithosphere

- L2-Earthquake monitoring and prediction:
  - **P-wave phase picking:** Taking three-component observed seismic waveforms as input, output the arrival times of the P-wave characteristic seismic signals.
  - **S-wave phase picking:** Taking three-component observed seismic waveforms as input, output the arrival times of the S-wave characteristic seismic signals.
  - **Earthquake or noise classification:** Distinguishing seismic signals from natural earthquakes versus artificial noise or vibrations, testing the LLM's understanding of geophysical signal patterns, noise discrimination, and time-series data analysis.
  - **Earthquake magnitude estimation:** Determine the earthquake magnitude based on the seismic amplitude at the location of the S-wave seismic phase.
  - **Earthquake source-receiver distance inference:** Single-station earthquake location is simplified to determining the distance from the earthquake hypocenter to the geophone through the distance between the P-wave and S-wave seismic phases.
- L2-Geophysics imaging:
  - **Salt body detection:** Identifying subsurface salt dome structures in geological or seismic data, testing the LLM's domain knowledge in geophysics, spatial pattern recognition, and geological feature interpretation.
  - **salt body location:** Precisely determining the spatial coordinates or depth of salt bodies within geological formations, evaluating the LLM's capability in spatial reasoning, multi-dimensional data integration, and quantitative analysis accuracy.

### 1.2.6 Oceansphere

- L2-Extreme Events:
  - **Enso identification:** Critical for mitigating global climate extremes, this task classifies ENSO events (e.g., El Niño and La Niña) by analyzing the Pacific SST anomaly maps.
  - **Iod identification:** Essential for monsoon forecasting and reducing compound risks in Indian Ocean nations, this task identifies Indian Ocean Dipole phases (positive/negative) from SST anomalies, similar to ENSO identification.
  - **Enso forecast:** As a complement to ENSO identification, this task predicts whether or what type of ENSO event will occur several months ahead using global SST anomaly maps of the past six months, which requires the model to capture the temporal evolution process.
  - **Iod forecast:** As a complement to IOD identification, this task predicts IOD event occurrence and type, similar to ENSO forecasting.
- L2-Phenomenon Detection:
  - **Eddy identification:** Fundamental for marine ecosystem management, this task identifies eddy types (cyclonic/anticyclonic) from the chlorophyll grayscale satellite image.

- **Marine fog detection:** Critical for maritime safety and intelligent navigation, this task identifies fog presence via satellite imagery.
- **Eddy Localizationtion:** As a complement to eddy localization, this task detects the location of eddies, enhancing search-and-rescue operations and pollution mitigation.

- L2-Marine Debris and Oil Pollution:

  - **Marine Pollution Type Classification:** Marine Pollution Type Classification refers to the scientific method of systematically categorizing marine pollutants based on their sources or characteristics (e.g., oil spills, plastic waste, chemical discharges)," which can test an LLM's domain knowledge in environmental science, multi-category semantic comprehension, and fine-grained classification capabilities.

### 1.2.7 Cryosphere

- L2-Glacier analysis:

  - **Glacial Lake Recognition:** A melting glacier could results in multiple glacial lakes. Identifying them could provide valuable information for analyzing the variation trend of glaciers. We provide the model with images of glacial lakes, glaciers, and regular lakes. The model is asked to output the quantity of glacial lakes. This L4 task not only assess the model's ability to identify glacial lakes from the others, it also assess whether the model is capable of accurately reasoning about the overall quantities.

  - **Glacier Melting Estimation:** To evaluate the model's ability to analyze glacier data, we first present the model with the observation of galcier melting rate data and a sample chart showing the correlation between the melting rate and displayed color. Then, we provide two predictive charts from different models and the model is required to identify which one better matches the provided real-world observation. Additionally, we show the model images of different glaciers at various times to see if it can determine which glacier is more likely to be in a melting state.

  - **Slide Recognition:** This task is designed to assess the model's ability to determine glacier landslide risks. First, we show it images of different glaciers and ask it to judge which one is more likely to experience a landslide based on their melting conditions. Then, we provide traverse and longitudinal melting profiles showing the melting rates and thickness of glaciers at different locations in Greenland, and ask the model to determine which glacier is more prone to landslides.

- L2-Sea ice forecast:

  - **SIC Estimate SIT:** To further test the model's ability to analyze sea ice concentration data, we provide it with a sea ice thickness variation trend chart, and sea ice concentration data from a date following the last point on that chart. Then model is instructed to forecast the sea ice thickness of the following day based on inputs.

  - **SIC Estimate SIV:** To explore the model's ability to analyze sea ice concentration data, we provide it with a sea ice volume variation trend chart, and sea ice concentration data from a date following the last point on that chart. We then assess whether the model can accurately forecast the sea ice volume for the following day based on those two inputs.

  - **SIT Trend Prediction:** In this L4 we further evaluate the model's ability to directly analyze the trend data and make reasonable forecasts. Similarly, we provide the model with the previous year's sea ice thickness variation curve and the trend up to a certain point in the following year. Then, the model is required to predict subsequent sea ice thickness according to input data.

  - **SIV Trend Prediction:** In this L4 task, we provide the model with the previous year's sea ice volume variation curve and the trend up to a certain point in the following year, to test the model's ability to make short-term forecasts of sea ice volume based on given data.

  - **Sea Ice Extent Estimation:** Questions in this L4 task are designed to assess the model's ability to distinguish between Antarctic and Arctic sea ice, determine the melting season, i.e. evaluate the sea ice extent changes over time, and estimate the sea ice extent area from given a sea ice concentration map.

Table 1: **Characteristics of each task (L4 dimension) in Human-activities sphere, Biosphere, Cross-sphere and Biosphere.** Human-activities sphere has 26 subtasks, Biosphere has 15 subtasks, Cross-sphere has 6 subtasks, Lithosphere has 7 subtasks.

| L1 | L2 | L3 | L4 Task Description | Format | Samples | Answer Type |
|---|---|---|---|---|---|---|
| Human-activities sphere | Surface Disaster | Perception | Change detection counting of post-disaster completely destroyedbuilding | VQA | 502 | Multiple Choice(A/B/C/D/E) |
| | | | Counting of post-disaster partially damaged building | VQA | 498 | Multiple Choice(A/B/C/D/E) |
| | | General Reasoning | Building damage prediction | VQA | 499 | Multiple Choice(A/B/C/D/E) |
| | | | Disaster prediction | VQA | 500 | Multiple Choice(A/B/C/D/E) |
| | | | Dissaster type classification | VQA | 500 | Multiple Choice(A/B/C/D/E) |
| | | | Geolocation estimation of disaster-affected regions from imagery | VQA | 502 | Multiple Choice(A/B/C/D/E) |
| | | | Individual building damage assessment | VQA | 107 | Multiple Choice(A/B/C/D/E) |
| | | | Multi-image individual visual localization task | VQA | 102 | Multiple Choice(A/B/C/D/E) |
| | | | Spatial relationships under complex conditions | VQA | 99 | Multiple Choice(A/B/C/D/E) |
| | | | Visual grounding of damaged individual buildings | VQA | 102 | Multiple Choice(A/B/C/D/E) |
| | | CoT | Individual building damage assessment | VQA | 107 | Multiple Choice(A/B/C/D/E) |
| | | | Multi-image individual visual localization task | VQA | 102 | Multiple Choice(A/B/C/D/E) |
| | | | Spatial relationships under complex conditions | VQA | 99 | Multiple Choice(A/B/C/D/E) |
| | | | Visual grounding of damaged individual buildings | VQA | 102 | Multiple Choice(A/B/C/D/E) |
| | Urban Development | Perception | Fine-grained object recognition | VQA | 514 | Multiple Choice(A/B/C/D/E) |
| | | | Overall counting | VQA | 502 | Multiple Choice(A/B/C/D/E) |
| | | General Reasoning | Counting under complex conditions | VQA | 754 | Multiple Choice(A/B/C/D/E) |
| | | | Individual building height estimation | VQA | 101 | Multiple Choice(A/B/C/D/E) |
| | | | Overall building height estimation | VQA | 100 | Multiple Choice(A/B/C/D/E) |
| | | CoT | Individual building height estimation | VQA | 101 | Multiple Choice(A/B/C/D/E) |
| | | | Overall building height estimation | VQA | 100 | Multiple Choice(A/B/C/D/E) |
| | Land Use | Perception | Overall land type classification | VQA | 509 | Multiple Choice(A/B/C/D/E) |
| | | | Fine-grained land type classification | VQA | 509 | Multiple Choice(A/B/C/D/E) |
| | | | Visual groudning of land types | Visual Grounding | 508 | Multiple Choice(A/B/C/D/E) |
| | | | Visual localization of land use types | VQA | 509 | Multiple Choice(A/B/C/D/E) |
| | | General Reasoning | Complex land counting | VQA | 449 | Multiple Choice(A/B/C/D/E) |
| Biosphere | Crop growth monitoring | Perception | Dead oil palm counting | VQA | 828 | Multiple Choice(A/B/C/D/E) |
| | | | Dead oil palm identification | VQA | 828 | Multiple Choice(A/B/C) |
| | Environmental pollution monitoring | Perception | Terrestrial oil spill area calculation | VQA | 123 | Multiple Choice(A/B/C/D/E) |
| | | | Terrestrial oil spill counting | VQA | 123 | Multiple Choice(A/B/C/D/E) |
| | Human footprint assessment | Scientific-Knowledge Reasoning | Human footprint assessment | VQA | 300 | Multiple Choice(A/B/C/D) |
| | | | Human footprint index estimation | VQA | 300 | Multiple Choice(A/B/C/D/E) |
| | Species Distribution Prediction | Perception | Tree species prediction | VQA | 500 | Multiple Choice(A/B/C/D/E) |
| | | | Tree species proportion prediction | VQA | 500 | Multiple Choice(A/B/C/D/E) |
| | | Expert- knowledge Deductive Reasoning | Animal classification | VQA | 108 | Multiple Choice(A/B/C/D/E) |
| | | | Geographical location inference of plant species | VQA | 500 | Multiple Choice(A/B/C/D/E) |
| | | | Global animal counting | VQA | 110 | Multiple Choice(A/B/C/D/E) |
| | | | Species distribution prediction | VQA | 1000 | Multiple Choice(A/B/C/D/E) |
| | Vegetation monitoring | Scientific-Knowledge Reasoning | Fractional vegetation cover estimation | VQA | 300 | Multiple Choice(A/B/C/D/E) |
| | | | Leaf area index estimation | VQA | 300 | Multiple Choice(A/B/C/D/E) |
| | | | Peak vegetation coverage area grounding | VQA | 300 | Multiple Choice(A/B/C/D/E) |
| Cross-sphere | Bird species prediction | Scientific-Knowledge Reasoning | Most likely species to occur | VQA | 900 | Multiple Choice(A/B/C/D/E) |
| | | | Species occurrence probability estimation | VQA | 453 | Multiple Choice(A/B/C/D/E) |
| | | | Species richness estimation | VQA | 900 | Multiple Choice(A/B/C/D/E) |
| | Carbon flux monitoring | Scientific-Knowledge Reasoning | Carbon flux estimation | VQA | 330 | Multiple Choice(A/B/C/D/E) |
| | Global Flood Forecasting | Scientific-Knowledge Reasoning | Flood detecting | VQA | 596 | Multiple Choice(A/B/C) |
| | | | Flood predicting | VQA | 277 | Multiple Choice(A/B/C) |
| Lithosphere | Earthquake monitoring and prediction | Perception | P-wave phase picking | VQA | 300 | Multiple Choice(A/B/C/D/E) |
| | | | S-wave phase picking | VQA | 300 | Multiple Choice(A/B/C/D/E) |
| | | | Earthquake or noise classification | VQA | 300 | Multiple Choice(A/B/C) |
| | | Scientific-Knowledge Reasoning | Earthquake magnitude estimation | VQA | 300 | Multiple Choice(A/B/C/D/E) |
| | | | Earthquake source-receiver distance inference | VQA | 300 | Multiple Choice(A/B/C/D/E) |
| | Geophysics imaging | Perception | Salt body location | Visual Grounding | 302 | Multiple Choice(A/B/C/D/E) |
| | | | Salt body detection | VQA | 329 | Multiple Choice(A/B/C) |

Table 2: **Characteristics of each task (L4 dimension) in Atmosphere, Oceansphere and Cryosphere.** Atmosphere has 33 subtasks, Oceansphere has 8 subtasks, Cryosphere has 8 subtasks.

| L1 | L2 | L3 | L4 Task Description | Format | Samples | Answer Type |
|---|---|---|---|---|---|---|
| Atmosphere | Interannual climate variation | Perception | ENSO feature analysis | VQA | 86 | Multiple Choice(A/B/C/D/E) |
| | | | Long-term Precipitation trend analysis | VQA | 50 | Multiple Choice(A/B/C/D/E) |
| | | | Long-term Temperature trend identification | VQA | 51 | Multiple Choice(A/B/C/D/E) |
| | Medium-term weather events | Perception | Cyclone movement identification | VQA | 185 | Multiple Choice(A/B/C/D/E) |
| | | | Cyclone phase identification | VQA | 90 | Multiple Choice(A/B/C/D/E) |
| | | | Event localization | VQA | 93 | Multiple Choice(A/B/C/D/E) |
| | | | Event intensity identification | VQA | 594 | Multiple Choice(A/B/C/D/E) |
| | | | Event onset identification | VQA | 42 | Multiple Choice(A/B/C/D/E) |
| | | | Event trend analysis | VQA | 575 | Multiple Choice(A/B/C/D/E) |
| | | | Geopotential pattern identification | VQA | 33 | Multiple Choice(A/B/C/D/E) |
| | | | Moisture flux analysis | VQA | 150 | Multiple Choice(A/B/C/D/E) |
| | | | System identification | VQA | 231 | Multiple Choice(A/B/C/D/E) |
| | | Scientific-Knowledge Reasoning | System evolution trend analysis | VQA | 91 | Multiple Choice(A/B/C/D/E) |
| | SEVIR Weather | Scientific-Knowledge Reasoning | Event type prediction | VQA | 300 | Multiple Choice(A/B/C/D/E) |
| | | | Miss alarm estimation | VQA | 300 | Multiple Choice(A/B/C/D/E) |
| | | | Movement prediction | VQA | 200 | Multiple Choice(A/B/C/D/E) |
| | | | Rotate center prediction | VQA | 93 | Multiple Choice(A/B/C/D/E) |
| | Seasonal weather events | Perception | Precipitation anomaly identification | VQA | 75 | Multiple Choice(A/B/C/D/E) |
| | | | Seasonal comparison | VQA | 101 | Multiple Choice(A/B/C/D/E) |
| | | | Temperature anomaly identification | VQA | 75 | Multiple Choice(A/B/C/D/E) |
| | Short-term weather events | Perception | Event intensity identification | VQA | 323 | Multiple Choice(A/B/C/D/E) |
| | | | Event localization | VQA | 133 | Multiple Choice(A/B/C/D/E) |
| | | | Event trend analysis | VQA | 297 | Multiple Choice(A/B/C/D/E) |
| | | | Event type identification | VQA | 139 | Multiple Choice(A/B/C/D/E) |
| | | Scientific-Knowledge Reasoning | Dynamic feature identification | VQA | 40 | Multiple Choice(A/B/C/D/E) |
| | | | Event evolution analysis | VQA | 90 | Multiple Choice(A/B/C/D/E) |
| | | | Thermodynamic feature identification | VQA | 40 | Multiple Choice(A/B/C/D/E) |
| | Typhoon | Scientific-Knowledge Reasoning | Pressure estimation | VQA | 847 | Multiple Choice(A/B/C/D/E) |
| | | | Radius of major gale axis estimation | VQA | 847 | Multiple Choice(A/B/C/D/E) |
| | | | Radius of minor gale axis estimation | VQA | 847 | Multiple Choice(A/B/C/D/E) |
| | | | Radius of major storm axis estimation | VQA | 847 | Multiple Choice(A/B/C/D/E) |
| | | | Radius of minor storm axis estimation | VQA | 847 | Multiple Choice(A/B/C/D/E) |
| | | | Wind estimation | VQA | 847 | Multiple Choice(A/B/C/D/E) |
| Cryosphere | Glacier analysis | Perception | Glacial Lake Recognition | VQA | 12 | Multiple Choice(A/B/C/D/E) |
| | | Scientific-Knowledge Reasoning | Glacier Melting Estimation | VQA | 10 | Multiple Choice(A/B/C/D) |
| | | | Slide Recognition | VQA | 8 | Multiple Choice(A/B/C) |
| | Sea ice forecast | Scientific-Knowledge Reasoning | SIC Estimate SIT | VQA | 20 | Multiple Choice(A/B/C/D/E) |
| | | | SIC Estimate SIV | VQA | 20 | Multiple Choice(A/B/C/D/E) |
| | | | SIT Trend Prediction | VQA | 30 | Multiple Choice(A/B/C/D/E) |
| | | | SIV Trend Prediction | VQA | 30 | Multiple Choice(A/B/C/D/E) |
| | | | Sea Ice Extent Estimation | VQA | 100 | Multiple Choice(A/B/C/D/E) |
| Oceansphere | Extreme Events | Perception | Enso identification | VQA | 146 | Multiple Choice(A/B/C/D/E) |
| | | | Iod identification | VQA | 140 | Multiple Choice(A/B/C/D/E) |
| | | Scientific-Knowledge Reasoning | Enso forecast | VQA | 152 | Multiple Choice(A/B/C/D) |
| | | | Iod forecast | VQA | 145 | Multiple Choice(A/B/C/D) |
| | Marine Debris and Oil Pollution | Perception | Marine Pollution Type Classification | VQA | 110 | Multiple Choice(A/B/C/D/E) |
| | Phenomenon Detection | Perception | Eddy identification | VQA | 204 | Multiple Choice(A/B/C/D/E) |
| | | | Marine fog detection | VQA | 200 | Multiple Choice(A/B/C) |
| | | | Eddy identificaEddy Localizationtion | Visual Grounding | 166 | Multiple Choice(A/B/C/D/E) |

## 1.3 Detailed results of specific sub-tasks (L-4 dimension).

**Worse performance of Qwen2.5-VL.** Qwen2.5-VL lagged behind contemporary open-source models like InterVL3 on Earth-related tasks, with both its 7B and 72B versions rarely ranking first in any L4 subtask. Despite its larger parameter size, the 72B model often scored zero on multiple tasks. However, this low accuracy shouldn't be seen as a lack of capability.Qwen2.5-VL often responds with "E (unable to answer)" when lacking domain knowledge—an honest approach. In contrast, some models tend to guess when uncertain, which is less desirable.

**Many models scored zero on various sub-tasks.** Even the top-performing InternVL3-78B failed on several. GPT-4o, a widely used closed-source model, recorded zero accuracy on nearly half the tasks. These results underscore the effectiveness and domain specificity of our sub-task design.

**No significant gap between closed-source and open-source models.** Although Gemini and Claude3 slightly lag behind LLaVA-OneVision and InternVL3 on sub-tasks, the gap is minimal. This indicates that open-source multimodal models are still well-suited for advancing Earth science research and agent development, without relying exclusively on closed-source alternatives.

**Poor performance of some subtasks in Cross-sphere.** In the cross-sphere species richness prediction task, no model surpassed 10% accuracy on 900 test samples, which aligns with expectations given the task's complexity. Integrating climate variables, satellite imagery, and vegetation factors creates a highly intricate prediction challenge beyond the capabilities of current models.

Table 3: **CoT performance of each L4 subtasks.** We mark the highest score of each metric in red , and second highest underlined.

| Task | Num. | Qwen2.5-VL-7B | | | LLaVA-OneVision-7b | | | InternVL3-8B | | | InternVL3-78B | | |
|---|---|---|---|---|---|---|---|---|---|---|---|---|---|
| | | Percision | Recall | F1 | Percision | Recall | F1 | Percision | Recall | F1 | Percision | Recall | F1 |
| Multi-image Visual Localization | 102 | 95.87 | 31.21 | 47.09 | 93.28 | 40.36 | 56.34 | 93.89 | 41.34 | 57.40 | 97.49 | 43.74 | 60.39 |
| Visual grounding of damaged individual buildings | 102 | 95.97 | 25.98 | 40.89 | 92.38 | 22.92 | 36.73 | 95.86 | 33.47 | 49.62 | 91.99 | 42.58 | 58.21 |
| Overall building height estimation | 100 | 83.97 | 14.50 | 24.73 | 83.73 | 18.17 | 29.86 | 93.82 | 20.00 | 32.97 | 92.59 | 19.50 | 32.22 |
| Spatial relationships under complex conditions | 99 | 94.51 | 35.98 | 52.12 | 91.75 | 22.27 | 35.84 | 93.79 | 36.60 | 52.65 | 96.25 | 32.55 | 48.65 |
| Individual building damage assessment | 107 | 93.21 | 37.54 | 53.52 | 87.93 | 12.96 | 22.59 | 92.79 | 40.74 | 56.62 | 95.58 | 39.06 | 55.46 |
| Individual building height estimation | 101 | 92.49 | 12.71 | 22.34 | 87.69 | 13.2 | 22.95 | 91.77 | 13.37 | 23.34 | 90.46 | 14.36 | 24.78 |

**InternVL3 outperform other MLLMs in CoT tasks.** The InterVL3 series performed strongly on CoT tasks, with both the 7B and 78B models achieving top results across all subtasks, showcasing their strength in geoscience chain-of-thought reasoning. Future geoscience reasoning tasks could benefit from further training and application of this series.

Table 4: **Visual Grounding performance one each L4 subtasks.**

| L4 | Num | Qwen2.5-VL-7B | | LLaVA-OneVision-7b | | InternVL3-8B | | InternVL3-78B | | GPT-4o | | Gemini-2.0 | | Claude-3-7 | |
|---|---|---|---|---|---|---|---|---|---|---|---|---|---|---|---|
| | | acc@0.5 | acc@0.7 | acc@0.5 | acc@0.7 | acc@0.5 | acc@0.7 | acc@0.5 | acc@0.7 | acc@0.5 | acc@0.7 | acc@0.5 | acc@0.7 | acc@0.5 | acc@0.7 |
| Salt Body Location | 302 | 0 | 0 | 5.3 | 0.33 | 8.94 | 1.66 | 4.3 | 0.33 | 0.08 | 0 | 0.13 | 0.04 | 0.02 | 0 |
| Eddy Localization | 166 | 3.01 | 0 | 1.81 | 0.6 | 6.63 | 0.6 | 13.86 | 3.61 | 0.12 | 0.01 | 0.34 | 0.06 | 0.2 | 0.07 |
| Visual Grounding of Land Types | 508 | 0.2 | 0.00 | 0.59 | 0.20 | 2.56 | 0.59 | 2.36 | 0.20 | 0.02 | 0.00 | 0.03 | 0.00 | 0.00 | 0.00 |

**Visual grounding performance is notably poor across all models** It exposing two main shortcomings: limited geoscientific knowledge and weak visual localization capabilities. Both open- and closed-source models fall short in these aspects.

Table 5: **VQA Performance in L4 dimension.** We mark the highest score of each metric in `red`, and second highest underlined.

| Domain | L4-task | Num | Qwen2.5-VL | | InternVL3 | | LLaVA-OV | GPT-4o | Gemini | Claude3 |
|---|---|---|---|---|---|---|---|---|---|---|
| | | | 7B | 72B | 8B | 78B | 7B | | | |
| Human-activities | Change detection... | 502 | 6.97 | 1.39 | 43.03 | 72.51 | 85.06 | 0 | 74.1 | 10.36 |
| | Partially damaged... | 498 | 1.41 | 0 | 4.02 | 45.98 | 50.6 | 0 | 11.24 | 0.8 |
| | Building damage... | 499 | 0.2 | 4.81 | 5.81 | 7.62 | 33.87 | 0 | 8.22 | 4 |
| | Disaster prediction | 500 | 0.8 | 5.6 | 4 | 0.6 | 0.6 | 0 | 0.2 | 2.8 |
| | Disaster type... | 500 | 1 | 6.2 | 6.6 | 8.2 | 1.6 | 0.4 | 4.2 | 11.2 |
| | Geolocation... | 502 | 12.15 | 9.76 | 36.25 | 44.82 | 31.67 | 2.59 | 36.45 | 33.47 |
| | Fine-grained object... | 514 | 10.89 | 19.46 | 21.98 | 30.35 | 48.05 | 0 | 47.67 | 16 |
| | Overall counting | 502 | 1.99 | 2.59 | 18.13 | 17.13 | 21.12 | 0.2 | 31.67 | 0 |
| | Counting complex... | 754 | 0.4 | 0 | 31.75 | 19.05 | 18.25 | 0.27 | 9.28 | 0.93 |
| | Overall land... | 509 | 0 | 0.2 | 1.96 | 1.38 | 1.38 | 0 | 1.57 | 1.18 |
| | Fine-grained land... | 509 | 3.93 | 6.68 | 26.13 | 6.48 | 39.88 | 39.29 | 30.26 | 32 |
| | Visual localization... | 509 | 1.38 | 3.54 | 5.11 | 4.52 | 1.77 | 0.39 | 6.68 | 1.57 |
| | Land types ... | 449 | 3.12 | 3.34 | 24.5 | 16.04 | 14.7 | 0 | 5.35 | 14.02 |
| | Individual building ... | 107 | 0.93 | 7.48 | 28.97 | 24.3 | 47.66 | 0 | 35.51 | 14.02 |
| | Multi individual ... | 102 | 10.78 | 13.73 | 28.43 | 53.92 | 67.65 | 8.82 | 41.81 | 43.14 |
| | Spatial relation ... | 99 | 22.22 | 21.21 | 33.33 | 36.36 | 23.23 | 2.02 | 39.39 | 16.16 |
| | Visual grounding ... | 102 | 35.29 | 37.25 | 52.94 | 52.94 | 65.69 | 28.43 | 56.86 | 37.25 |
| | Overall height ... | 509 | 0 | 0.2 | 1.96 | 1.38 | 1.38 | 0 | 1.57 | 1.18 |
| | Individual height ... | 101 | 0 | 0 | 45.54 | 0 | 38.61 | 0 | 0 | 0 |
| Biosphere | Dead oil .. counting | 828 | 52.9 | 47.95 | 48.67 | 73.67 | 73.67 | 0 | 56.04 | 60.51 |
| | oil ..identification | 828 | 85.51 | 70.65 | 81.52 | 83.33 | 85.63 | 0 | 51.81 | 77.29 |
| | oil spill area ... | 123 | 0 | 0 | 5.69 | 5.69 | 0 | 0 | 0 | 0 |
| | oil spill counting ... | 123 | 0 | 0.81 | 41.46 | 53.66 | 22.76 | 0 | 7.32 | 0 |
| | footprint assess .. | 300 | 0.33 | 0.67 | 36 | 26.33 | 22.67 | 0.67 | 7.67 | 26.67 |
| | footprint index .. | 300 | 0 | 0 | 3.33 | 3.33 | 23.33 | 0 | 0 | 20 |
| | species prediction .. | 500 | 5 | 15.2 | 35.6 | 46.4 | 46.8 | 0.8 | 15.8 | 13.2 |
| | species proportion.. | 500 | 0 | 0.2 | 26.2 | 1.8 | 18.4 | 0 | 5.2 | 2.4 |
| | Animal classification | 108 | 4.63 | 1.85 | 15.74 | 27.78 | 43.52 | 0 | 9.26 | 2.78 |
| | Geographical ... | 500 | 4.6 | 13.4 | 18.2 | 26 | 50 | 10.4 | 38.6 | 75.8 |
| | Species distribution... | 1000 | 2.1 | 68.6 | 73.3 | 78.1 | 40.1 | 16.8 | 91.5 | 90.2 |
| | Fractional ... | 300 | 0 | 0 | 13 | 25 | 56 | 0 | 3.67 | 46 |
| | Leaf area index... | 300 | 0 | 0 | 7.33 | 14 | 50 | 0 | 0 | 29.33 |
| | animal counting... | 110 | 0 | 3.64 | 0 | 6.36 | 2.73 | 0.91 | 7.27 | 0 |
| | Peak vegetation... | 300 | 9 | 0.67 | 25 | 8.33 | 24 | 0 | 18.3 | 24 |
| Cross-sphere | Most likely species... | 900 | 0.11 | 18.89 | 20.89 | 40.67 | 21.89 | 0.22 | 36.67 | 59.22 |
| | Species occurrence... | 453 | 2.65 | 4.64 | 58.5 | 13.69 | 8.17 | 0 | 3.31 | 29.8 |
| | Species richness ... | 900 | 0 | 0 | 9 | 0.33 | 7.67 | 0 | 0 | 0 |
| | Carbon flux .. | 330 | 11.52 | 0 | 24.85 | 0.61 | 25.45 | 0 | 0.61 | 0 |
| | Flood detecting | 596 | 44.8 | 0 | 52.18 | 51.51 | 52.35 | 0 | 28.52 | 51 |
| | Flood predicting | 277 | 0 | 0 | 91.7 | 8.3 | 0 | 0 | 32.49 | 44.04 |
| Cryosphere | Glacial Lake .. | 12 | 25 | 25 | 75 | 75 | 75 | 50 | 66.67 | 66.67 |
| | Glacier Melting... | 10 | 30 | 10 | 40 | 50 | 30 | 10 | 30 | 40 |
| | Slide Recognition... | 8 | 65.5 | 0 | 37.5 | 62.5 | 62.5 | 12.5 | 62.5 | 50 |
| | SIC Estimate SIT | 20 | 0 | 0 | 55 | 100 | 45 | 85 | 80 | 90 |
| | SIC Estimate SIV | 20 | 0 | 0 | 45 | 80 | 40 | 50 | 70 | 80 |
| | SIT Trend Prediction | 30 | 3.33 | 0 | 50 | 90 | 50 | 40 | 46.7 | 60 |
| | SIV Trend Prediction | 30 | 0 | 0 | 36.67 | 80 | 36.67 | 13.33 | 53.33 | 20 |
| | Sea Ice Extent... | 100 | 19 | 12 | 55 | 59 | 32 | 39 | 59 | 29 |
| Lithosphere | P-wave phase picking | 300 | 8 | 11.33 | 10.67 | 16 | 8 | 6 | 22.33 | 11 |
| | S-wave phase picking | 300 | 36.67 | 35.33 | 61.67 | 41 | 32 | 16.67 | 49.33 | 28.33 |
| | Earthquake or noise .. | 300 | 44.67 | 86.33 | 63 | 59.33 | 52.33 | 50 | 93.33 | 95 |
| | magnitude estim... | 300 | 1.33 | 0 | 38.33 | 0 | 32.67 | 0 | 26.67 | 1.67 |
| | source-receiver ... | 300 | 20.33 | 0.67 | 35.33 | 6.67 | 33 | 1.67 | 14 | 21.67 |
| | Salt body detection | 329 | 0.91 | 0.91 | 15.81 | 17.33 | 14.29 | 2.43 | 27.96 | 11.25 |

Table 6: **VQA Performance in L4 dimension.** We mark the highest score of each metric in red, and second highest underlined.

| Sphere | L4-task | Num | Qwen2.5-VL | | InternVL3 | | LLaVA-OV | GPT-4o | Gemini | Claude3 |
|---|---|---|---|---|---|---|---|---|---|---|
| | | | 7B | 72B | 8B | 78B | 7B | | | |
| Atmosphere | Cyclone move... | 185 | 8.65 | 2.16 | 37.84 | 47.03 | 34.59 | 6.49 | 38.38 | 44.32 |
| | Cyclone phase... | 90 | 0 | 0 | 48.89 | 0 | 25.56 | 1.11 | 25.56 | 5.75 |
| | Event intensity... | 594 | 17.34 | 44.61 | 38.72 | 0 | 30.14 | 13.21 | 53.62 | 33.28 |
| | Event localization | 93 | 18.28 | 5.38 | 46.24 | 41.94 | 33.33 | 18.98 | 51.82 | 43.07 |
| | Event onset... | 42 | 0 | 0 | 26.19 | 0 | 35.71 | 16.67 | 30.95 | 38.1 |
| | Event trend... | 575 | 0.18 | 0 | 48.52 | 0 | 36.87 | 2.96 | 39.82 | 14.7 |
| | Geopotential... | 33 | 18.18 | 12.12 | 18.18 | 15.15 | 12.12 | 18.18 | 9.09 | 30.3 |
| | Moisture flux... | 150 | 0 | 0 | 66.00 | 0 | 53.34 | 0 | 0 | 6.12 |
| | System... | 231 | 4.33 | 17.75 | 33.77 | 0 | 22.94 | 18.18 | 40.69 | 52 |
| | System evolution... | 91 | 2.2 | 0 | 40.66 | 0 | 56.04 | 17.58 | 57.14 | 70 |
| | Event intensity... | 323 | 18.89 | 13.62 | 50.77 | 0 | 34.37 | 30.64 | 59.54 | 18.31 |
| | Event localization | 133 | 1.5 | 0.75 | 47.37 | 0 | 13.53 | 25.85 | 50.68 | 21.05 |
| | Event trend... | 297 | 3.03 | 0 | 31.65 | 0 | 35.69 | 4.71 | 41.08 | 19.57 |
| | Event type... | 139 | 23.74 | 17.27 | 36.69 | 0 | 25.9 | 23.74 | 27.34 | 0 |
| | Dynamic feature... | 40 | 45 | 67.5 | 37.5 | 0 | 2.5 | 15 | 27.5 | 30.77 |
| | Event evolution... | 90 | 0 | 0 | 24.44 | 0 | 2.22 | 0 | 21.11 | 8 |
| | Thermodynamic... | 40 | 0 | 0 | 15 | 0 | 0 | 7.5 | 20 | 15.79 |
| | Pressure... | 847 | 0 | 0 | 0.83 | 0 | 30.34 | 0 | 0 | 0 |
| | Radius (gale)... | 847 | 0 | 0 | 21.49 | 0.59 | 24.91 | 0 | 0 | 35.42 |
| | Radius (storm)... | 847 | 0 | 0 | 23.02 | 0.71 | 14.76 | 0 | 0 | 47.23 |
| | minor gale ... | 847 | 0 | 0 | 0 | 5.79 | 15.47 | 0 | 0 | 0 |
| | minor storm ... | 847 | 0 | 0 | 1.89 | 0 | 4.84 | 0 | 0 | 0 |
| | Wind estimation | 847 | 0 | 0 | 0 | 5.79 | 30.46 | 0 | 0 | 0 |
| | Precipitation ... | 75 | 0 | 1.33 | 21.33 | 28 | 22.67 | 6.67 | 28 | 16 |
| | Seasonal ... | 101 | 8.91 | 9.9 | 46.53 | 0 | 45.54 | 27.52 | 40.5 | 40 |
| | Temperature ... | 75 | 16 | 18.67 | 45.33 | 57.33 | 48 | 29.33 | 49.33 | 48 |
| | ENSO feature... | 86 | 41.86 | 63.95 | 75.58 | 0 | 77.91 | 90.7 | 75.58 | 73.26 |
| | Long.. Precipitation | 50 | 0 | 0 | 34 | 48 | 26 | 20 | 26 | 32 |
| | Long.. Temperature | 51 | 0 | 0 | 49.02 | 60.78 | 27.45 | 39.22 | 25.49 | 27.45 |
| | Event type .. | 300 | 15 | 0 | 36 | 19 | 42.67 | 20.67 | 1 | 16 |
| | Miss alarm .. | 300 | 0 | 0 | 19 | 22 | 32 | 0 | 0.67 | 8.33 |
| | Movement .. | 200 | 45 | 21 | 76.5 | 60.5 | 81 | 2 | 69 | 64.5 |
| | Rotate center... | 93 | 3.23 | 2.15 | 62.37 | 75.27 | 46.24 | 0 | 6.45 | 58.06 |
| Oceansphere | ENSO .. | 146 | 21.92 | 34.93 | 33.56 | 17.81 | 23.97 | 31.51 | 26.03 | 51.37 |
| | IOD Identification | 140 | 4.29 | 19.29 | 12.86 | 4.29 | 12.14 | 50 | 5.71 | 12.86 |
| | ENSO Forecast | 152 | 0 | 3.29 | 23.03 | 36.18 | 15.79 | 6.58 | 23.03 | 19.74 |
| | IOD Forecast | 145 | 0 | 0 | 4.83 | 18.62 | 18.62 | 0 | 0.69 | 0 |
| | Eddy Identification | 204 | 3.93 | 0.49 | 3.92 | 4.9 | 44.1 | 1.47 | 4.9 | 14.22 |
| | Marine Fog .. | 200 | 67.5 | 55 | 61 | 57 | 53.5 | 3 | 55.5 | 55.5 |
| | Marine Pollution... | 110 | 0 | 0.91 | 2.73 | 2.73 | 3.64 | 0.91 | 2.73 | 8.18 |

 **1.4 Detailed results of specific sub-tasks (L-3 capability).**

This section highlights the performance of MLLMs across all L-3 capabilities. The VQA task is split into perception, General reasoning and Scientific-Knowledge Reasoning dimensions, with results shown in Tables 7.

Table 7: **VQA Performance in L3 dimension.** We mark the highest score of each metric in red , and second highest underlined.

| L4-task | Qwen2.5-VL | | InternVL3 | | LLaVA-OV | GPT-4o | Gemini | Claude3 |
|---|---|---|---|---|---|---|---|---|
| | 7B | 72B | 8B | 78B | 7B | | | |
| Perception | 13.42 | 15.37 | 35.77 | 24.68 | 34.66 | 15.40 | 33.33 | 26.97 |
| General Reasoning | 7.32 | 10.86 | 28.67 | 27.48 | 30.71 | 3.62 | 21.22 | 13.74 |
| Scientific-Knowledge Reasoning | 8.2 | 5.72 | 30.89 | 27.49 | 30.18 | 8.74 | 23.66 | 31.62 |

Model performance varies across L3 dimensions: InterVL3 excels in perception, LLaVA-OneVision in general reasoning, and Claude3 in scientific knowledge reasoning. Overall, InterVL3 shows consistently strong results, while Qwen2.5VL and GPT-4o fall notably behind.

## 1.5 Samples and challenging Cases of OmniEarth-Bench

In this section, we construct a detailed table (Tab. 8) analyzing model performance and error causes for L-4 subtask. We then use examples to thoroughly illustrate the errors for typical subtask. In this section, we present a case study analysis of the error types made by Gemini-2.0-Flash [1], Qwen2.5-VL [2], and InterVL3 [3] on various sub-tasks in OmniEarth-Bench. We classify the errors into the following 6 categories:

**Spatio-temporal Frame Confusion** :The model mis-interprets either the time sequence or the spatial orientation / coordinate frame of the data, leading to reversed trend, direction, or geographic reference. See examples in Fig. 1, Fig. 7, etc.

**Threshold / Severity Mis-estimation** :Numeric values are read incorrectly or wrong thresholds applied, so strength or severity categories are wrong. See examples in Fig. 6, Fig. 9, etc.

**Image-feature Misinterpretation** : Visual cues (texture, color, shape) are mis-read; key features are missed or artefacts are mistaken for real features. See examples in Fig. 10, Fig. 11, etc.

**Domain-knowledge / Semantic Mis-match** :Mis-application of non-visual expertise – ecology, climate thresholds, hazard mechanics – so scene is matched to an incorrect knowledge template. See examples in Fig. 3.

**Over-cautious / Refusal** :Adequate information is available, but the model answers "Unable to decide" (or hedges) to avoid committing. See examples in Fig. 2,Fig. 4, etc.

**Target Mis-location** :The object or area specified in the prompt is not correctly identified, so all subsequent reasoning is off target. See examples in Fig. 8.

Table 8: Table index of case study figures by sub-tasks (L-3 capability) with associated (error) categories for each MLLM.

| Case | L-1 task | L-4 task | Gemini | Qwen2-VL | Internvl3 |
|---|---|---|---|---|---|
| Fig. 1 | Atmosphere | Movement Prediction | Spatio-temporal Frame Confusion | Spatio-temporal Frame Confusion | Correct |
| Fig. 2 | Biosphere | Dead Oil Palm counting | Correct | Over-cautious / Refusal | Over-cautious / Refusal |
| Fig. 3 | Biosphere | Species Distribution Prediction | Correct | Domain-knowledge / Semantic Mis-match | Correct |
| Fig. 4 | Cross-sphere | Most likely species to occur | Correct | Over-cautious / Refusal / Domain-knowledge / Semantic Mis-match | Over-cautious / Refusal |
| Fig. 5 | Cross-sphere | Global Flood Forecasting | Correct | Image-feature Misinterpretation / Threshold / Severity Mis-estimation | Image-feature Misinterpretation / Over-cautious / Refusal |
| Fig. 6 | Cryosphere | SIC Estimate SIT | Correct | Threshold / Severity Mis-estimation / Image-feature Misinterpretation | Threshold / Severity Mis-estimation / Image-feature Misinterpretation |
| Fig. 7 | Cryosphere | Sea Ice Extent Estimation | Spatio-temporal Frame Confusion | Spatio-temporal Frame Confusion | Correct |
| Fig. 8 | Lithosphere | P-wave phase picking | Target Mis-location / Domain-knowledge / Semantic Mis-match | Target Mis-location / Domain-knowledge / Semantic Mis-match | Target Mis-location / Domain-knowledge / Semantic Mis-match |
| Fig. 9 | Oceansphere | ENSO Identification | Threshold / Severity Mis-estimation | Spatio-temporal Frame Confusion / Image-feature Misinterpretation | Spatio-temporal Frame Confusion / Image-feature Misinterpretation |
| Fig. 10 | Oceansphere | Marine Fog Detection | Image-feature Misinterpretation | Image-feature Misinterpretation | Image-feature Misinterpretation |
| Fig. 11 | Human-activities sphere | Fine-grained object type recognition | Image-feature Misinterpretation | Image-feature Misinterpretation | Image-feature Misinterpretation |
| Fig. 12 | Lithosphere | earthquake source-receiver distance inference | Correct | Target Mis-location | Over-cautious / Refusal |

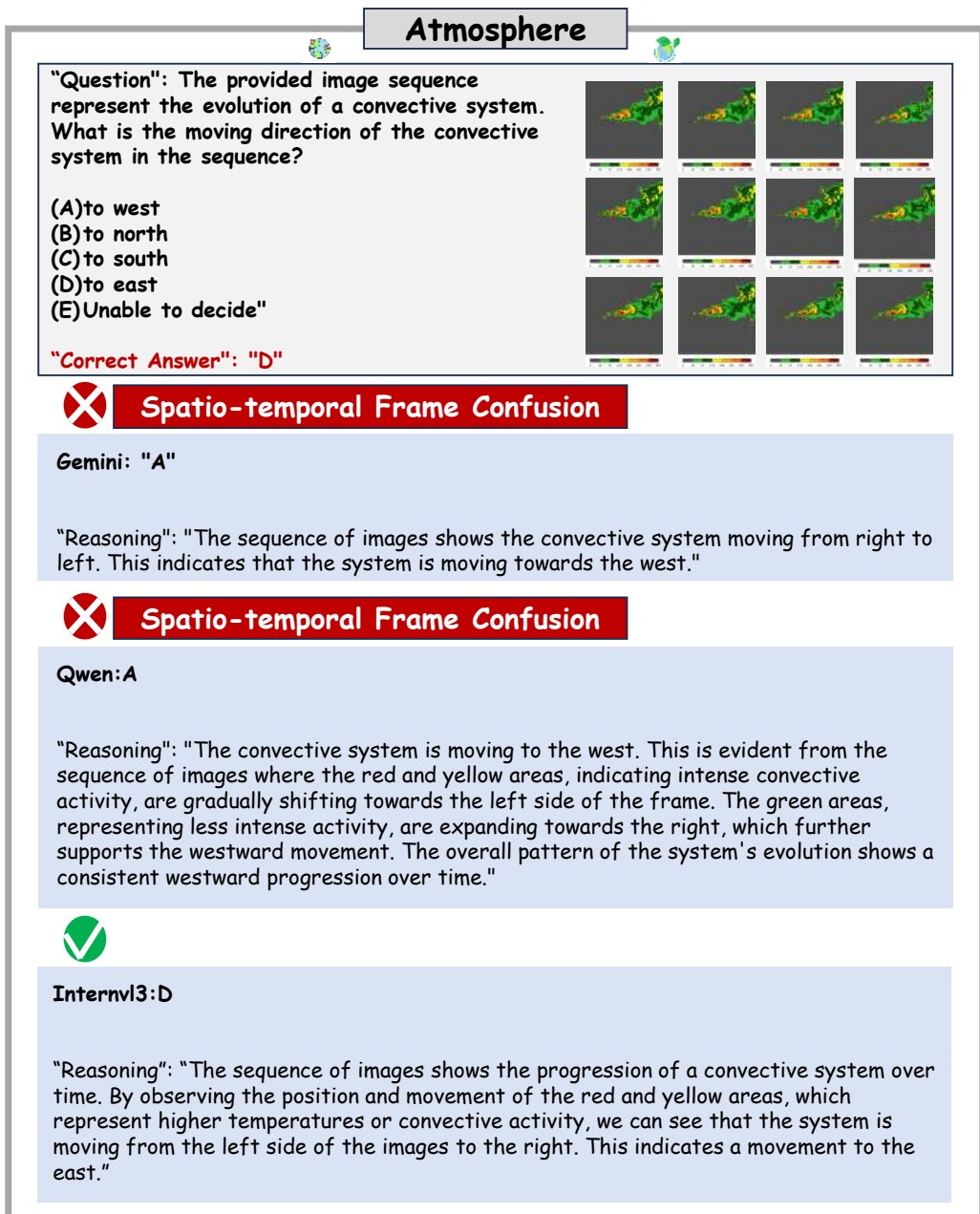

Figure 1: A sample case of Movement Prediction subtask.

## Biosphere

"Question": "This is an unmanned aerial vehicle (UAV) image capturing oil palm trees. How many dead trees are in this image?"
(A) 3
(B) 8
(C) 11
(D) 1
(E) Unable to decide

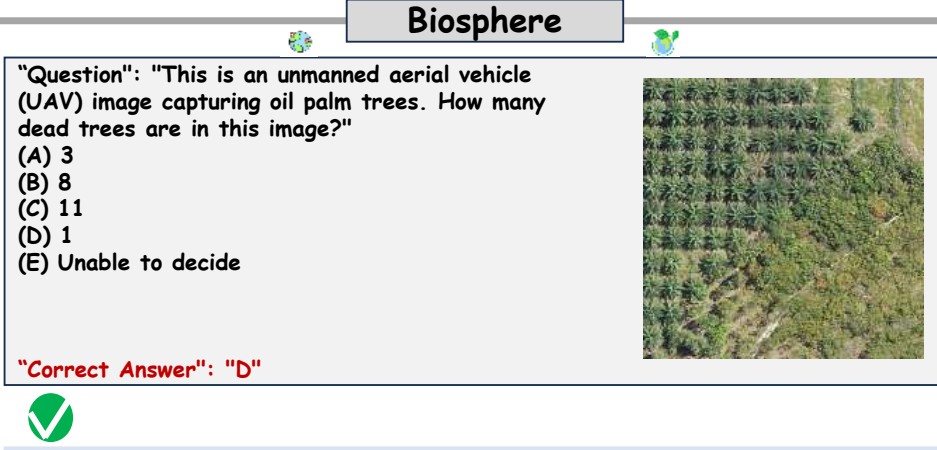

"Correct Answer": "D"

Gemini: D

"Reasoning": "Based on the image, I can only identify one tree that appears to be dead or severely damaged. It is located in the upper-middle section of the image, and it has a significantly different color and structure compared to the healthy green oil palm trees."

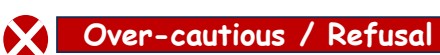 ❌ **Over-cautious / Refusal**

Qwen:E

"Reasoning": "The image shows a dense plantation of oil palm trees with no clear signs of dead trees. The trees appear uniformly green and healthy, with no visible brown or withered foliage that would indicate dead trees. The image resolution and angle may not be sufficient to detect subtle signs of tree death. Without more detailed information or a closer view, it is not possible to accurately count dead trees. Therefore, the best answer is 'E' - Unable to decide."

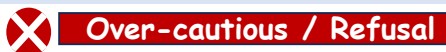 ❌ **Over-cautious / Refusal**

Internvl3:E

"Reasoning": "The image shows a large number of oil palm trees, but it is difficult to determine the exact number of dead trees without a closer inspection or additional information. The trees appear healthy and green, but there are no clear indicators of dead trees visible from this aerial view."

Figure 2: A sample case of Dead Oil Palm counting subtask.

## Biosphere

"Question": "Which species is most likely to live in the region shown at the image? The latitude is 60.1805513699° N, and the longitude is 24.9494441843° E."

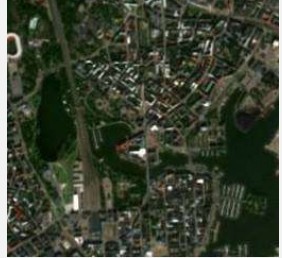

(A)Fringilla coelebs gengleri
(B)Acridotheres tristis
(C)Passer domesticus
(D)Anthornis melanura
(E)Unable to decide

"Correct Answer": "C"

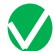

**Gemini: D**

"Reasoning": "The coordinates 60.1805513699° N, 24.9494441843° E indicate Helsinki, Finland. The image shows an urban environment. Passer domesticus (House Sparrow) is a common bird species found in urban areas worldwide, including Europe. The other options are less likely: Fringilla coelebs gengleri is a subspecies of Chaffinch found in the British Isles, Acridotheres tristis (Indian Myna) is native to Asia, and Anthornis melanura (New Zealand Bellbird) is native to New Zealand."

### ❌ Domain-knowledge / Semantic Mis-match

**Qwen:E**

"Reasoning": "The image shows a region with a mix of urban and green spaces, likely in a temperate climate zone. The coordinates place this region in Helsinki, Finland. Fringilla coelebs gengleri, a subspecies of the common chaffinch, is native to northern Europe and well-adapted to such environments. Acridotheres tristis, the common myna, is native to South Asia and not typically found in this region. Passer domesticus, the house sparrow, is common in urban areas but is not specific to this location. Anthornis melanura melanura, the tui, is native to New Zealand and not found in Europe. Therefore, Fringilla coelebs gengleri is the most likely species to live in this region."

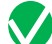

**Internvl3:E**

"Reasoning": "The image shows a densely populated urban area with water bodies, which is characteristic of many cities in temperate regions. The latitude and longitude provided suggest a location in the northern hemisphere, likely in a region with a temperate climate. The species listed are birds, and among them, Passer domesticus, commonly known as the house sparrow, is known to inhabit urban environments. The other species listed are either not typically found in urban settings or are more specific to different regions. Therefore, Passer domesticus is the most likely species to live in the region shown in the image."

Figure 3: A sample case of Species Distribution Prediction subtask.

## Cross-sphere

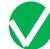

**"Question": "This image shows the satellite view of a bird hotspot, which is located at longitude 34.356469 and latitude -0.738459 in the state of Nyanza, Kenya. …Which of the following bird species is most likely to occur in this hotspot?"**
**(A) Pitta angolensis**
**(B) Amadina fasciata**
**(C) Streptopelia semitorquata**
**(D) Spatula querquedula**
**(E) Unable to decide**

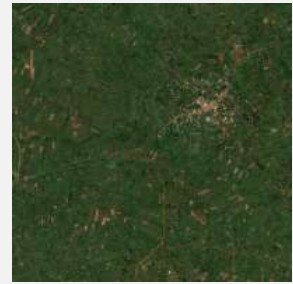

**"Correct Answer": "D"**

**Gemini: C**

**"Reasoning": "The image shows a landscape with a mix of vegetation and some built-up areas. The climate data indicates a relatively warm and wet environment. Streptopelia semitorquata (Red-eyed Dove) is a common bird in African woodlands and cultivated areas, which aligns with the habitat and climate suggested by the image and data. The other options are less likely: Pitta angolensis prefers dense forests, Spatula querquedula is a migratory duck and Amadina fasciata prefers drier areas."**

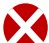

### ❌ Over-cautious / Refusal

**Qwen: B**

"Reasoning": "The satellite image shows a region with a mix of green vegetation and some human settlements, indicating a semi-arid to sub-humid environment. The bioclimatic variables suggest a warm climate with moderate precipitation, typical of savanna or woodland habitats. Amadina fasciata, commonly known as the black-rumped shama, is well-adapted to such environments. It prefers open woodlands and savannas, which align with the visual cues from the image. The other options are less likely: Pitta angolensis prefers dense forests, Streptopelia semitorquata is more common in arid regions, Spatula querquedula is a waterfowl requiring wetlands, and selecting 'E' would be too uncertain given the clear habitat match for Amadina fasciata."

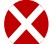

### ❌ Over-cautious / Refusal

**Internvl3: A**

"Reasoning": "The satellite image shows a bird hotspot in Nyanza, Kenya, with specific bioclimatic variables. Pitta angolensis, also known as the Angola Pitta, is known to inhabit areas with dense vegetation and a range of temperatures and precipitation levels similar to those described. The image shows a mix of agricultural fields and forested areas, which aligns with the habitat preferences of the Angola Pitta. The other options, such as Amadina fasciata, Streptopelia semitorquata, and Spatula querquedula, have different habitat requirements and are less likely to be found in this type of environment."

Figure 4: A sample case of Most likely species to occur subtask.

## Cross-sphere

"Question": "As a pan-Arctic researcher, you are given one sea ice concentration map (the first image) and one plot for daily trend in sea ice thickness (the second image). Considering the date of SIC map is the following day of the end of SIT trend plot, which choice would most likely to be the averaged SIT of the given SIC map?"

(A)0.946
(B)1.775
(C)1.369
(D)1.052
(E)Unable to decide"

"Correct Answer": "D"

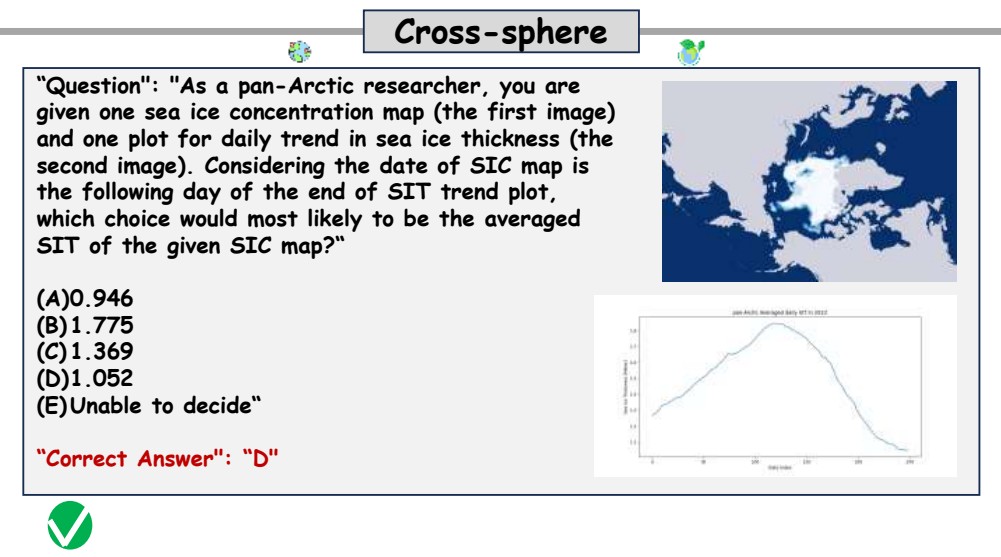

✅

**Gemini: B**

"Reasoning": "The SIC map is for the day after the SIT trend plot ends. The SIT trend plot ends at a date index of approximately 250, where the SIT is around 1.05 meters. Therefore, the most likely average SIT for the SIC map is 1.052."

❌ **Image-feature Misinterpretation**

**Qwen:C**

"Reasoning": "The SIT plot shows a clear downward trend, ending at approximately 1.1 meters. The SIC map indicates a significant amount of sea ice coverage, suggesting that the SIT is not at its lowest point. Given the options, 1.369 meters is the most reasonable choice as it is higher than the endpoint of the SIT plot but lower than the peak, aligning with the expected SIT for the next day."

❌ **Image-feature Misinterpretation**

**Internvl3:A**

"Reasoning": "The SIT trend plot shows a decreasing trend in sea ice thickness over time. The date of the SIC map is the following day after the end of the SIT trend plot. By examining the plot, the SIT value at the end of the trend (around day 250) is approximately 1.1 meters. Given the decreasing trend, the SIT on the following day would likely be lower than 1.1 meters. Among the provided choices, 0.946 is the only value lower than 1.1 meters, making it the most likely averaged SIT for the given SIC map."

Figure 5: A sample case of Global Flood Forecasting subtask.

## Cryosphere

**"Question":** "The first image is the GloFAS dis24 data, which represents the forecasted river discharge in cubic meters per second (m^3/s). The second image is the ERA5 temperature data at 2 meters above ground level, measured in Kelvin (K). The third image is the volumetric soil water content in the top layer of the soil, measured in cubic meters per cubic meter (m^3/m^3). The fourth image is the snow depth water equivalent, measured in meters (m). The fifth image is the total precipitation sum, measured in millimeters (mm). Whether flood occurs in the region on the date shown in the images ?"

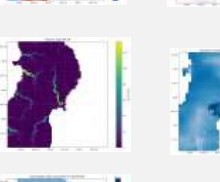

**(A)Flooding occurs**
**(B)Flooding does not occur**
**(C)Unable to decide"**

**"Correct Answer": "B"**

**Gemini: B**

"Reasoning": "The river discharge (dis24) shows moderate levels in the rivers. The total precipitation sum is very low, indicating no significant rainfall. The soil water content is also not particularly high. The snow depth water equivalent is negligible. Therefore, based on the available data, flooding is unlikely to occur."

### Threshold / Severity Mis-estimation

**Qwen:C**

"Reasoning": "The GloFAS dis24 data shows relatively low river discharge values, with no significant areas indicating high discharge that would typically suggest flooding. The temperature data does not show any extreme conditions that would contribute to flooding, such as rapid snowmelt. The soil water content is moderate, with no signs of saturation that could lead to runoff. The snow depth water equivalent is very low, indicating minimal snowmelt contribution. The total precipitation sum is also low, with no heavy rainfall that could cause flooding. However, without additional context or data, such as real-time river levels or local topography, it is not possible to definitively conclude whether flooding occurs or not."

### Domain-knowledge / Semantic Mis-match

**Internvl3:A**

"Reasoning": "The first image shows high river discharge values, indicating potential flooding. The volumetric soil water content is high, suggesting saturated soil conditions. The snow depth water equivalent is also significant, which could contribute to runoff. The temperature is relatively low, which might slow down evaporation, further increasing the risk of flooding. These factors combined suggest that flooding is likely to occur in the region on the date shown."

Figure 6: A sample case of SIC Estimate SIT subtask.

## Cryosphere

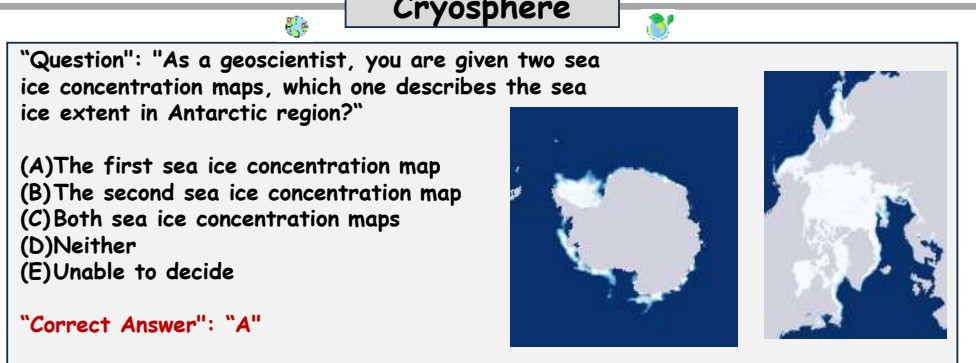

**"Question":** "As a geoscientist, you are given two sea ice concentration maps, which one describes the sea ice extent in Antarctic region?"

(A)The first sea ice concentration map
(B)The second sea ice concentration map
(C)Both sea ice concentration maps
(D)Neither
(E)Unable to decide

**"Correct Answer": "A"**

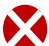 **Spatio-temporal Frame Confusion**

**Gemini: B**

"Reasoning": "The second image shows a significantly reduced sea ice extent compared to the first image. During the melting season, sea ice extent decreases substantially due to warmer temperatures. Therefore, the second image is more likely to represent sea ice extent during the melting season."

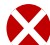 **Spatio-temporal Frame Confusion**

**Qwen:B**

"Reasoning": "The second sea ice concentration map shows a significantly reduced area of sea ice compared to the first map. This is indicative of the melting season, where sea ice extent typically decreases. The first map displays a larger and more continuous ice coverage, suggesting it represents a time of year with less melting, such as winter or early spring. The second map's fragmented and reduced ice coverage aligns with the characteristics of the melting season, where ice retreats and breaks up. Therefore, the second map is more likely to represent the sea ice extent during the melting season."

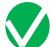

**Internvl3:A**

"Reasoning": "The first sea ice concentration map shows a large ice-covered area in the southern hemisphere, which is characteristic of the Antarctic region. The second map shows ice in the northern hemisphere, which is characteristic of the Arctic region. Therefore, the first map describes the sea ice extent in the Antarctic region."

Figure 7: A sample case of Sea Ice Extent Estimation subtask.

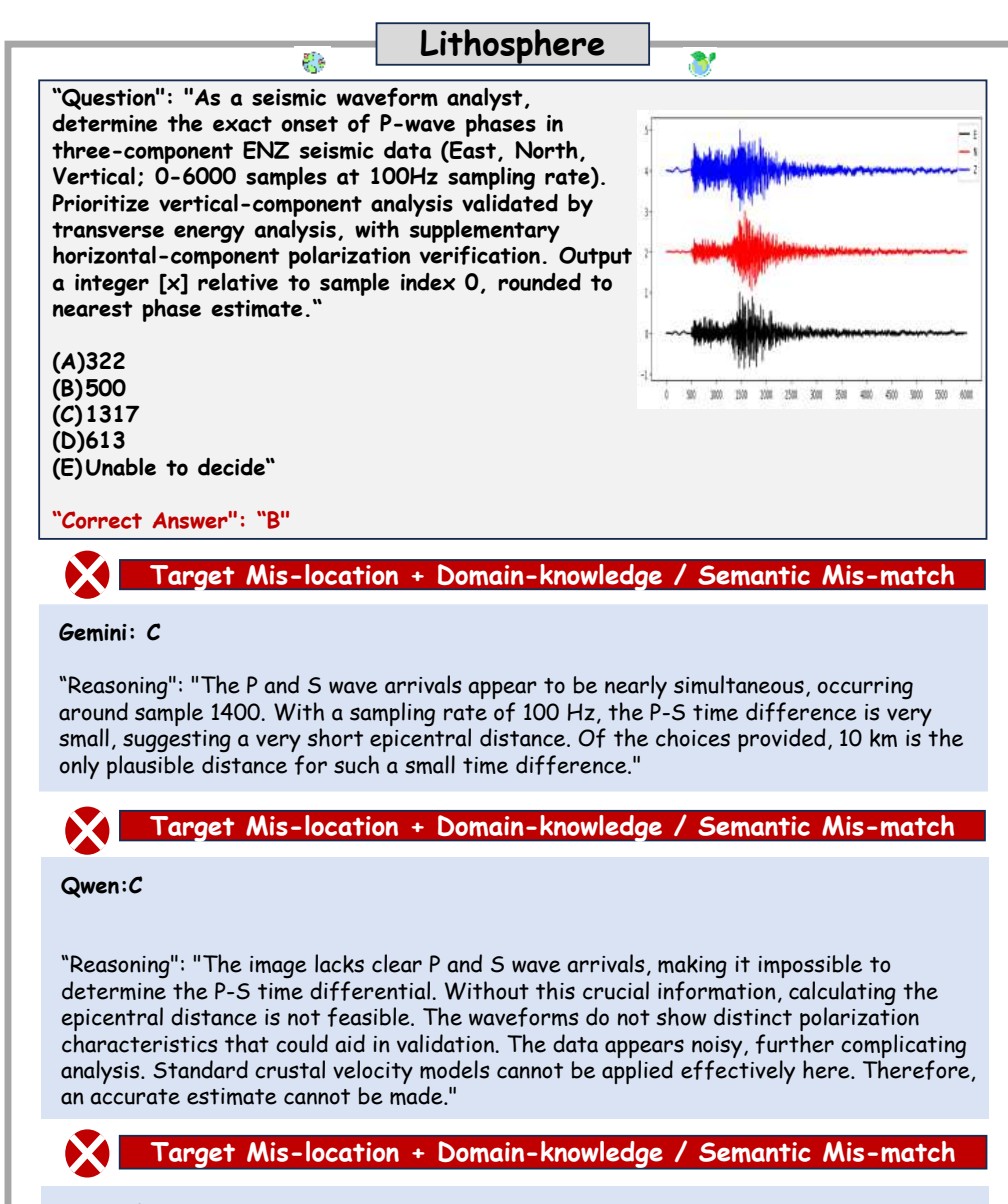

"Question": "As a seismic waveform analyst, determine the exact onset of P-wave phases in three-component ENZ seismic data (East, North, Vertical; 0-6000 samples at 100Hz sampling rate). Prioritize vertical-component analysis validated by transverse energy analysis, with supplementary horizontal-component polarization verification. Output a integer [x] relative to sample index 0, rounded to nearest phase estimate."

(A)322
(B)500
(C)1317
(D)613
(E)Unable to decide"

"Correct Answer": "B"

❌ **Target Mis-location + Domain-knowledge / Semantic Mis-match**

**Gemini: C**

"Reasoning": "The P and S wave arrivals appear to be nearly simultaneous, occurring around sample 1400. With a sampling rate of 100 Hz, the P-S time difference is very small, suggesting a very short epicentral distance. Of the choices provided, 10 km is the only plausible distance for such a small time difference."

❌ **Target Mis-location + Domain-knowledge / Semantic Mis-match**

**Qwen:C**

"Reasoning": "The image lacks clear P and S wave arrivals, making it impossible to determine the P-S time differential. Without this crucial information, calculating the epicentral distance is not feasible. The waveforms do not show distinct polarization characteristics that could aid in validation. The data appears noisy, further complicating analysis. Standard crustal velocity models cannot be applied effectively here. Therefore, an accurate estimate cannot be made."

❌ **Target Mis-location + Domain-knowledge / Semantic Mis-match**

**Internvl3:A**

"Reasoning": "The waveform shows distinct P-wave arrivals in the East (E) and North (N) components, with a noticeable delay between them. The vertical (Z) component shows a clear P-wave arrival as well. By analyzing the time difference between the P-wave arrivals in the E and N components, we can estimate the epicentral distance. Assuming standard crustal velocities, the time difference corresponds to a distance of approximately 28 kilometers. This is consistent with the choices provided, making option (B) the best estimate."

Figure 8: A sample case of P-wave phase picking subtask.

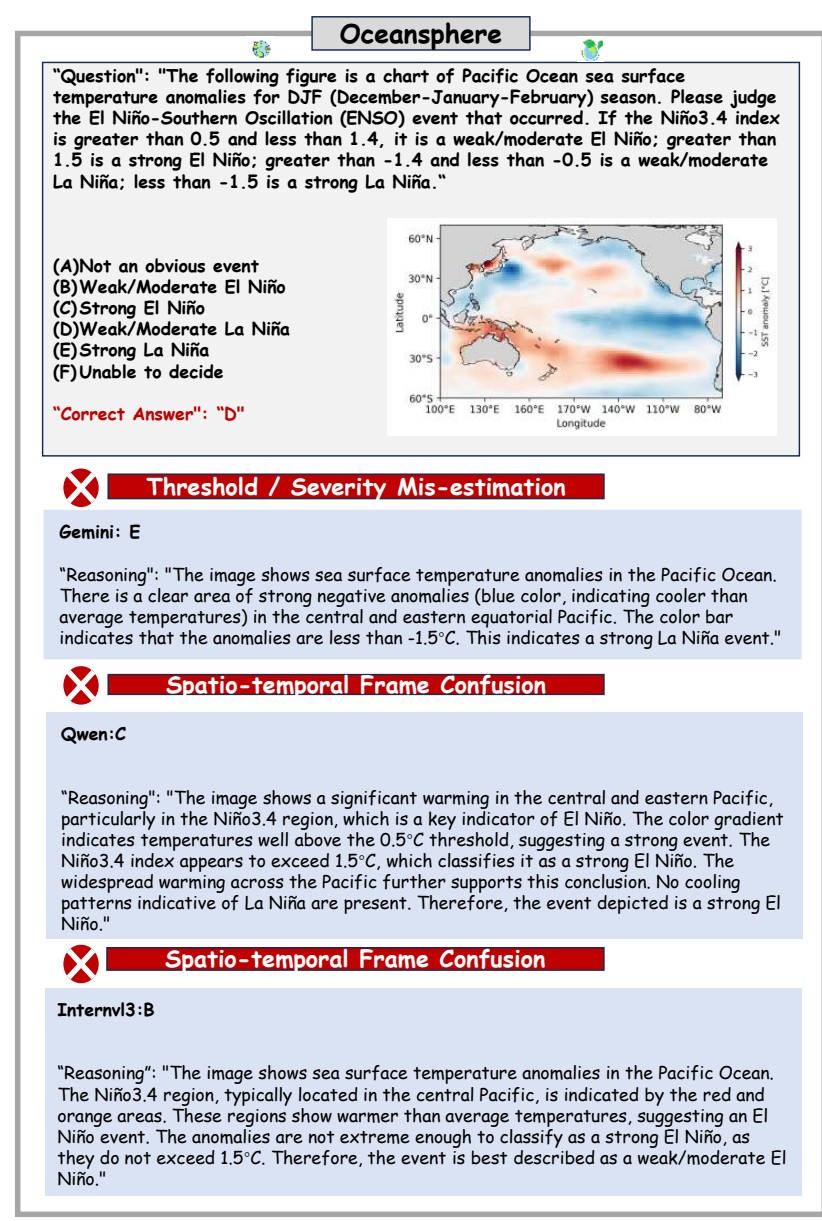

Figure 9: A sample case of ENSO Identification subtask.

## Oceansphere

"Question": "The following image is an RGB image derived using the visible bands of satellite data. Please determine whether there is marine fog in the area shown in the image."

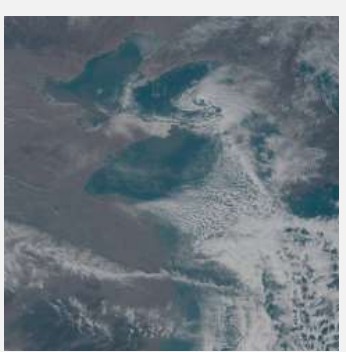

(A)Yes, there is marine fog
(B)No, there is no marine fog
(C)Unable to decide

"Correct Answer": "B"

### ❌ Image-feature Misinterpretation

**Gemini: A**

"Reasoning": "The image shows extensive areas of low-lying, featureless white or greyish features over the water. These features are consistent with marine fog or low stratus clouds. The texture and spatial distribution suggest a widespread phenomenon, likely fog."

### ❌ Image-feature Misinterpretation

**Qwen: A**

"Reasoning": "The image shows a large body of water with a significant area covered by a uniform layer of white clouds, which is characteristic of marine fog. The fog appears to be hugging the coastline and extending over the water, a typical pattern for marine fog. The surrounding landmasses are visible with clear distinctions, indicating that the fog is localized over the water. The texture of the cloud layer is smooth and consistent, further suggesting the presence of marine fog. The absence of sharp cloud edges and the gradual transition from the fog to clearer areas support this conclusion."

### ❌ Image-feature Misinterpretation

**Internvl3:A**

"Reasoning": "The image shows a satellite view with visible bands that highlight different features. The presence of marine fog can be inferred from the areas where the water appears lighter and more diffuse compared to the surrounding darker waters. This lighter appearance is characteristic of fog, which scatters light and reduces visibility. Additionally, the fog appears to be concentrated along the coastline, which is typical for marine fog formation."

Figure 10: A sample case of Marine Fog Detection subtask.

## Human-activities sphere

"Question": "Recognize the category of objects from satellite and aerial images given thebounding boxes for referring objects. Bounding box in the format (xmin, ymin,xmax, ymax), where the top left corner is (x_min, y_min) and the bottom-right corner is (x_max, y_max). The resolution of satellite image is 600 x 600. Bounding box:[<132><142><186><187>]"

(A)flat_roof
(B)gable_roof
(C)row roof shed
(D)Other
(E)Unable to decide

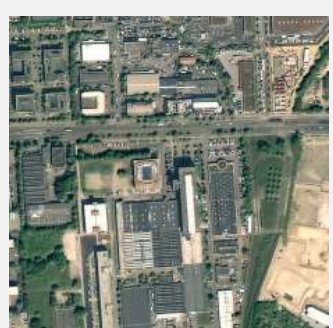

"Correct Answer": "D"

### ❌ Image-feature Misinterpretation

**Gemini: A**

"Reasoning": "The bounding box [132, 142, 186, 187] refers to a building with a flat roof. The roof appears to be uniformly flat without any visible gables or other roof structures."

### ❌ Image-feature Misinterpretation

**Qwen: A**

"Reasoning": "The image shows a large industrial or commercial area with numerous buildings. The roofs of these buildings are predominantly flat, which is a common feature in industrial and commercial architecture. The flat roofs are visible as large, smooth surfaces with no significant slope or gable structures. The presence of parking lots and the layout of the buildings further suggest that these are likely warehouses or factories, which typically have flat roofs for practical reasons such as ease of construction and maintenance. Therefore, the category of objects in the bounding box is best described as 'flat_roof'."

### ❌ Image-feature Misinterpretation

**Internvl3:A**

"Reasoning": "The bounding box coordinates provided correspond to a flat-roofed structure in the image. The area within the bounding box shows a rectangular shape with a flat surface, typical of flat roofs commonly found in industrial or commercial buildings. The lack of visible sloping edges or peaks further supports this classification."

Figure 11: A sample case of Fine-grained object type recognition subtask.

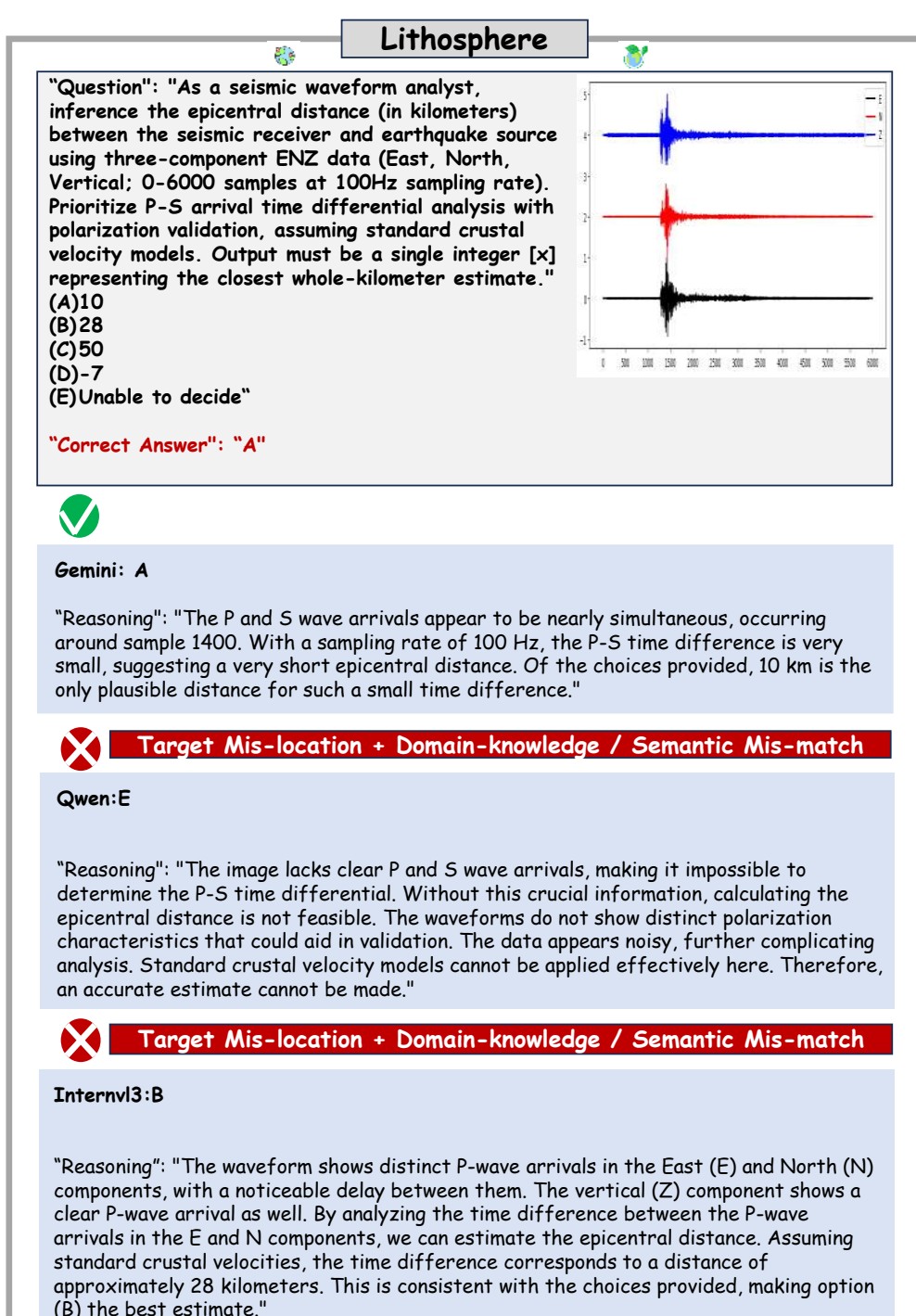

Figure 12: A sample case of earthquake source-receiver distance inference subtask.

## 1.6 Datasheets

In this section, we document essential details about the proposed datasets and benchmarks following the CVPR Dataset and Benchmark guidelines and the template provided by Gebru *et al.* [4].

### 1.6.1 Motivation

The questions in this section are primarily intended to encourage dataset creators to clearly articulate their reasons for creating the dataset and to promote transparency about funding interests. The latter may be particularly relevant for datasets created for research purposes.

1. *"For what purpose was the dataset created?"*

    **A:** Existing benchmarks for Earth science multimodal learning exhibit critical limitations in systematic coverage of geosystem components and cross-sphere interactions, often constrained to isolated subsystems (only in Human-activities sphere or atmosphere) with limited evaluation dimensions ($\leq 16$ tasks). To address these gaps, we introduce **OmniEarth-Bench**, the first comprehensive multimodal benchmark spanning all six Earth science spheres (atmosphere, lithosphere, Oceansphere, cryosphere, biosphere and Human-activities sphere) and cross-spheres with one hundred expert-curated evaluation dimensions.

2. *"Who created the dataset (e.g., which team, research group) and on behalf of which entity?"*

    **A:** The dataset was created by the following authors:

    - Fengxiang Wang (National University of Defense Technology)
    - Mingshuo Chen (Beijing University of Posts and Telecommunications)
    - Xuming He (ZJU)
    - Yi-Fan Zhang
    - Feng Liu (Shanghai AI Lab)
    - Zijie Guo (Zijie Guo)
    - Zijie Guo (SYSU)
    - Jiong Wang (Shanghai AI Lab)
    - Jingyi Xu (Shanghai AI Lab)
    - Zhangrui Li (Shanghai AI Lab)
    - Fenghua Ling (Shanghai AI Lab)
    - Ben Fei (Shanghai AI Lab)
    - Weijia Li (Weijia Li)
    - Long Lan (National University of Defense Technology)
    - Wenlong Zhang (Shanghai AI Lab)
    - Lei Bai ((Shanghai AI Lab))

3. *"Who funded the creation of the dataset?"*

    **A:** The dataset creation was funded by the affiliations of the authors involved in this work.

### 1.6.2 Composition

Most of the questions in this section are intended to provide dataset consumers with the information they need to make informed decisions about using the dataset for their chosen tasks. Some of the questions are designed to elicit information about compliance with the EU's General Data Protection Regulation (GDPR) or comparable regulations in other jurisdictions. Questions that apply only to datasets that relate to people are grouped together at the end of the section. We recommend taking a broad interpretation of whether a dataset relates to people. For example, any dataset containing text that was written by people relates to people.

1. *"What do the instances that comprise our datasets represent (e.g., documents, photos, people, countries)?"*

    **A:** Our Benchmark comprises not only publicly available open-source datasets but also a significant portion of data manually extracted by experts from satellite imagery and raw observational sources. For example, Vegetation Monitoring uses satellite imagery from MODIS and expert-curated data from the Global Land Surface Satellite (GLASS), including

Leaf Area Index, Fractional Vegetation Cover and Peak Vegetation Coverage Area. All datasets utilized in OmniEarth-Bench are publicly accessible and nonprofit.

2. *"How many instances are there in total (of each type, if appropriate)?"*

   **A:** OmniEarth-Bench consists of seven spheres, with a total of 24,109 annotated data instances. Among them, the cross-sphere category contains 3,456 instances, the Human-activities sphere has 10,002 instances, the Biosphere has 6,221 instances, the Atmosphere has 6,395 instances, the Lithosphere has 2,131 instances, the Oceansphere has 1,374 instances, and the Cryosphere has 230 instances.

3. *"Does the dataset contain all possible instances or is it a sample (not necessarily random) of instances from a larger set?"*

   **A:** The images in OmniEarth-Bench are sourced from existing [5, 6, 7, 8, 9, 10, 11, 12, 13, 14, 15, 16, 17, 18, 19, 20, 21, 22, 23, 24, 25, 26, 27, 28, 29, 30, 31, 32, 33, 34, 35] datasets, but all textual annotations were independently created by us.

4. *"Is there a label or target associated with each instance?"*

   **A:** Yes, each instance has been annotated and quality-checked by specialized experts.

5. *"Is any information missing from individual instances?"*

   **A:** No, each individual instance is complete.

6. *"Are relationships between individual instances made explicit (e.g., users' movie ratings, social network links)?"*

   **A:** Yes, the relationship between individual instances is explicit.

7. *"Are there recommended data splits (e.g., training, development/validation, testing)?"*

   **A:** The dataset is designed to evaluate the performance of MLLMs across various Earth spheres, so we recommend using it in its entirety as a test set.

8. *"Is the dataset self-contained, or does it link to or otherwise rely on external resources (e.g., websites, tweets, other datasets)?"*

   **A:** OmniEarth-Bench is self-contained and will be open-sourced on platforms like Hugging Face, integrated into evaluation tools such as LLMs-Eval [36, 37] for easy use.

9. *"Does the dataset contain data that might be considered confidential (e.g., data that is protected by legal privilege or by doctor–patient confidentiality, data that includes the content of individuals' non-public communications)?"*

   **A:** No, all data are clearly licensed.

10. *"Does the dataset contain data that, if viewed directly, might be offensive, insulting, threatening, or might otherwise cause anxiety?"*

    **A:** No, OmniEarth-Bench does not contain any data with negative information.

### 1.6.3 Collection Process

In addition to the goals outlined in the previous section, the questions in this section are designed to elicit information that may help researchers and practitioners create alternative datasets with similar characteristics. Again, questions that apply only to datasets that relate to people are grouped together at the end of the section.

1. *"How was the data associated with each instance acquired?"*

   **A:** The images in OmniEarth-Bench are sourced from existing [5, 6, 7, 8, 9, 10, 11, 12, 13, 14, 15, 16, 17, 18, 19, 20, 21, 22, 23, 24, 25, 26, 27, 28, 29, 30, 31, 32, 33, 34, 35] datasets, all textual annotations were independently created by experts.

2. *"What mechanisms or procedures were used to collect the data (e.g., hardware apparatuses or sensors, manual human curation, software programs, software APIs)?"*

   **A:** Our Benchmark comprises not only publicly available open-source datasets but also a significant portion of data manually extracted by experts from satellite imagery and raw observational sources. For example, Vegetation Monitoring uses satellite imagery from MODIS and expert-curated data from the Global Land Surface Satellite (GLASS), including Leaf Area Index, Fractional Vegetation Cover and Peak Vegetation Coverage Area. Moreover, for the Eddy data in oceansphere, the chlorophyll (CHL) data used in this study

were obtained by applying the OCI empirical algorithm to Level-2 data acquired by the Geostationary Ocean Color Imager I (GOCI) aboard the Oceanography and Meteorology Satellite (COMS). After careful selection and integration, we compiled a comprehensive dataset covering 33 different data modalities across all Earth spheres. Tab.**??** is a summary of the data sources used for each Earth sphere.

3. *"If the dataset is a sample from a larger set, what was the sampling strategy (e.g., deterministic, probabilistic with specific sampling probabilities)?"*

   **A:** Please refer to the details listed in the main text Section 3.1.

### 1.6.4 Preprocessing, Cleaning, and Labeling

The questions in this section are intended to provide dataset consumers with the information they need to determine whether the "raw" data has been processed in ways that are compatible with their chosen tasks. For example, text that has been converted into a "bag-of-words" is not suitable for tasks involving word order.

1. *"Was any preprocessing/cleaning/labeling of the data done (e.g., discretization or bucketing, tokenization, part-of-speech tagging, SIFT feature extraction, removal of instances, processing of missing values)?"*

   **A:** Yes. During image collection, we prioritized selecting valuable satellite images for annotation. For linguistic annotation, three Level-3 subtasks—Regional Land Use Classification, Regional Counting, and Regional Counting with Change Detection—were marked with red circles. This method, mimicking human interaction, was essential for providing clear, fine-grained region-level analysis on ultra-high-resolution images.

2. *"Was the 'raw' data saved in addition to the preprocessed/cleaned/labeled data (e.g., to support unanticipated future uses)?"*

   **A:** Yes, raw data is accessible.

3. *"Is the software that was used to preprocess/clean/label the data available?"*

   **A:** Yes, the necessary software used to preprocess and clean the data is publicly available.

### 1.6.5 Uses

The questions in this section are intended to encourage dataset creators to reflect on tasks for which the dataset should and should not be used. By explicitly highlighting these tasks, dataset creators can help dataset consumers make informed decisions, thereby avoiding potential risks or harms.

1. *"Has the dataset been used for any tasks already?"*

   **A:** No.

2. *"Is there a repository that links to any or all papers or systems that use the dataset?"*

   **A:** Yes, we will provide such links in the GitHub and the Huggingface repository.

3. *"What (other) tasks could the dataset be used for?"*

   **A:** OmniEarth-Bench is suitable for various tasks across other Earth spheres. It covers 103 subtasks spanning six major Earth spheres plus Cross-sphere, and is capable of handling various other tasks.

4. *"Is there anything about the composition of the dataset or the way it was collected and preprocessed/cleaned/labeled that might impact future uses?"*

   **A:** No.

5. *"Are there tasks for which the dataset should not be used?"*

   **A:** N/A.

### 1.6.6 Distribution

Dataset creators should provide answers to these questions prior to distributing the dataset either internally within the entity on behalf of which the dataset was created or externally to third parties.

1. *"Will the dataset be distributed to third parties outside of the entity (e.g., company, institution, organization) on behalf of which the dataset was created?"*

   **A:** No. The datasets will be made publicly accessible to the research community.

2. *"How will the dataset be distributed (e.g., tarball on website, API, GitHub)?"*

   **A:** We will provide OmniEarth-Bench in the GitHub and the Huggingface repository.

3. *"When will the dataset be distributed?"*

   **A:** We will create a repository to release the data once the paper is officially published, ensuring compliance with the anonymity principle.

4. *"Will the dataset be distributed under a copyright or other intellectual property (IP) license, and/or under applicable terms of use (ToU)?"*

   **A:** Yes, the dataset will be released under the Creative Commons Attribution-NonCommercial-ShareAlike 4.0 International License.

5. *"Have any third parties imposed IP-based or other restrictions on the data associated with the instances?"*

   **A:** No.

6. *"Do any export controls or other regulatory restrictions apply to the dataset or to individual instances?"*

   **A:** No.

### 1.6.7   Maintenance

As with the questions in the previous section, dataset creators should provide answers to these questions prior to distributing the dataset. The questions in this section are intended to encourage dataset creators to plan for dataset maintenance and communicate this plan to dataset consumers.

1. *"Who will be supporting/hosting/maintaining the dataset?"*

   **A:** The authors of this work serve to support, host, and maintain the datasets.

2. *"How can the owner/curator/manager of the dataset be contacted (e.g., email address)?"*

   **A:** The curators can be contacted via the email addresses listed on our paper or webpage.

3. *"Is there an erratum?"*

   **A:** There is no explicit erratum; updates and known errors will be specified in future versions.

4. *"Will the dataset be updated (e.g., to correct labeling errors, add new instances, delete instances)?"*

   **A:** Future updates (if any) will be posted on the dataset website.

5. *"Will older versions of the dataset continue to be supported/hosted/maintained?"*

   **A:** Yes. This initial release will be updated in the future, with older versions replaced as new updates are posted.

6. *"If others want to extend/augment/build on/contribute to the dataset, is there a mechanism for them to do so?"*

   **A:** Yes, we will provide detailed instructions for future extensions.

### 1.7   Limitation and Potential Societal Impact

In this section, we discuss the limitations and potential societal impact of this work.

### 1.7.1   Potential Limitations

While **OmniEarth-Bench** provides a comprehensive benchmark for evaluating the perception and reasoning capabilities of MLLMs, there are several limitations to consider:

- **Scope of Sensors:** Although our benchmark includes 29,7791 annotations and 103 subtasks, it may not cover all possible real-world scenarios. There could be additional sensor data, like multispectral data that were not included in this study, potentially limiting the generalizability of our findings.

- **Model and Dataset Diversity:** In this paper, we extensively evaluated general-purpose MLLMs. As new models emerge, their evaluation results will be added to our open-source leaderboard. Additionally, OmniEarth-Bench will also be expanded in dataset size and task diversity.

### 1.7.2 Potential Negative Societal Impact

- **Safety Risks:** OmniEarth-Bench is designed to evaluate the performance of vision-language multimodal models in six spheres and cross-sphere scenarios. However, excessive reliance on evaluation datasets may lead to overconfidence in autonomous systems, such as multimodal large models. It is crucial to implement adequate safety measures and human supervision when deploying these MLLMs to ensure public safety.

- **Environmental Impact:** Training MLLMs on large datasets and evaluating them using OmniEarth-Bench requires a certain amount of computational resources. To facilitate future research, we will maintain a leaderboard of MLLMs, removing the need for repeated evaluations of existing models.