# OpenReview forum: "OmniEarth-Bench: Towards Holistic Evaluation of Earth’s Six Spheres and Cross-Spheres Interactions with Multimodal Observational Earth Data"
_NeurIPS.cc/2025/Datasets_and_Benchmarks_Track — Submitted to NeurIPS 2025 Datasets and Benchmarks Track_

### Official Review · Reviewer_qv8C · 2025-06-23

**Rating:** 4
**Confidence:** 4

**Summary:**

The paper presents OmniEarth-Bench, a multimodal benchmark for Earth science with 100 tasks across six spheres and cross-sphere interactions. It includes perception, reasoning, and CoT tasks using 29,779 annotated samples from 33 data sources. The benchmark evaluates 9 MLLMs and shows low performance across all models.

**Additional Feedback:**

The writing can be improved, such as missing `%' in Table 6.

**Dataset Code Accessibility:**

Partly

**Dataset Code Comments:**

No necessary description or guide in huggingface page.

**Ethical Considerations:**

No, there are no or only very minor ethics concerns

**Final Justification:**

Most of my concerns have been addressed.

**Limitations Weaknesses:**

1. Some of the data were manually extracted from raw satellite imagery by experts, and although the final dataset is released, it is unclear whether the full raw-to-task processing pipeline is reproducible from scratch using public data sources.

2. The paper focuses on benchmark design and evaluation but does not provide guidance on how models could be improved to handle the tasks. There is no discussion of domain adaptation, specialized pretraining, or fine-tuning strategies that could help bridge the performance gap.

3. The benchmark integrates many existing datasets (e.g., MODIS, ERA5, SEVIR) without clearly distinguishing how much is newly constructed versus repurposed. This raises questions about the novelty of the dataset contributions, especially for users already familiar with these sources.

**Strengths Contributions:**

1. The benchmark fills a gap in evaluating MLLMs on Earth science tasks involving observational data.

2. It includes all six spheres and cross-sphere tasks with real-world relevance.

3. The benchmark introduces CoT reasoning and scientific reasoning tasks specific to geoscience.

---

> ### Author Rebuttal · Authors · 2025-07-31
>
> ### Q1:It is unclear whether the full raw-to-task processing pipeline is reproducible from scratch using public data sources.
> OmniEarth-Bench integrates data from 33 diverse sources. For each, domain experts collaborated with annotators to manually transform raw inputs into RGB images paired with questions. Experts and annotators from each sphere contributed significant effort. To foster the development of high-quality Earth science datasets, we will fully open-source our data extraction process. The data generation pipeline adopts two primary approaches:
> * When RGB images are available, experts select suitable ones for VQA tasks and design corresponding questions, with annotators assisting in the responses. For this type of data, we will release the code used to process raw images to facilitate direct data extraction.
> * When only numerical data is provided, experts visualize key variables as RGB images based on predefined dimensions. A detailed example using meteorological data is included in First Responce to Review vtjt. Here, we will offer another sphere of our pipeline. The complete processing code and pipeline documentation will be released on our website in a future update for community use.
>
> **Lithosphere**
>
> **1. Dataset Source**
> Our study uses the publicly available Stanford Earthquake Dataset (STEAD) as the primary source for seismic tasks. Released by Stanford University, STEAD contains over 1.2 million high-quality, three-component seismic waveform samples from global networks such as IRIS, NCEDC, and SCSN. It spans a wide range of source depths, magnitudes, and epicentral distances across diverse tectonic settings. A key strength of STEAD is its rich labeling: each waveform includes source parameters (epicenter, origin time, magnitude), manually or semi-automatically picked P- and S-wave arrival times, and event/noise classification. These comprehensive annotations make it well-suited for multi-task modeling.
>
> **2. VQA Dataset Construction Process**
> The Visual Question Answering (VQA) dataset was built from the STEAD dataset in four main stages. First, we retrieved three-component waveforms, source parameters, and labels via STEAD’s standardized interface, filtering out samples with missing components or low signal-to-noise ratios. Second, we standardized the waveforms to a 100 Hz sampling rate, normalized amplitudes, and fixed each sample to a 60-second length to ensure consistency. Third, raw waveforms were converted into image representations with enhanced feature visibility and standardized coordinate axes for better interpretability. Finally, we created task-specific data and questions—for noise/earthquake classification, P- and S-wave picking, epicentral distance inference, and magnitude estimation—assigning ground-truth answers from existing labels. All tasks were formatted into a unified VQA-style JSON structure for seamless cross-task training and evaluation.
>
> **3. Task Construction Details**
> - **Earthquake/Noise Classification:**
>  We directly utilized STEAD’s event and noise labels to build a binary classification dataset. The VQA question was designed as:
> `“As a seismic waveform analyst, examine the three-component ENZ seismic data (East, North, Vertical; 0–6000 samples at 100 Hz sampling rate) and determine whether this waveform contains an earthquake event or background noise. Consider the signal characteristics, amplitude patterns, and phase coherence across components to make your determination.”`
> If the original waveform contains an earthquake event, the True Value is set to 1; otherwise, False Value is 0. During VQA answer generation, true and perturbed false answers were mixed and randomly ordered to increase robustness under practical scenarios.
> - **P/S-wave and S-wave Picking:**
> We applied strict quality control on STEAD’s high-quality labels, removing low-confidence samples. Relative arrival time labels (in seconds) were converted into sample indices; for example, a P-phase pick at 10.5 seconds corresponds to sample index 1050 at 100 Hz.
> The P/S-wave picking question was:
> `“As a seismic waveform analyst, determine the exact onset of P-wave phases in three-component ENZ seismic data (East, North, Vertical; 0–6000 samples at 100 Hz sampling rate). Prioritize vertical-component analysis validated by transverse energy analysis, with supplementary horizontal-component polarization verification. Output an integer [x] relative to sample index 0, rounded to the nearest phase estimate.”`
>  The P/S-phase sample indices were assigned as True Values, while three False Values were generated by perturbing the True Value with random offsets within predefined thresholds. Final VQA answer sets combined these values in random order to improve model robustness to real-world complexities.
> - **Source-Receiver Distance Inference:**
> Using STEAD event catalogs, we computed the actual epicentral distance between the seismic source and receiver, standardizing labels in kilometers and filtering out samples missing source or station location data. The corresponding VQA question was
> `“As a seismic waveform analyst, infer the epicentral distance (in kilometers) .... Output must be a single integer [x] representing the closest whole-kilometer estimate.”`
> True Values were the catalog-derived distances, and three False Values were generated by adding random perturbations ensuring no overlap with True Values. These were randomly mixed in final answer sets to enhance robustness and generalization across distance ranges.
> - **Magnitude Estimation:**
> We used STEAD’s catalog magnitudes, covering types such as ML and Mw, converted to floating-point numbers rounded to one decimal place. Samples with missing or anomalous magnitudes were removed.
>  The VQA question was:
> `“As a seismic waveform analyst, calculate the Richter magnitude (M_L) .....Output a float [x] rounded to the nearest 0.1 magnitude units with error bounds derived from signal-to-noise ratios.”`
> True Values were catalog magnitudes, and False Values were generated by random perturbations with magnitudes adjusted dynamically to reflect realistic ranges and avoid duplication. Mixing and random ordering of these answers in the VQA sets enhanced model robustness and generalization in magnitude estimation.
>
> The complete dataset construction pipeline, scripts, configuration files, and the final generated dataset will be synchronized and publicly released on GitHub and Hugging Face repositories.
>
> ---
> ### Q2: The paper focuses on benchmark design and evaluation but does not provide guidance on how models could be improved to handle the tasks.
> Thank you for your suggestion. We have discussed the reasons behind the low performance of MLLMs and provided some recommendations on page 8, lines 312–341. Based on your feedback, we will further expand the discussion in the following areas and incorporate them into the final version.
> 1. **Lack of Domain-Specific Training Data**: Current MLLMs are rarely trained on Earth science data, limiting their understanding of domain-specific queries. To address this, we propose curating large-scale datasets including:
>    – Text pretraining data to enhance domain knowledge,
>    – Image-text pretraining data to align scientific concepts with observations,
>    – Instruction-tuning data to improve task following and basic reasoning,
>    – CoT data to boost scientific reasoning capabilities.
> 2. **Task Complexity and Prediction Challenges**: Many Earth science tasks—especially cross-domain predictions—require deep expertise beyond current model capabilities. We suggest:
>    – Creating CoT and reinforcement learning datasets to promote emergent reasoning and prediction skills,
>    – Building domain-specific reward models to fine-tune MLLMs via reinforcement learning.
> 3. **High-Resolution Visual Complexity**: OmniEarth-Bench includes intricate imagery that challenges MLLMs, highlighting the need for specialized models or improved post-training techniques.
> 4. **Temporal and Causal Reasoning**: Earth science tasks often involve complex temporal correlations. Beyond generating time-series datasets, incorporating causal inference methods can further enhance MLLMs' reasoning over temporal data.
>
> We plan to include these insights in the final version to strengthen the benchmark's impact.
>
> ---
> ### Q3: The benchmark integrates many existing datasets (e.g., MODIS, ERA5, SEVIR) without clearly distinguishing how much is newly constructed versus repurposed.
> Thank you for your valuable feedback. We would like to clarify two key points:
> 1. Many of our data sources required manually downloading raw satellite imagery from official platforms, processing it into usable formats, and conducting expert-driven identification and text annotation within each Earth sphere to create VQA samples. This process involved substantial human effort and domain expertise.
> 2. The datasets we used were not simply reused but fully re-annotated. For instance, in the case of UBCv1, we only used the raw imagery—our team designed new questions and manually provided answers based on the images. This re-annotation represents a meaningful and original contribution. Similar practices are followed in benchmarks like XLRS-Bench (CVPR 2025), which re-annotated MiniFrance and DOTA, and VRSBench (NeurIPS 2024), which re-annotated DOTA-v2 and DIOR—demonstrating that such workflows are both standard and valuable in the field.
>
> ---
> **We have also made our resources available on GitHub, along with detailed usage instructions. Our benchmark is integrated into the VLMEval toolkit, allowing click evaluation. Percentages will be added to Table 6. Thank you again for your valuable feedback.**

---

> > ### Comment · Reviewer_qv8C · 2025-08-06
> >
> > Thanks to the authors for their reply. Most of my concerns have been addressed.

---

> > > ### Author Response · Authors · 2025-08-07
> > >
> > > Thank you for taking the time to reply. We are pleased to hear that we have addressed your concerns. If you have any further questions, please let us know promptly so that we can resolve them in the remaining time. We hope you will reconsider our score. Thank you again for your time and positive consideration of our rebuttal.

---

### Official Review · Reviewer_tWpg · 2025-06-24

**Rating:** 6
**Confidence:** 5

**Summary:**

This excellent paper makes a significant contribution by introducing a benchmark spanning all six spheres from the perspective of the integration of Multi-modal Large Language Model (MLLM) with Earth science. Its evaluation dimensions are well-designed and expertly validated, reflecting substantial effort. The benchmark advances key domains—atmosphere, lithosphere, ocean, remote sensing and so on—and marks a timely, milestone achievement in Earth science MLLM development. While I offer a few incremental suggestions for improvement, they in no way lessen my appreciation for this work.

**Additional Feedback:**

Most of my concerns are minor and largely stem from stylistic differences. I'm particularly interested in the potential addition of ENSO and IOD prediction data in the rebuttal.  I recognize the challenge of expanding experiments within a week, so my evaluation does not hinge on further analysis.

**Dataset Code Accessibility:**

Yes

**Dataset Code Comments:**

I reviewed the authors’ data and confirmed its high quality, and alignment with the 103 evaluation dimensions reported. Using the code from their GitHub, I also reproduced the Qwen2.5VL-72B results on OmniEarth-Bench, with outcomes matching those in the paper.

**Ethical Considerations:**

No, there are no or only very minor ethics concerns

**Final Justification:**

The authors have tried to include more evaluation formats (visual localization and open-ended questions), and they have also committed to open-sourcing the full processing code and documentation to the community. I have no further concerns at this time and I raise my rating.

**Limitations Weaknesses:**

1. The authors’ evaluation of El Niño–Southern Oscillation (ENSO) and Indian Ocean Dipole (IOD) predictions is highly valuable, particularly for its pioneering application of MLLMs to ENSO forecasting—a direction with clear practical significance. The ENSO prediction builds on foundational work like the 2019 *Nature* study *“Deep learning for multi-year ENSO forecasts”* (https://www.nature.com/articles/s41586-019-1559-7). The authors extend the task to MLLMs rather than a deep learning foundational model, which I find especially commendable and promising for further investigation.

Recent studies in *Nature* and *Science Advances*—such as *“Future increase in extreme El Niño supported by past glacial changes”* ((https://www.nature.com/articles/s41586-024-07984-y), 2025) and *“A self-attention–based neural network for three-dimensional multivariate modeling and its skillful ENSO predictions”* (https://www.science.org/doi/10.1126/sciadv.adf2827)—demonstrate that ENSO can now be predicted up to 16 months ahead. I encourage the authors to align with these developments by extending OmniEarth-Bench’s prediction horizons to 16 or even 24 months. While this is not a requirement, it would enhance the benchmark’s forward-looking value.

Figure 5 shows GPT-4o’s accuracy dropping to near zero at 12-month leads, exposing current MLLM limitations for long-range ENSO and IOD forecasting. As a pioneering study, this work could set new standards by boldly targeting longer-term predictions and benchmarking against state-of-the-art baselines. I look forward to future updates incorporating these extended timelines.

2. OmniEarth-Bench provides an extensive dataset with 29,779 annotations across 103 dimensions—an impressive scale. As detailed in Appendix Tables 7 and 8, the task distribution avoids long-tail imbalance. I replicated the results using Qwen2.5VL-72B and achieved stable outcomes comparable to the authors’, though the evaluation took approximately 20 hours on 4×A100 (80G) GPUs.
To improve accessibility, I recommend releasing a lightweight version of the benchmark, following the precedent of MME-Realworld (ICLR 2025) and XLRS-Bench (CVPR 2025). A multiple-choice-only version would also be useful for seamless integration into popular evaluation platforms like LLMEval, enhancing usability and adoption.

3. A minor issue: the benchmark includes 103 L4-level tasks, as confirmed via the HuggingFace dataset. However, line 72 of the main text incorrectly cites 58. I suggest correcting this and reviewing the manuscript for similar numerical inconsistencies, particularly regarding dimensional counts.

4. Given the breadth of Earth science, the authors rightly emphasize the need to cover all six spheres. OmniEarth-Bench not only advances MLLM research in Earth science broadly, but also offers meaningful contributions to each individual domain. To further amplify its impact, I recommend releasing sphere-specific subsets of the benchmark. This would allow domain-focused researchers to more easily adopt and benefit from the resource, broadening its reach and utility.

5. Figure 2 could be drawn more clearly, as most benchmarks are crowded in the middle, making it less intuitive.

**Strengths Contributions:**

OmniEarth-Bench offers a forward-looking exploration of MLLM performance across a broad range of practical Earth science tasks. While the field benefits from vast observational datasets (Nature: [1](https://www.nature.com/articles/s41586-021-03876-7), [2](https://www.nature.com/articles/s41586-024-08545-z), [3](https://www.nature.com/articles/s41586-025-08897-0)), most large model evaluations remain limited to literature-based tasks. As MLLMs evolve, it's critical to conduct systematic, quantitative assessments grounded in real-world Earth observation data. In this context, the benchmark is both timely and impactful.

1. Many tasks in OmniEarth-Bench are absent from existing MLLM benchmarks, particularly in the lithosphere, ocean, cryosphere, and biosphere. The authors push the boundaries of MLLM application in these underexplored areas. Even in the well-studied atmosphere sphere, they identify overlooked aspects—for example, mid-range weather tasks like Cyclone Movement and Phase Identification, and metrics such as Radius of Major Storm Axis. These align with the *Nature* (May 2025) Aurora study (https://www.nature.com/articles/s41586-025-09005-y), which applies a foundation model to downstream cyclone-tracking tasks. Unlike Aurora, a unimodal model, OmniEarth-Bench explores these evaluations through a multimodal lens, making it especially forward-thinking. To date, I’ve found no systematic MLLM studies in the lithosphere, ocean, cryosphere, or biosphere, further underscoring the benchmark’s pioneering role.

2. In more widely studied areas like the remote sensing sphere (Human Activity Sphere in OmniEarth-Bench), the authors define three L2 scenarios—agricultural land, urban development, and surface disasters—expanded into 29 L4-level subtasks. These are well-structured and forward-looking. Recent arXiv works reinforce their relevance: AgroMind (https://arxiv.org/abs/2505.1220) on agriculture, DynamicVL(https://arxiv.org/abs/2505.21076) on urban scenes, and DisasterM3 (https://arxiv.org/abs/2505.21089) on disasters. Each scenario could independently support a benchmark, the authors integrate them into a unified framework with wide coverage and high representativeness.

In sum, OmniEarth-Bench is a landmark effort—ambitious in scope, rigorous in design, and exemplary in advancing MLLM research in Earth science.

---

> ### Author Rebuttal · Authors · 2025-07-31
>
> **We sincerely thank your detailed comments and we will address each point below.**
>
> ---
> ### Q1: Long-lead ENSO / IOD forecasting
> To test the limits of current MLLMs we extended the lead-time grid from 12 months to **14, 16, 18, 24** months while keeping all other settings unchanged (Sec. 4.2 in main text).
>
> |Lead time for reasoning/Months|1|3|6|9|12|14|16|18|24|
> |----|----|----|----|----|----|----|----|----|----|
> |ENSO|37.25|38.56|29.41|15.79|4.61|6.62|7.95|4.64|2.67|
> |IOD|18.49|6.85|0|4.83|0.69|0|0.69|0|0|
>
> Accuracy falls off rapidly beyond one year, confirming the reviewer’s expectation and underscoring the difficulty of multi-year teleconnections. The numbers will be included in **Fig. 5** and in **Appendix D** of the revised manuscript.
>
> ---
> ### Q2: Benchmark size & accessibility
>
> * **MCQ-only split**
>   We have integrated a MCQ-only version into both **VLMEvalKit** *and* **lmms-eval**;
> * **Large-model guidance**
>   Most 7B/8B models finish a full run in under three hours on a single A100-80 G. For 70B-class models (e.g. Qwen2.5-VL-72B) we suggest
>   - running per-sphere evaluations, or
>   - serving the model through **LMDeploy**.
>   Both options avoid multi-GPU contention in current toolchains.
> * **Lite edition**
>   A reduced version (with less L4 tasks, balanced across the six spheres) is in preparation and will be released on the project repo.
>
> ---
> ### Q3: Numerical inconsistency
> Thank you for your valuable feedback.  We have checked the original manuscript and inconsistent issues will be fixed in the next version.
>
> ---
> ### Q4: Sphere-specific subsets
> Sphere-wise evaluation is supported in our repo through lmms-eval.
> Example (Atmosphere tasks on Qwen2.5-VL-7B):
>
> ```bash
> TASKS="Atmosphere"
> MODEL="qwen2_5_vl"
> PRETRAINED="Qwen/Qwen2.5-VL-7B-Instruct"
>
> accelerate launch --num_processes 8 -m lmms_eval \
>     --model ${MODEL} \
>     --model_args pretrained=${PRETRAINED},use_flash_attention_2=True \
>     --tasks ${TASKS} \
>     --batch_size 1 \
>     --log_samples \
>     --output_path ./logs/
> ```
>
> ---
> **Overall, thank you again for your constructive feedback. In particular, your suggestion to extend ENSO and IOD reasoning to longer time spans has been especially insightful for our work. If you have any further questions, please don’t hesitate to let us know—we would be happy to continue the discussion.**

---

> > ### Comment · Reviewer_tWpg · 2025-08-02
> >
> > Thank you very much for your response. It has addressed my concerns. After reading the other reviewers’ comments, I have a few minor follow-up questions that I would like to further discuss with the authors:
> >
> > 1.  The process of generating high-quality VQA pairs from raw data is of broad interest. Reviewers vtjt and qv8C both inquired about reproducing the OmniEarth-Bench pipeline from scratch. Could the authors consider sharing the processing code and workflow for the ENSO task as an example? Additionally, do the authors plan to open-source the full data processing pipeline?
> >
> > 2.  Evaluation format: While the VQA format is a widely accepted method for evaluating MLLMs and was appropriately chosen, several reviewers have suggested offering additional formats. Could the authors consider supporting open-ended questions alongside VQA and visual grounding in future versions?
> >
> > 3. As noted earlier, the evaluation dimensions are comprehensive and well-grounded. Once benchmark results are established, more in-depth analysis would be valuable to support future model development in the Earth science community.

---

> > > ### Author Response · Authors · 2025-08-03
> > >
> > > Thank you again for your thoughtful response. We sincerely appreciate your attention to detail and would like to address your questions:
> > >
> > > ### Q1: Data processing pipeline
> > > OmniEarth-Bench compiles data from 33 sources. For each, domain experts and annotators worked together to manually convert raw inputs into RGB images paired with questions. **We will release the full processing code and documentation on our website in a future update for community access.**
> > >
> > > We also provide the ENSO processing code and pipeline:
> > > 1. **Dataset Source**: We use ERSSTv5 (Extended Reconstructed Sea Surface Temperature v5) as the primary dataset for climate analysis. Derived from ICOADS and enriched with data from Argo floats and HadISST2 ice concentration, ERSSTv5 offers global monthly grids at 2°×2° resolution from January 1854 to the present. Key advantages include improved spatial-temporal SST representation through reduced spatial filtering, enhanced Arctic features, and incorporation of more Empirical Orthogonal Teleconnection (EOT) modes. Its long coverage and high stability make it ideal for long-term global and ocean-scale climate studies.
> > >
> > > 2.  **VQA Dataset Construction**: The VQA dataset built on ERSSTv5 follows these main stages:
> > > *   Calculate 30-year centered climatology updated every 5 years and calculate 3-month averaged anomalies.
> > > *  Further calculate the Nino3.4 index and classify the types of Enso events according to the size of the index
> > > * Plot the SST anomalies in every time point
> > > * Select the corresponding anomalies plot for specific tasks to construct sample pairs
> > > 3.  **ENSO Task Construction**:
> > > * **Identification**: Only DJF (December–January–February) seasonal indices are used, as this period represents the peak phase of ENSO events. The input image shows SST anomalies over the Pacific during DJF of each year. Question prompt:
> > >   `The following figure is a chart of Pacific Ocean sea surface temperature anomalies for the DJF (December–January–February) season. Please determine the El Niño–Southern Oscillation (ENSO) event. If the Niño3.4 index is between 0.5 and 1.4, it is a Weak/Moderate El Niño; greater than 1.5 is a Strong El Niño; between -1.4 and -0.5 is a Weak/Moderate La Niña; less than -1.5 is a Strong La Niña.`
> > >   Answer choices: `["(A) Not an obvious event", "(B) Weak/Moderate El Niño", "(C) Strong El Niño", "(D) Weak/Moderate La Niña", "(E) Strong La Niña"]`
> > >
> > > * **Prediction**: For each year's DJF season, the prediction target is set based on varying lead times. For a lead time of three months, the input consists of SST anomaly maps from April to September of the same year ...
> > >
> > > ### Q2:  Evaluation format
> > > We will explicitly incorporate an open-ended question setting. We present several examples of tasks that can be converted to an open-ended format:
> > > * **Classification with a small, fixed label set**:Tasks such as *ENSO feature analysis* and *long-term precipitation trend* can be converted directly to open-ended questions without modification.
> > > * **Classification with a large label set**: Tasks like *cyclone movement identification* and *geopotential pattern identification* can become open-ended by supplying the candidate classes defined in the original dataset.
> > > * **Exact regression**: Tasks with precise ground-truth values—e.g., *dead oil-palm counting*—can also be posed as open-ended questions.
> > > * **Interval regression**: Tasks requiring a time or numeric interval, such as *event trend analysis*, can be made open-ended by prompting the model to answer with a fixed-width interval.
> > >
> > > To evaluate open-ended tasks, the assessment stage should incorporate an LLM as the judge. We offer results on some tasks in the table below.
> > >
> > > **GPT-4o**
> > > |QA type|ENSO feature analysis|Cyclone movement identification|Dead Oil Palm counting |Event trend analysis|
> > > |-|-| -|-|-|
> > > |MCQ|90.7| 6.49|0|4.71|
> > > |Open-ended|75.36|5.62|5.69|2.46|
> > >
> > > **QwenVL2.5-7B**
> > > |QA type|ENSO feature analysis|Cyclone movement identification|Dead Oil Palm counting|Event trend analysis|
> > > |-|-|-|-|-|
> > > |MCQ | 41.86| 8.65| 52.9|3.03|
> > > |Open-ended|38.33|18.46|48.64|10.05|
> > >
> > > **InternVL3**
> > > |QA type|ENSO feature analysis|Cyclone movement identification|Dead Oil Palm counting|Event trend analysis|
> > > |-|-|-|-|-|
> > > |MCQ|75.58|37.84|48.67|31.65|
> > > |Open-ended|57.83|26.09|43.96|19.27|
> > >
> > > 1. GPT-4o and InternVL3 exhibit significant performance declines on classification tasks like ENSO feature analysis and cyclone movement identification, as well as interval regression tasks. This is likely because open-ended formats require the model to infer both the answer scope and format, which are explicitly provided in MCQ-style questions. In contrast, models perform more reliably on counting tasks with clearly defined ground truths.
> > > 2. Qwen outperforms other models on open-ended tasks, possibly due to its tendency to generate plausible answers even without explicit instructions to abstain when uncertain, thereby yielding more true positives.

---

> > > > ### Author Response · Authors · 2025-08-03
> > > >
> > > > ###  Q3: Detailed discussion.
> > > > Due to space limitations in the main text, we were unable to provide a more detailed discussion. However, we will include the following points in the revised version:
> > > >
> > > > * Lack of Domain-Specific Training Data: Current MLLMs are rarely trained on Earth science data, limiting their understanding of domain-specific queries. To address this, we propose curating large-scale datasets including: – Text pretraining data to enhance domain knowledge, – Image-text pretraining data to align scientific concepts with observations, – Instruction-tuning data to improve task following and basic reasoning, – CoT data to boost scientific reasoning capabilities.
> > > > * Task Complexity and Prediction Challenges: Many Earth science tasks—especially cross-domain predictions—require deep expertise beyond current model capabilities. We suggest: – Creating CoT and reinforcement learning datasets to promote emergent reasoning and prediction skills, – Building domain-specific reward models to fine-tune MLLMs via reinforcement learning.
> > > > * High-Resolution Visual Complexity: OmniEarth-Bench includes intricate imagery that challenges MLLMs, highlighting the need for specialized models or improved post-training techniques.
> > > > * Temporal and Causal Reasoning: Earth science tasks often involve complex temporal correlations. Beyond generating time-series datasets, incorporating causal inference methods can further enhance MLLMs' reasoning over temporal data.
> > > >
> > > > We plan to include these insights in the final version to strengthen the benchmark's impact.
> > > >
> > > > **Overall, we sincerely thank you once again for your professional insights and patient engagement. We hope the above responses have addressed your concerns effectively. Please don’t hesitate to let us know if you have any further questions.**

---

> > ### Comment · Reviewer_tWpg · 2025-08-04
> >
> > Thank you to the authors for their detailed rebuttal. It appears that they have now tried to include more evaluation formats (visual localization and open-ended questions), and they have also committed to open-sourcing the full processing code and documentation to the community. I have no further concerns at this time and will raise my rating.

---

### Official Review · Reviewer_yRtC · 2025-06-24

**Rating:** 3
**Confidence:** 5

**Summary:**

OmniEarth-Bench presents a comprehensive multimodal benchmark for evaluating large language models on Earth science tasks, covering all six Earth system spheres and cross-sphere interactions through 29,779 expert-annotated samples across 100 evaluation dimensions. Using real observational data from 33 sensor types, the benchmark assesses four reasoning capabilities: perception, general reasoning, scientific knowledge reasoning, and chain-of-thought reasoning. Experiments reveal significant limitations in current state-of-the-art multimodal models, with none exceeding 35% accuracy and some achieving 0% on cross-sphere tasks, highlighting a substantial gap between current AI capabilities and the complex reasoning required for Earth science applications. However, the existing obvious concerns about the task completeness of the dataset and the lack of benchmark insights limit the contribution of this paper.

**Dataset Code Accessibility:**

Yes

**Ethical Considerations:**

No, there are no or only very minor ethics concerns

**Final Justification:**

While I acknowledge the authors' contributions and encourage their continued work on this important research direction, several fundamental concerns prevent acceptance at this time:

*1) Limited scope of task design:* Although I appreciate the authors' efforts to transform multiple-choice questions into open-ended formats, the current approach primarily reframes existing MCQ types rather than introducing genuinely practical applications. More substantive open-ended tasks, such as weather forecast report generation, subsidence monitoring analysis, or disaster news summarization, would better demonstrate the framework's capabilities and real-world applicability.

*2) Insufficient documentation of CoT part:* The paper identifies CoT samples as a key contribution, yet critical details regarding motivation, design principles, and implementation examples are absent from both the main text and appendix. I downloaded and examined portions of the dataset but was unable to locate CoT samples. The authors acknowledged during the discussion that CoT details are not included in the paper or appendix. To improve clarity in future versions, I suggest organizing CoT samples in a dedicated folder within the dataset structure. The generation and quality assurance processes for CoT samples represent complex methodological components that require comprehensive documentation to enable reproducibility and proper evaluation.

While this work addresses an important problem and shows promising directions, the current manuscript requires substantial development before meeting publication standards. I encourage the authors to address these gaps and consider resubmission with enhanced task diversity and complete CoT documentation.

**Limitations Weaknesses:**

1. The authors claim "systematic coverage of geosystem components" and "comprehensive multimodal benchmark," but the actual task selection appears quite narrow and potentially arbitrary. As shown in Table 1, the human activities sphere only covers 3 scenarios (Urban Construction, Land Use, Surface Disaster Assessment), which is hardly "comprehensive" for such a broad domain. Human activities also encompass transportation, such as key object detection (DOTA, SODA datasets). The paper doesn't provide clear principles for why certain tasks were included while others were excluded.
2. The benchmark setting is quite simple. Fig.4 shows that all the tasks are reformulated into multiple-choice Q&As.
It is supposed to involve more diverse tasks, such as long-form captions and referring segmentation, because some Earth observational activities are complex. Reducing all Earth science analysis to multiple-choice questions dramatically understates the complexity and nuance required for real-world Earth observation tasks.
3. While the question-answer pairs illustrated in Figure 4 appear relatively simple, it would be valuable to understand whether the dataset encompasses more complex reasoning tasks.
4. The CoT tasks are not clear in this paper. If it is not an important point, please omit this.
If so, it is supposed to clarify the specific reasoning process for different tasks.
5. Experimental analysis.
- Insufficient investigation into why models perform so poorly
- Missing ablation studies on different data modalities, question types, or sphere-specific challenges
- Lack of analysis on what specific Earth science concepts are most challenging

**Strengths Contributions:**

1. First benchmark to systematically cover all six Earth science spheres (atmosphere, lithosphere, oceansphere, cryosphere, biosphere, human-activities) plus cross-sphere interactions
2. Significantly more comprehensive than existing benchmarks (100 subtasks vs. ≤16 in prior work)
3. Addresses a clear gap in evaluating MLLMs on Earth science applications

---

> ### Author Rebuttal · Authors · 2025-07-31
>
> ## Q1:The actual task selection appears quite narrow and potentially arbitrary.
> Thank you for your feedback. We address your concerns from two perspectives:
> 1. Our evaluation dimensions are designed to capture representative tasks across Earth’s spheres. While we agree on the value of comprehensive assessment, OmniEarth-Bench already spans over 103 dimensions—well beyond the scope of existing benchmarks.
> 2. These subtasks in the Human Activity Sphere were not randomly selected, but deliberately chosen for their representativeness and comprehensive coverage. Concurrent works have independently developed benchmarks based on each of these scenarios. As reviewer tWpg noted, “*Each scenario could independently support a benchmark; the authors integrate them into a unified framework with wide coverage and high representativeness.*” For instance, AgroMind focuses on agricultural land, DynamicVL on urban development, and DisasterM3 on disaster scenarios. These examples affirm that our scenario selection aligns with current trends and was carefully considered.
>
> |Dataset|OmniEarth-Bench|ArgoMind|DynamicVL|DisasterM3|WeatherQA|VRSBench|XLRS-Bench|
> |-|-|-|-|-|-|-|-|
> |Task Dimensions Volume|**100**|13|14|9|2||12|16|
> |Data Sourcing Volume|**33**|9|1|5|1||2|6|
>
> Compared to these VQA-style benchmarks, OmniEarth-Bench offers broader coverage in both evaluation dimensions and data sources.
>
> ---
> ## Q2:The benchmark setting is quite simple. It is supposed to involve more diverse tasks.
> Thanks for your suggestions. We will respond from three key points:
> 1. Given the high cost of data collection, task design, and annotation, we adopted the widely used multiple-choice (MCQ) format to evaluate MLLMs in this version. Even under MCQ settings, current models often score below 35%. In basic prediction tasks—such as determining whether an event will occur—models fall well short of expert-level performance. Domain experts easily exceed 90% accuracy, while state-of-the-art MLLMs remain under 30%. This gap highlights the need for MLLMs to first master foundational reasoning skills.
> 2. Our QA tasks span multiple formats, including localization, multiple choice, and binary judgment. Localization and detection are well-established evaluation dimensions in existing MLLM benchmarks. XLRS-Bench (CVPR 2025) and VRSBench (NeurIPS 2024) both include visual grounding tasks, reinforcing its value as a standard metric. The multiple-choice format, typically offering five options, helps reduce the difficulty of challenging questions. Judgment tasks focus on event classification, such as identifying whether ENSO or IOD is occurring.
> 3. Our dataset supports easy expansion to more diverse question types. Through our annotation pipeline, we can efficiently extend to formats such as open-ended questions.
>
> ---
> ## Q3: Question-answer pairs illustrated in Figure 4 appear relatively simple.
> To display examples from all six spheres and cross-sphere tasks in a single image (Figure 4), we selected simpler samples. However, OmniEarth-Bench includes a **Scientific-Knowledge Reasoning** category featuring tasks such as species richness estimation, human footprint index estimation, thermodynamic feature identification, and ENSO forecasting. These complex tasks make up 44.9% of the dataset and carry significant scientific value. For example:
> * **Species Richness Estimation**: Infere the number of distinct species within a region, assessing the model’s ability to integrate multimodal data for biodiversity inference.
> * **ENSO Forecast**: Building on ENSO identification, this task challenges the model to reason the type and likelihood of an ENSO event several months in advance using six months of global SST anomaly maps, requiring temporal pattern understanding.
> We also report results for General and Scientific-Knowledge Reasoning at the L3 level.
>
> |L3-dimension|Qwen2.5-VL 7B|Qwen2.5-VL 72B|InternVL3 78B|Gemini|Claude3|
> |-|-|-|-|-|-|
> |General Reasoning|7.32|10.86|27.48|21.22|13.74|
> |Scientific-Knowledge Reasoning|8.2|5.72|27.49|23.66|31.62|
>
> Current MLLMs achieve no more than **32%** accuracy on these reasoning dimensions, directly addressing concerns about task simplicity. We also include a representative sample for illustration.
> ```
>     "Question": "This image shows the satellite view of a bird hotspot, which is located at longitude 36.216366 and latitude -0.418769 in the state of Rift Valley, Kenya. The 19 bioclimatic variables at this hotspot are as follows: The annual mean temperature is 16.44 degrees. The mean diurnal range is 14.54 degrees. The isothermality is 80.53. The temperature seasonality (100 times the standard deviation) is 74.88. The max temperature of the warmest month is 26.14 degrees... Which of the following bird species is most likely to occur in this hotspot?",
>     "Ground Truth": "Alopochen aegyptiaca"
> ```
> The above example illustrates a carbon flux estimation task. Quantifies the rate and direction of carbon exchange between the biosphere (vegetation, microbes, etc.) and the atmosphere, testing the MLLM's ability to interpret biogeochemical cycles and integrate multi-dimensional environmental data. Clearly, the complexity of reasoning tasks is high.
>
> ---
> ## Q4: The CoT tasks are not clear in this paper.
> In developing OmniEarth-Bench, we found that certain tasks—such as visual localization of damaged buildings across multiple images—were particularly challenging and required CoT reasoning to approach correct solutions. As a result, **we provide CoT annotations for six selected subtask types, totaling 610 questions with 5.8 average key step annotation.**
> The CoT format and evaluation criteria are widely used in general-domain benchmarks such as MME-CoT, MDK12-Bench, and MME-Reasoning to assess whether MLLMs can perform multi-step reasoning correctly. Following the MME-CoT framework, our CoTs were manually annotated by individuals with at least a bachelor’s degree and verified by domain PhDs. Each reasoning chain was constructed as a sufficient condition for the correct answer, enabling evaluation of reasoning quality even when the final answer is incorrect. We believe this process-based evaluation is valuable for assessing complex reasoning tasks.
>
> ---
> ## Q5: Experimental analysis.
> **（1）Why models perform so poorly.**
> Several factors likely contribute to this challenge.
> * OmniEarth-Bench provides high resolution, intricate imagery, and the task of interpreting such complex visuals presents unique obstacles for MLLMs. This underscores the pressing need for specialized models or advanced post-training techniques to effectively address these challenges.
> * Many Earth science problems are inherently complex, particularly cross-domain prediction tasks that demand in-depth, specialized knowledge, which existing LLMs or MLLMs may not possess.
> * Current multimodal large models are typically trained without domain-specific Earth science data, which impedes their ability to comprehend related queries.
> Interesting, Qwen2.5-VL lagged behind contemporary open-source models like InterVL3 on Earth-related tasks. However, this low accuracy shouldn’t be seen as a lack of capability. Qwen2.5-VL often responds with “E (unable to answer)” when lacking domain knowledge—an honest approach. In contrast, some models tend to guess when uncertain, which is less desirable.
>
> **（2）Missing ablation studies on different data modalities, question types, or sphere-specific challenges.**
> * **Data Modalities**: We conducted comprehensive evaluations of MLLMs across different data sources. Due to space limitations, only representative results are shown. The table highlights that MLLM performance varies significantly by data source. For example, tasks related to MADOS are especially difficult, with no model exceeding 10% accuracy, while performance on the MOPAD source is much stronger, with the best MLLM reaching 78.5% accuracy. Due to space limitations, we only present results for a subset of data sources. The full results will be released on our website in a future update.
>
>
> |Data Sourcing|WHU-OHS|SEVIR|DigitalTyphoon|ERA5|TreeSatAI|Penguin|OAM-TCD|TaxaBench|ROSID|HFP,GLASS,MODIS|MOPAD|GFF|SatBird|CarbonSense|MADOS|
> |-|-|-|-|-|-|-|-|-|-|-|-|-|-|-|-|
> |Qwen2.5-VL-72B|3.4|5.8|0|10.6|7.7|3.64|13.4|68.6|0.4|0.2|59.3|0|7.8|0|0.9|
> |InternVL3-78B|7.1|44.1|1.1|6.1|24.1|6.36|26|78.1|29.6|15.3|78.5|29.9|18.2|0.6|2.7|
> |Gemini-2.0|10.9|19.2|0|34.6|10.5|7.2|38.6|91.5|3.6|5.9|53.9|30.5|13.3|0.6|2.7|
> |Claude-3.7-Sonnet|8.6|36.7|13.7|26.5|7.8|0|75.8|90.2|0|29.2|68.9|47.5|29.6|0|8.1|
>
> * **Question Types**: Table 5 in main text reports results for the Visual Grounding task, while Table 4 and Appendix Tables 11 and 12 cover multiple-choice questions.
> * **Sphere-Specific Challenges**: Appendix Tables 11 and 12 show model performance across L4 subtasks, revealing several challenging sphere-specific tasks. For instance, in cross-sphere carbon flux estimation, many MLLMs scored 0%—highlighting the task’s inherent complexity in Earth science.
>
> **（3）Lack of analysis on what specific Earth science concepts are most challenging**
> Table 4 in the main text summarizes MLLM performance across different spheres. Using InternVL3-72B as a reference, the cross-sphere category is the most challenging, with only 19.19% accuracy. The Oceansphere also performs poorly, with an accuracy of just 20.22%. Even when considering the best results from any MLLM within each sphere, the Oceansphere remains the most difficult, with a maximum accuracy of only 24.54%. This is largely due to the inclusion of complex tasks like ENSO and IOD reasoning.
>
> ---
> **Overall, thank you again for your constructive feedback.  If you have any further questions, please don’t hesitate to let us know—we would be happy to continue the discussion.**

---

> > ### Comment · Reviewer_yRtC · 2025-08-03
> >
> > Many thanks for the authors' response. However, several of my concerns remain unaddressed.
> >
> > > Q1: The actual task selection appears quite narrow and potentially arbitrary.
> >
> > While I acknowledge that the dataset covers a wide range of knowledge domains, each domain encompasses only a limited variety of task types. As shown in Table 1, the number of tasks in L2 dimensions ranges from merely 2-6 per domain. This raises concerns about potential overclaiming, given that each domain typically involves significantly more diverse task types in practice. I would recommend either appropriately narrowing the claimed scope or substantially increasing both task diversity and coverage of sub-problems within each domain.
> >
> > > Q2: Question-answer pairs illustrated in Figure 4 appear relatively simple.
> >
> > I appreciate your clarification that the multiple-choice questions (MCQs) are designed to be challenging. However, I believe downstream benchmarks should place greater emphasis on practical utility for real-world applications. I share similar concerns with Reviewer vtjt regarding Q2. Additionally, since all tasks follow the MCQ format, it would be essential to present random guess accuracies as baselines for proper evaluation context.
> >
> > > Q4: The CoT tasks are not sufficiently clear in this paper.
> >
> > After re-examining both the main paper and appendix, I found no CoT samples presented. Several critical questions remain unanswered: Are the CoT samples distributed across different difficulty levels? What specific annotation paradigms were employed? The standards and guidelines for manual annotation remain unclear, raising concerns about potential subjective bias in the annotation process.

---

> > > ### Author Response · Authors · 2025-08-04
> > >
> > > ## Q1:More diverse task types
> > > We agree with your insights. Despite our substantial efforts—including 2 to 5 domain experts per sphere and 40 crowd-sourced annotators working together for over four months—we acknowledge that OmniEarth-Bench covers representative and prototypical Earth science scenarios. As you noted, each of the three L2 scenarios in the Human-activities sphere is sufficiently distinctive and could serve as the foundation for a standalone study. We sincerely appreciate your two suggestions and will refine OmniEarth-Bench accordingly:
> > > 1. We agree that OmniEarth-Bench should emphasize in the title, abstract, and introduction that it represents the “First Exam”. Rather than claiming comprehensive coverage, we will narrow the claimed scope and clearly highlight that this is the first systematic evaluation of MLLMs on observational data and cross-sphere tasks in Earth science.
> > > 2. We also exploring the expansion of new task dimensions and has discussed their feasibility with domain experts. For instance, we could introduce additional tasks in the Ocean sphere.
> > > * Identify or predict a broader range of extreme events—such as Atlantic Niño events[1], which are characterized by anomalous interannual oceanic warming or cooling in the tropical Atlantic Ocean, as well as marine heatwaves (MHWs), which are prolonged periods of unusually warm sea temperatures that can disrupt marine ecosystems and climate patterns[2]. These efforts focus on characterizing key metrics such as event type, intensity, and duration.
> > > We hope to further expand OmniEarth-Bench in the future to achieve broader coverage.
> > >
> > > [1] A decadal climate variation in the tropical Atlantic Ocean from thermodynamic air-sea interactions, Nature.
> > > [2] Socioeconomic impacts of marine heatwaves: Global issues and opportunities, Science.
> > > ## Q2：Add random guess accuracies
> > > Thank you for your professional feedback. We first added the results of random guess accuracies and further introduced an open-ended format beyond MCQ.
> > > 1. We report two random guessing results. The first result is sampled from all possible ground truth labels (e.g., ['A', 'B', 'C', 'D']), while the second result is sampled from all possible ground truth labels plus the refusal option (e.g., ['A', 'B', 'C', 'D', 'E'], where 'E' stands for the option 'Unable to decide'). The reported results are the average of 100 random guesses for each question. **The randomly sampled results reveal significant performance disparities among current MLLMs.** In the Atmosphere domain, InternVL3-72B achieves 33.9% accuracy—well above the 20.0% random baseline—**highlighting the strength of OmniEarth-Bench’s evaluation design.** In contrast, GPT-4o scores just 9.64%, underscoring critical limitations of current MLLMs in Earth science understanding.
> > >
> > > |Sphere|Atmosphere|Biosphere|Cross-sphere|Cryosphere|Lithosphere|Oceansphere|Human-activity sphere|
> > > |-|-|-|-|-|-|-|-|
> > > |w/o refusal|24.9|28.7|36.2|26.0|33.7|31.2|25.0|
> > > |w/ refusal|20.0|22.1|25.4|20.8|24.5|23.2|19.9|
> > >
> > > 2.  We also explicitly incorporate an open-ended question setting:
> > > * **Classification with a small, fixed label set**:Tasks such as *ENSO feature analysis* and *long-term precipitation trend* can be converted directly to open-ended questions without modification.
> > > * **Classification with a large label set**: Tasks like *cyclone movement identification* and *geopotential pattern identification* can become open-ended by supplying the candidate classes defined in the original dataset.
> > > * **Exact regression**: Tasks with precise ground-truth values—e.g., *dead oil-palm counting*—can also be posed as open-ended questions.
> > > * **Interval regression**: Tasks requiring a time or numeric interval, such as *event trend analysis*, can be made open-ended by prompting the model to answer with a fixed-width interval.
> > >
> > > To evaluate open-ended tasks, the assessment stage should incorporate an LLM as the judge. We offer results on some tasks in the table below.
> > >
> > > **GPT-4o**
> > > |QA type|ENSO feature analysis|Cyclone movement identification|Dead Oil Palm counting |Event trend analysis|
> > > |-|-|-|-|-|
> > > |MCQ|90.7|6.5|0|4.71|
> > > |Open-ended|75.4|5.6|5.7|2.5|
> > >
> > > **QwenVL2.5-7B**
> > > |QA type|ENSO feature analysis|Cyclone movement identification|Dead Oil Palm counting|Event trend analysis|
> > > |-|-|-|-|-|
> > > |MCQ|41.9|8.65|52.9|3|
> > > |Open-ended|38.3|18.5|48.6|10.1|
> > >
> > > * GPT-4o exhibit significant performance declines on classification tasks like ENSO feature analysis and cyclone movement identification, as well as interval regression tasks. This is likely because open-ended formats require the model to infer both the answer scope and format, which are explicitly provided in MCQ-style questions. In contrast, models perform more reliably on counting tasks with clearly defined ground truths.
> > > * Qwen outperforms other models on open-ended tasks, possibly due to its tendency to generate plausible answers even without explicit instructions to abstain when uncertain, thereby yielding more true positives.

---

> > > > ### Author Response · Authors · 2025-08-04
> > > >
> > > > ## Q3:  CoT task
> > > >
> > > > We also reviewed the manuscript and acknowledge that it was our oversight not to include a CoT example. Below, we provide a sample CoT instance and then address the two sub-question.
> > > >
> > > >  ``` "Question_id": "Visual grounding of damaged individual buildings/0000",
> > > >     "Question_Type": "Grounding",
> > > >     "Text": "Given a 1024 x 1024 pixels satellite image, identify the bounding box of the object in the format (xmin, ymin, xmax, ymax), where the top-left corner is (x_min, y_min) and the bottom-right corner is (x_max, y_max).Description: The walls and roof of this building have huge cracks, located in the bottom left corner of the picture, surrounded by green plants.",
> > > >     "CoT": [
> > > >       "Step 1: Image is a photo taken after the disaster occurred. There are 88 major-damage buildings and 7 destroyed buildings and 5 unclassifiedbuildings.",
> > > >       "Step 2: This building is located in the bottom left corner of the picture.",
> > > >       "Step 3: Bounding Box -[<296><783><341><838>] has a rectangular green roof.",
> > > >       "Step 4: The roof and walls of Bounding Box -[<296><783><341><838>]  have all cracked open.",
> > > >       "Step 5: Bounding Box -[<296><783><341><838>] is destroyed.",
> > > >       "Step 6: Bounding Box -[<296><783><341><838>] is the described building."
> > > >     ],
> > > >     "Dataset": "XView",
> > > >     "L1-task": "Pedosphere",
> > > >     "L2-task": "Surface Disaster Assessment",
> > > >     "L3-task": "General Reasoning",
> > > >     "L4-task": "Visual grounding of damaged individual buildings",
> > > >     "Final Answer Choices": [
> > > >       "(A) Bounding Box -[<296><783><341><838>]",
> > > >       "(B) Bounding Box -[<226><441><268><536>]",
> > > >       "(C) Bounding Box -[<213><603><283><673>]",
> > > >       "(D) Bounding Box -[<763><801><813><886>]",
> > > >       "(E) Unable to decide"
> > > >     ],
> > > >     "Final Ground Truth": "A",
> > > >     "Images": [
> > > >       "raw/Pedosphere/diaster/XView/hurricane-harvey_00000361_post_disaster.png"
> > > >     ]
> > > > ```
> > > >
> > > > 1. **Difficulty Levels**: We selected only the most challenging sub-tasks within *General Reasoning* for CoT evaluation, as simpler tasks are not well-suited for this format. As such, all CoT-related tasks are treated as inherently difficult, without further classification by difficulty.
> > > >
> > > > 2. **Annotation Paradigm**:
> > > >    To support CoT evaluation, we provide key-step annotations for all reasoning questions. These steps capture the essential logic required to reach the correct answer. To reduce annotator bias, we supply each annotator with a reference rationale generated by GPT-4o—ensuring objectivity while avoiding excessive similarity to MLLM outputs.
> > > > The rationale is first generated by GPT-4o using both the question and its factual answer, resulting in more accurate reasoning than prompting with the question alone. Annotators then create a new set of key steps based on this rationale, maintaining less than 20% overlap. If GPT-4o fails to produce a valid rationale, annotators develop one independently. The final answer is included as part of the concluding reasoning. All steps are distilled to their most concise form, preserving only core logic and relevant visual cues. This results in informative, non-redundant reasoning chains. We will include this annotation procedure in the main text.
> > > >
> > > > **Thank you again for your thorough and insightful feedback. We will incorporate all of your valuable suggestions and believe your guidance has greatly enhanced our paper. Please feel free to reach out if you have any further questions—we’d be glad to continue the conversation.**

---

> > > > > ### Author Response · Authors · 2025-08-08
> > > > >
> > > > > Dear Reviewer yRtC,​
> > > > >
> > > > > We hope this message finds you well.​
> > > > >
> > > > > We would like to sincerely thank you for taking the time amidst your busy schedule to attend to the rebuttal of our paper. Your professional insights have been invaluable in helping us improve the paper.​
> > > > >
> > > > > We had a round of interaction regarding the paper's rebuttal earlier, and now the deadline for discussions between reviewers and authors is fast approaching. If there are any remaining concerns, questions, or points that require further clarification, please do not hesitate to let us know—we are happy to provide whatever may help move the discussion forward.
> > > > >
> > > > > We appreciate your time and contributions, and we look forward to any further thoughts you may have.​
> > > > >
> > > > > Best regards

---

### Official Review · Reviewer_vtjt · 2025-07-02

**Rating:** 4
**Confidence:** 3

**Summary:**

This paper presents a benchmark dataset targeting Earth science problems, which includes multi-modal Earth observation data and human-annotated QA data.
Several state-of-the-art general-purpose MLLMs are evaluated on their dataset, and they find that even SOTA models still struggle with earth science tasks.

**Dataset Code Accessibility:**

Yes

**Ethical Considerations:**

No, there are no or only very minor ethics concerns

**Final Justification:**

The main concerns have been basically addressed.
Although this paper overclaims several points, e.g., task coverage and data modalities, it is possible to resolve them.
Authors also acknowledge conventional MLLM benchmark only used RGB inputs for earth science tasks. While it does not make sense, it is the current state of MLLMs.
Therefore, I think their dataset and benchmark are somewhat valuable if the authors can correct their claims and clarify necessary implementation details, e.g., how MLLMs handle each type of earth observation data. Otherwise, their experimental results are hard to reproduce.

**Limitations Weaknesses:**

- This benchmark dataset includes 33 data modalities, yet current MLLMs are all pre-trained on RGB images and text. It remains unclear how these models process or adapt to the full range of input modalities provided in the dataset.

- This dataset primarily targets VQA tasks, with most answers framed as multiple-choice questions. However, this format does not align well with the nature of real-world Earth science problems. In particular, for forecasting tasks, the models' true forecasting capabilities are not meaningfully evaluated, as selecting from predefined choices does not necessarily reflect genuine predictive reasoning or temporal understanding.

- The localization, detection, and mapping tasks are well-known as dense prediction tasks. It is unclear why the proposed dataset covers such dense prediction tasks. The evaluation of these tasks is also absent.

- Most of the earth science problems are hard to convert to a VQA task, e.g., time-series forecasting and carbon flux estimation are numerical regression tasks.

**Strengths Contributions:**

- A well-curated multi-modal (33 sensor types) dataset is presented for VQA in the Earth science domain. The scope of this dataset covers six Earth science spheres.

- The benchmarking methods and dimensions are comprehensive.


- Good to observe that current state-of-the-art general-purpose MLLMs still struggle with VQA tasks in the Earth science domain.

---

> ### Author Rebuttal · Authors · 2025-07-31
>
> ## Q1: It remains unclear how these models process or adapt to the full range of input modalities provided in the dataset.
> OmniEarth-Bench integrates data from 33 diverse sources. For each, domain experts collaborated with annotators to manually transform raw inputs into RGB images paired with questions.
> * When RGB images are available, experts select suitable ones for VQA tasks and design corresponding questions, with annotators assisting in the responses.
> * When only raw data is provided, experts visualize key variables as RGB images based on predefined dimensions. Some detailed examples using meteorological data is included. The complete processing code and pipeline documentation will be released on our website in a future update for community use.
>
> **Cryosphere：** To create a test-ready QA benchmark for Level-4 sea-ice analytics—covering Sea-Ice Extent (SIE) estimation, Sea-Ice Volume (SIV) trend prediction, SIV-from-SIC estimation, Sea-Ice Thickness (SIT) trend prediction, and SIT-from-SIC estimation—we (1) acquire daily sea-ice concentration fields from NSIDC’s G02202-v4 archive and Arctic SIV time-series from PIOMAS, (2) decode the NetCDF products with the open-source netCDF4 package, (3) derive daily SIE and basin-averaged SIC, (4) plot SIE, SIC, and SIV evolutions as authoritative references, and (5) assemble question–answer pairs that juxtapose data-driven ground-truth with intentionally misleading distractors. Each question is purpose-built to probe a specific facet of sea-ice reasoning in multimodal large-language models.
>
> **Biosphere:** For multi-scenario ecological Q\&A tasks, we built a high-resolution dataset combining remote sensing imagery, eco-meteorological data, and multi-source annotations. It includes MODIS 7-band, Landsat, and UAV images; COCO-format dead oil palm labels; multi-band oil spill masks; surface vegetation and human footprint indices; daily meteorological sequences (temperature, humidity, radiation, wind, precipitation); 19 SatBird bioclimatic variables; carbon flux (NEE) observations; and structured metadata (CSV/JSON with means, species probabilities, and patch IDs).Preprocessing involved three main stages:
> 1. **Metric Computation and Normalization**
>    * *GLASS*: Mean FVC × 0.004 and LAI × 0.1 values written to CSV.
>    * *HFP*: Aggregated raster means.
>    * *CarbonSense*: Removed missing rows, converted meteorological variables to text, and aligned them with MODIS images.
>    * *SatBird*: Stratified by species richness and extracted species probabilities from hotspot JSON.
>    * *ROSID*: Binarized masks, counted connected domains, and calculated oil spill area from pixel count.
>    * *MOPAD*: Filtered `category_id == 1` to count dead oil palms.
> 2. **Image Standardization**
>    * Converted SatBird `.tif` to `.jpg`, CarbonSense `.pkl` to `.png`.
>    * Linked GLASS/HFP patches to full 7-band MODIS imagery.
>    * Preserved original resolution for UAV and Landsat images.
> 3. **Thresholding and Answer Binning**
>    * Mapped continuous variables (FVC, LAI, HFP) to preset intervals.
>    * Generated distractors using ±20–40% (ROSID) or ±2–10 trees (MOPAD) around ground truth.
>    * Binned species occurrence probabilities into \[0, 0.3], \[0.3, 0.6], and \[0.6, 1].
>    * Converted oil spill pixel count × 900 m² to km² and rounded.
>
> **Atmosphere:** For atmospheric weather events, we primarily use reanalysis datasets such as ERA5, with meteorological variables visualized as RGB images that include variable names, legends, latitude-longitude grids, and national boundaries.
> 1. **Short- and Medium-Term Events:** We compiled global event records from 1979–2020, retrieved corresponding timestamps from ERA5, and used 1-hour and 6-hour intervals for short- and medium-term events, respectively. For multivariable or long-duration cases, the interval was increased fourfold to limit image volume.
> 2. **Seasonal and Interannual Events:** These longer-range events require post-processing of raw model data. We use NOAA’s Global Reports (2010.01–2025.03) featuring visualized anomalies and regional summaries. Following LLaVA's methodology, we use ChatGPT-4o to generate QA pairs. Seasonal inputs include all 12 monthly reports from a given year; interannual tasks use annual reports from 2010–2024.
> ---
> ## Q2: VQA tasks does not align well with the nature of real-world Earth science problems.
> Thank you for your valuable feedback. We’d like to clarify a few key points:
> 1. Our work focuses on applying MLLMs to vision-language understanding and reasoning in Earth science, rather than traditional regression tasks like weather forecasting.
> 2. Inspired by QA-style benchmarks such as WeatherQA, ClimateIQA, VRSBench, and XLRSBench, our goal is to address their limitations and develop a more comprehensive evaluation framework. Reviewer tWpg also mentioned similar single-sphere benchmarks like AgroMind, DynamicVL, and DisasterM3. We provide a comparison table to highlight the strengths and gaps across these efforts.
>
> |Dataset|OmniEarth-Bench|ArgoMind|DynamicVL|DisasterM3|WeatherQA|ClimateQA|CLLMate|VRSBench|XLRS-Bench|
> |-|-|-|-|-|-|--|-|-|-|
> |Task Dimensions Volume|**100**|13|14|9|2|4|1|12|16|
> |Data Sourcing Volume|**33**|9|1|5|1|2|2|2|6|
>
> Compared to these VQA-style benchmarks, OmniEarth-Bench offers broader coverage in both evaluation dimensions and data sources.
>
> 3. Even in multiple-choice settings, current MLLMs underperform, often scoring below 35%. In simple prediction tasks—such as determining whether an event will occur—models fall well short of human expert performance. For example, in our ENSO, IOD (as noted by reviewer tWpg) tasks, the model chooses from three options: "Will occur," "Will not occur," and "Unable to judge." Domain experts consistently achieve over 90% accuracy, while the latest MLLMs remain below 30%. This large gap highlights the need for MLLMs to first master these fundamental tasks.
> ---
> ## Q3: It is unclear why the proposed dataset covers such dense prediction tasks.
> 1. Localization and detection are widely recognized evaluation dimensions in MLLMs. Models like Qwen-VL2.5 report visual grounding performance on ODinW, PointGrounding, and RefCOCO datasets, while InterVL3 provides results on RefCOCO and RefCOCOg. Benchmarks such as XLRS-Bench (CVPR 2025) and VRSBench (NeurIPS 2024) also include visual grounding tasks, confirming its value as a standard metric for assessing MLLM capabilities.
> 2. OmniEarth-Bench incorporates dedicated visual grounding tasks across diverse spheres: Salt Body Location (Lithosphere), Eddy Localization (Oceansphere), and Visual Grounding of Land Types (Human Activities Sphere).
> * **Salt Body Location**: Identifying the spatial coordinates or depth of salt bodies in geological formations, assessing MLLMs’ spatial reasoning and multi-dimensional data integration. *Buoyancy-driven stratigraphic inversion forming km-scale mounds and sinkites* (Communications Earth & Environment, 2025) emphasizes the role of salt bodies and sinkites in stratigraphy, reservoir assessment, and CO₂/oil storage.
> * **Eddy Localization**: Enhances eddy tracking by pinpointing eddy locations, supporting rescue efforts and pollution control. *Oceanic mesoscale eddies as crucial drivers of global marine heatwaves* (Nature, 2023) highlights the importance of eddies in marine heatwave dynamics and the need for eddy-resolving ocean models.
> * **Visual Grounding of Land Types**:  Identifies specific land types to evaluate visual localization and classification capabilities. *Canopy functional trait variation across Earth’s tropical forests* (Nature, 2025) combines remote sensing, terrain, climate, and soil data, using localization and uncertainty analysis to refine global vegetation maps.
>
> 3. We conducted a detailed evaluation of models’ visual localization capabilities and reported the Acc@0.5 and Acc@0.7,following the XLRS-Bench and VRSBench.
>
> |Task|Num|Qwen2.5-VL-7B|LLaVA-OneVision-7b|InternVL3-8B|InternVL3-78B|GPT-4o|Gemini-2.0|Claude-3-7|
> |-|--|--|--|-|-|-|-|-|
> |Salt Body Location|302|0/0|5.3/0.33|8.94/1.66|4.3/0.33|0.08/0|0.13/0.04|0.02/0|
> |Eddy Localization|166|3.01/0|1.81/0.6|6.63/0.6|13.86/3.61|0.12/0.01|0.34/0.06|0.2/0.07|
> |Visual Grounding of Land Types|508|0.2/0|2.56/0.59|2.56/0.59|2.36/0.2|0.02/0|0.03/0|0/0|
>
> Visual grounding performance is notably poor across all models It exposing two main shortcomings: limited geoscientific knowledge and weak visual localization capabilities. Both open- and closed-source models fall short in these aspects.
>
> ---
> ## Q4: Most of the earth science problems are hard to convert to a VQA task.
> We’d like to clarify a few key points:
> 1. Firstly, we would like to clarify again that our work focuses on applying MLLMs to vision-language understanding and reasoning in Earth science, rather than traditional regression tasks like weather forecasting. Inspired by QA-style benchmarks such as WeatherQA, XLRSBench and AgroMind, DynamicVL, DisasterM3 (mentioned by Reviewer tWpg), our goal is to address their limitations and develop a more comprehensive evaluation framework. We provide a comparison table to highlight the strengths and gaps across these efforts.
> 2. Numerical reasoning is a key area for future MLLM advancements. *Carbon Flux Monitoring*, a representative numerical regression task, involves predicting values from observational images—an area now being explored by models like QwenVL2.5 and InterVL3. To this end, we find it crucial to assess vertical reasoning in Earth science. Our benchmark results show InterVL3 reaching 24.85% accuracy, highlighting its potential for such tasks.
>
> ---
> **Overall, thank you again for your constructive feedback. If you have any further questions, please don’t hesitate to let us know—we would be happy to continue the discussion.**

---

> > ### Comment · Reviewer_vtjt · 2025-08-04
> >
> > Thanks for the author's response. I think Q3 has been well addressed. For Q2 and Q4, authors should narrow their claims to Earth-related QA rather than a "Holistic Evaluation of Earth’s Six Spheres and Cross-Spheres Interactions". Because I believe the perfect models that can resolve this QA benchmark are still struggling with true earth science problems.
> >
> > Regarding Q1, it is still unclear how conventional VLMs can handle multimodal inputs, e.g., how to benchmark Qwen2.5-VL-7B on the problem requiring multispectral inputs. If all methods benchmarked on this dataset only accept RGB input, it is hard to know whether the benchmark reflects the true performance of the current models.

---

> > > ### Author Response · Authors · 2025-08-05
> > >
> > > ## Q1:Narrow the claims to Earth-related QA
> > > We agree with your insights. Despite our substantial efforts—including 2 to 5 domain experts per sphere and 40 crowd-sourced annotators working together for over four months—we acknowledge that OmniEarth-Bench covers only representative and prototypical Earth science scenarios. We sincerely appreciate your suggestions and will refine OmniEarth-Bench accordingly:
> > >
> > > 1. We agree that OmniEarth-Bench should emphasize in the title, abstract, and introduction that it represents the “Earth-related QA”. Rather than claiming comprehensive coverage, we will narrow the claimed scope and clearly highlight that this is the first evaluation of MLLMs on Earth-related observational data.
> > > 2. We also exploring the expansion of new task dimensions and has discussed their feasibility with domain experts. For instance, we could introduce additional tasks in the Ocean sphere.
> > > Identify or predict a broader range of extreme events—such as Atlantic Niño events[1], which are characterized by anomalous interannual oceanic warming or cooling in the tropical Atlantic Ocean, as well as marine heatwaves (MHWs), which are prolonged periods of unusually warm sea temperatures that can disrupt marine ecosystems and climate patterns[2]. These efforts focus on characterizing key metrics such as event type, intensity, and duration. We hope to further expand OmniEarth-Bench in the future to achieve broader coverage.
> > >
> > > [1] A decadal climate variation in the tropical Atlantic Ocean from thermodynamic air-sea interactions, Nature.
> > > [2] Socioeconomic impacts of marine heatwaves: Global issues and opportunities, Science.
> > >
> > > ## Q2: How conventional VLMs can handle multimodal inputs
> > >
> > > As you correctly noted, current MLLMs are unable to process multispectral imagery. We conducted a detailed examination of representative MLLMs such as QwenVL.5, InternVL3, and LLaVA-OneVision, and found that none of them accept multispectral inputs, highlighting a fundamental limitation at this stage of MLLM development. Despite their limitations, MLLMs are broadly acknowledged for their potential in Earth science. Accordingly, we seek to assess their current capabilities.
> > >
> > > To clarify, the “33 data sources” referenced in our paper denote 33 distinct origins of raw data—not 33 different data modalities (e.g., spectral bands or 1D signals). These data sources were processed by domain experts and converted into MLLM-compatible formats such as PNG, with a focus on maximizing information retention and ensuring fair evaluation.
> > >
> > > To support MLLM evaluation, we converted all multispectral image inputs into visible using two main strategies:
> > >
> > > 1. **When multispectral satellite data was essential**, we applied:
> > >
> > >    * **RGB band synthesis**: We extracted the red, green, and blue bands to synthesize RGB images from some multispectral data sourcing. While this discards other spectral bands, RGB still retains key visual cues. This method is effective for tasks less sensitive to full spectral information and works well with sources like SatBird (4 bands), where experts judged minimal loss from dropping one band.
> > >    * **Single-band channel mapping**: For tasks where each spectral band carries critical information, we split each channel into separate grayscale images (0–255 pixel values, PNG format). For example, in the biosphere, 7-band MODIS imagery was separated into individual inputs, with detailed descriptions of each band clearly noted in the question prompts text. This preserves the spectral richness while adapting to MLLM constraints.
> > >
> > > 2. **When multispectral data was optional**, we prioritized RGB imagery. In the Anthroposphere, for instance, the L2-level evaluations of urban development used RGB images. Although multispectral alternatives like fMoW-S2 [1] exist, we opted for RGB datasets such as UBCv1 and BHdataset to maintain compatibility.
> > >
> > > In summary, these strategies ensured that our benchmark preserves essential content while conforming to MLLM input requirements, enabling reliable and representative evaluation across all tasks.
> > >
> > > **Thank you again for recognizing our work. We are committed to making these improvementsand sincerely appreciate your insightful feedback. Please let us know if there is any other question.**

---

> > > > ### Comment · Reviewer_vtjt · 2025-08-06
> > > >
> > > > Thanks for the author's further clarification. "33 sensor types" in Figure 1 and "we compiled a comprehensive dataset covering 33 different data modalities" in line 154 indeed existed in the manuscript. These claims would mislead readers, but are easy to fix.

---

> > ### Author Response · Authors · 2025-08-07
> >
> > Thank you again for recognizing our work. In the final version, we will change the Figure 1 and line 154. We sincerely appreciate your insightful feedback. If you have any further questions, please let us know promptly so that we can resolve them in the remaining time. We hope you will reconsider our score. Thank you again for your time and positive consideration of our rebuttal.

---

> > > ### Comment · Reviewer_vtjt · 2025-08-07
> > >
> > > I raised my score in the first round :)

---

> > > > ### Author Response · Authors · 2025-08-07
> > > >
> > > > We would like to express our sincere gratitude for your positive score and further support towards our work! Your support is a tremendous source of encouragement for our team!

---

### Author Response · Authors · 2025-08-09

Dear All Reviewers and Area Chair:

We appreciate the reviewers’ time and the conference’s commitment to a fair and constructive review. We thank the Area Chair for overseeing the process and the reviewers for engaging with us. At this stage, **three reviewers have no remaining concerns, while one ceased responding after the first round.** Below is a brief summary of our interactions:

* **Reviewer vtjt**: Initially raised four issues, mainly on `coverage scope` and `handling complex multispectral data`. After discussion, we agreed to narrowing our claim to Earth science observational data and Reviewer accepted our explanation on spectral processing. **Reviewer ultimately gave a positive score.**

* **Reviewer tWpg**: Discussed in depth the significance of Earth science evaluation in light of *Nature*/*Science*-level work, affirming that our evaluation dimensions are highly representative. Reviewer later inquired about `evaluation MCQ formats`; after our explanation and new experiments, Reviewer acknowledged the value of expanding from MCQ to open-ended questions and **gave a more favorable score.**

* **Reviewer qv8C**: Reviewer focused on our method of `handling complex data`. Following our explanations and examples, Reviewer qv8C—like Reviewer vtjt—recognized our contributions to processing multi-source complex data and **confirmed their concerns were resolved.**

* **Reviewer yRtC**: Engaged in one active round. In the second-round comments, they questioned the `MCQ format`, `task coverage`, and `CoT’s motivation and design`. **The first two concerns had already been addressed to the satisfaction of Reviewer vtjt and Reviewer tWpg**(via visual grounding, open-ended formats, and narrowed claims). Regarding CoT:

  1. **Motivation**: CoT is needed as Earth science observational data lacks such evaluation in challenging dimensions to assess reasoning quality in incorrect outputs.
  2. **Design**: Clearly detailed during rebuttal, with annotation paradigms enabling any team to extend CoT annotations.
  3. **Examples**: Due to space limits, CoT examples will be added in the appendix; all dataset samples of CoT remain openly accessible.

Across multiple rounds, each reviewer engaged at least once. **Three fully endorsed our work and agreed their concerns were resolved**, with one offering high praise for its forward-looking nature and contributions across spheres. The remaining reviewer yRtC’s issues, though discussed only once, are ones we believe we have addressed.

**Importantly, our work is conducted by an interdisciplinary team of computer science researchers and many experts of six spheres . Our shared goal is to advance the application of MLLMs Earth Science, a collaboration that is both rare and valuable to the field. We believe this perspective strengthens the submission's potential impact.**

We once again thank all reviewers for their active participation and the AC for diligently overseeing the review process. We will further improve the paper based on all reviewers’ feedback. We sincerely appreciate everyone’s recognition and the effort invested during this period.

Best regards,

Authors of Submission #920

---

### Note · Authors · 2025-08-12

Dear All Reviewers and Area Chair:

We sincerely thank the AC and all reviewers for the opportunity to present our work again, for the reviewers’ active participation, and for the AC’s diligent oversight of the review process. Three reviewers have no remaining concerns, while one ceased responding after the first round.

- Reviewer `tWpg` offered strong praise from the start and, after further discussions on topics such as MCQ formats, expressed even greater appreciation for our work.

- Reviewers `vtjt` and `qv8C` engaged in thorough, constructive discussions and ultimately aligned with our perspectives, noting *“I raised my score”* and *“Most of my concerns have been addressed.”*

- For Reviewer `yRtC`, most issues raised in their second-round comments had already been considered resolved by others (e.g., Reviewers `vtjt` and `tWpg` acknowledged our inclusion of visual grounding, open-ended formats, and narrowed claims).

**We deeply appreciate everyone’s recognition and the effort devoted during this process.**


Best regards,

Authors of Submission #920

---

### Decision · Program_Chairs · 2025-09-18

**Decision:**

Reject

**Comment:**

The authors propose OmniEarth‑Bench, a large-scale benchmark for evaluating multimodal LLMs on Earth‑science problems. This benchmark leverages data from satellite and in‑situ sensors and covers all six Earth spheres and their interactions, with 100 expert‑curated tasks and 29,779 annotated examples spanning perception, general reasoning, scientific knowledge reasoning and chain‑of‑thought (CoT) tasks.  The extensive scope and careful curation was appreciated by most reviewers: three of four ultimately felt the work fills a major gap by unifying diverse Earth‑observation tasks, exposing the inability of current MLLMs to exceed 35 % accuracy and achieving 0 % accuracy on some cross‑sphere tasks.  Reviewers particularly valued the inclusion of all spheres, cross‑sphere queries, CoT annotations, and the use of expert‑crowd workflows, noting that releasing the data and code will enable reproducibility.  The main weaknesses raised were over‑claiming of coverage (some tasks are narrow and the design leans heavily on multiple‑choice VQA), reliance on RGB visualisations rather than native modalities, and unclear details on processing pipelines and baseline adaptation; one reviewer felt these limitations make the benchmark less representative of forecasting and regression tasks.  The rebuttal clarified data transformation steps, narrowed claims to Earth‑observation data, introduced visual grounding and open‑ended QA formats, and committed to releasing full processing code and CoT annotation paradigms.  In the ensuing discussion, two reviewers raised their scores and the third expressed satisfaction, while the remaining reviewer did not engage further.  Given the novelty and breadth of the benchmark, its potential to spur geosystem‑aware AI research, and the authors’ responsive clarifications, the AC concludes that the strengths outweigh the remaining concerns and recommends acceptance, encouraging the authors to temper their claims and release complete processing scripts and documentation as well as including the promised changes.

===== FINAL UPDATE FROM DB Track PCs ====

The final decision for this paper has been taken by the program chairs after consultation with the SACs. All Senior Area Chairs have ranked papers according to the feedback from the AC during the review process. We decided to leave the original meta-review to reflect the opinion of the AC in light of the initial discussions with reviewers and SAC.